# Sample Margin-Aware Recalibration of Temperature Scaling

**Haolan Guo** [*1]   **Linwei Tao** [*1]   **Haoyang Luo** [2]   **Minjing Dong** [2]   **Chang Xu** [1]

## Abstract

Deep neural networks frequently exhibit over-confidence, undermining reliability in safety-critical applications. Existing adaptive methods rely on indirectly learned proxies of sample difficulty. We establish the *logit margin* as a direct and principled hardness indicator. We prove that margin constrains the feasible temperature range for a target confidence. Empirically, margin strongly correlates with decision boundary proximity and reveals systematic calibration patterns across difficulty levels. We further identify a fundamental flaw in NLL-based optimization: minimizing NLL can paradoxically worsen calibration. To address this, we introduce Charbonnier-SoftECE, a smooth objective that provably upper-bounds the smooth calibration error (smCE). Building on these insights, we propose SMART (Sample Margin-Aware Recalibration of Temperature), a lightweight method that learns a sample-wise margin-to-temperature mapping guided by our calibration-centric objective. Experiments demonstrate state-of-the-art calibration across CNNs and ViTs on standard, long-tailed, and distribution-shifted benchmarks, with minimal inference-time overhead. Code is available at: `https://github.com/Misakaaaaaz/ICML2026-SMART`.

## 1. Introduction

Deep neural networks have achieved remarkable success across diverse domains, but safety-critical deployment (e.g., autonomous driving (Feng et al., 2019), medical diagnosis (Zwaan & Hautz, 2019)) requires reliable uncertainty, i.e., *calibration* (Guo et al., 2017). Modern models are often miscalibrated, typically overconfident (Guo et al., 2017;

Wei et al., 2022; Luo et al., 2025), which can cause high-confidence errors with severe consequences.

Calibration methods are broadly training-time or post-hoc. Training-time approaches use data augmentation, regularization, or modified losses (Hendrycks et al., 2019; Müller et al., 2019; Mukhoti et al., 2020; Tao et al., 2023), but are inconvenient for already-trained models. Post-hoc methods (Zadrozny & Elkan, 2002; 2001) are plug-and-play; temperature scaling (TS) (Guo et al., 2017) is especially popular but applies a single global temperature and ignores sample-wise heterogeneity. Extensions use class-wise temperatures (Frenkel & Goldberger, 2021), grouping (Yang et al., 2023), or sample-adaptive signals (Ding et al., 2021; Tomani et al., 2022).

Xiong et al. (2024) calibrate predictions based on sample proximity, assigning larger temperatures to less proximate samples; Yang et al. (2023) apply larger temperatures to groups that are harder to distinguish (e.g., birds and airplanes sharing the same background in CIFAR-10); and Ding et al. (2021) exploit feature-space sparsity to adaptively guide temperature. Despite their methodological differences, these approaches share the same underlying motivation: sample hardness drives calibration. However, they rely on *indirectly learned* proxies of difficulty. In contrast, we propose a direct and simple measure, the *logit margin*, defined as the gap between the largest and second-largest logits. Empirical results on ImageNet/ViT-B/16 (Figure 1b–d) show that margin groups reveal different calibration behavior at similar confidence levels, with higher-margin groups tending to be more under-confident. Moreover, the strong correlation between the margin and the minimum perturbation required to reach the decision boundary under attack (Figure 1a) highlights the margin's reliability as a hardness indicator. Finally, our theoretical analysis in Appendix A.1 demonstrates that the feasible temperature for a target confidence is constrained by the margin, underscoring its effectiveness as a principled signal of sample difficulty for post-hoc calibration.

Another limitation of many scaling-based methods is their focus on optimizing the NLL loss, which theoretically does not guarantee a reduction in calibration error. In fact, as we prove in Appendix A.4, there exist SMART-feasible local scalings that strictly decrease NLL while worsening a

---

[*]Equal contribution   [1]University of Sydney   [2]City University of Hong Kong. Correspondence to: Haolan Guo <hguo4658@uni.sydney.edu.au>.

*Proceedings of the $43^{rd}$ International Conference on Machine Learning*, Seoul, South Korea. PMLR 306, 2026. Copyright 2026 by the author(s).

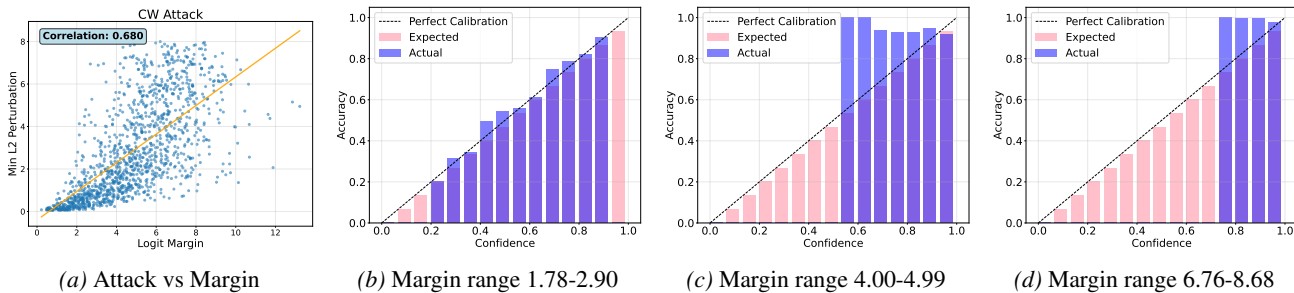

*(a)* Attack vs Margin     *(b)* Margin range 1.78-2.90     *(c)* Margin range 4.00-4.99     *(d)* Margin range 6.76-8.68

*Figure 1.* Margin-based calibration behavior. Panel (a) shows the relationship between C&W minimum perturbation (Carlini & Wagner, 2017) and logit margin on CIFAR-10. Panels (b–d) show reliability diagrams for ImageNet/ViT-B/16 margin groups; the diagonal denotes perfect calibration, curves above it indicate under-confidence, and curves below it indicate over-confidence.

smooth calibration criterion, demonstrating that likelihood optimization can move against calibration. To address this misalignment, we adopt Charbonnier-SoftECE. This objective directly targets calibration and, as established by our theoretical analysis in Appendix A.2, provably upper-bounds the smooth calibration error (smCE), aligning optimization with calibration improvement. We use Charbonnier-SoftECE and SmoothSoftECE interchangeably in this paper.

Building on these insights, we introduce **S**ample **M**argin-**A**ware **R**ecalibration of **T**emperature (SMART), a lightweight post-hoc calibration method that learns a direct mapping from logit margins to temperatures. Using Charbonnier-SoftECE as its learning objective, SMART directly optimizes a calibration-controlled reliability objective. Experiments on various benchmarks and architectures validate the state-of-the-art effectiveness and efficiency of SMART, even with a small validation set.

**Contributions.** Our work is theory-driven and makes three key contributions:

(i) We provide formal and empirical analysis showing that logit margin is a principled hardness indicator that tightly constrains the feasible temperature range, outperforming existing proxies with minimal computation (Section 3.2).

(ii) We prove a fundamental mismatch between NLL optimization and calibration quality, and address it through a Charbonnier-SoftECE objective that provably upper-bounds smooth calibration error (Sections 3.3–3.4).

(iii) We develop SMART, a lightweight margin-aware temperature mapping that achieves state-of-the-art calibration on CNNs and ViTs across long-tailed and out-of-distribution datasets, remaining effective with as few as 50 validation samples (Sections 3.5 and 4).

## 2. Related Work

**Post-hoc and adaptive recalibration.** Post-hoc methods keep the classifier fixed and fit a held-out calibrator. Classical non-parametric methods learn score-to-accuracy maps

by binning confidence scores (Zadrozny & Elkan, 2001), averaging over quantile-binning models with Bayesian evidence (Naeini et al., 2015), fitting monotone isotonic maps (Zadrozny & Elkan, 2002), or using splines to approximate the empirical recalibration function (Gupta et al., 2021). Parametric alternatives include Platt Scaling, which fits a sigmoid to raw scores (Platt, 1999); Beta calibration, which handles skewed binary score distributions (Kull et al., 2017); Dirichlet calibration, which extends this idea to multiclass probabilities through a log-probability linear map (Kull et al., 2019); and Temperature Scaling (TS), a single-parameter logit rescaling that is simple and accuracy-preserving (Guo et al., 2017). Later adaptive methods increase expressiveness by conditioning the correction on classes, predictions, inputs, or learned structure: class-based TS learns one temperature per class (Frenkel & Goldberger, 2021), PTS predicts prediction-specific temperatures with a neural network (Tomani et al., 2022), AdaTS estimates sample-dependent temperatures (Joy et al., 2023), LTS predicts spatially local temperatures for segmentation (Ding et al., 2021), semantic grouping learns subset-specific calibrators (Yang et al., 2023), ProCal corrects proximity bias (Xiong et al., 2024), Feature Clipping reduces over-confident feature magnitudes (Tao et al., 2025), and Mix-n-Match composes or ensembles off-the-shelf calibrators (Zhang et al., 2020). TS4CP studies temperature scaling for conformal prediction sets (Dabah & Tirer, 2025), rather than probability recalibration itself. This line shows that one global confidence map is often too coarse, but most additional signals are class-level, high-dimensional, spatial/semantic, or indirectly learned. This motivates a post-hoc calibrator that keeps the accuracy-preserving simplicity of TS while using an explicit low-dimensional hardness signal; SMART uses the top-two margin and justifies this choice through margin-temperature bounds.

**Training-time and objective-level calibration.** Training-time methods alter the data, model, or loss before deployment. Mixup trains on convex combinations of examples and labels (Thulasidasan et al., 2019), AugMix improves ro-

bustness and uncertainty under corruptions through stochastic augmentations and consistency (Hendrycks et al., 2019), ensembles average multiple predictive distributions (Lakshminarayanan et al., 2017), label smoothing softens targets to discourage over-confidence (Müller et al., 2019), and focal losses down-weight easy examples to improve confidence behavior (Mukhoti et al., 2020; Gupta et al., 2020). Recent calibration-specific losses include Dual Focal loss, which reshapes focal learning around competing logits (Tao et al., 2023); MDCA, which adds a multi-class confidence-accuracy regularizer (Hebbalaguppe et al., 2022); UWG, which weights gradients by sample uncertainty (Lin et al., 2025); and Relaxed Softmax, which modifies the softmax layer for confidence auto-calibration during detector training (Neumann et al., 2018). A related objective-level line makes calibration errors differentiable: MMCE uses an RKHS kernel calibration penalty (Kumar et al., 2018), AvUC optimizes the accuracy-versus-uncertainty relationship (Krishnan & Tickoo, 2020), Meta-Calibration optimizes a differentiable ECE surrogate (Bohdal et al., 2021), and SoftECE replaces hard bins with soft assignments for direct calibration optimization (Karandikar et al., 2021). These methods are complementary, but they typically require retraining, architectural changes, or an objective surrogate without specifying a compact post-hoc temperature signal. SMART is therefore not simply a SoftECE-style smoothing variant: it combines a calibration-centric Charbonnier-SoftECE objective with a margin-based post-hoc temperature map.

**Evaluation metrics.** Expected Calibration Error (ECE) (Guo et al., 2017) remains widely used, but its limitations include bin-size sensitivity (Nixon et al., 2019; Kumar et al., 2019; Roelofs et al., 2022; Gupta et al., 2020). SmoothECE addresses binning discontinuity by smoothing observations with an RBF kernel before computing ECE and reliability diagrams (Błasiok & Nakkiran, 2024), while Blasiok et al. (2023) connect proper losses to smooth calibration. ECE is not a proper scoring rule; unlike NLL/log loss (Gruber & Buettner, 2022), it measures the reliability gap that post-hoc calibration primarily aims to reduce. For a fair comparison, we therefore report ECE, Adaptive-ECE (Ding et al., 2020), and classwise-ECE (Kull et al., 2019), while using NLL only as a complementary likelihood diagnostic where reported.

## 3. Methodology

We first present preliminaries in Section 3.1, then establish margin as a principled hardness indicator in Section 3.2. We identify fundamental limitations of NLL-based calibration objectives in Section 3.3, introduce our Charbonnier-SoftECE objective in Section 3.4, and present the SMART framework in Section 3.5.

### 3.1. Preliminaries

A classification model is *calibrated* if its predictive confidence matches its actual accuracy. For classifier $f_\theta$, input $\mathbf{x}$ with true label $y$, and predicted class $\hat{y}$, perfect calibration requires $\mathbb{P}(y = \hat{y} \mid p_\theta(\hat{y} \mid \mathbf{x}) = p) = p$ for all confidence values $p \in [0, 1]$.

**Expected Calibration Error (ECE).** For classification model $f_\theta$ producing logits $\mathbf{z}_i \in \mathbb{R}^K$, the predictive probability for class $k$ is $p_\theta(y_i = k \mid \mathbf{x}_i) = \frac{\exp(z_{i,k})}{\sum_{j=1}^{K} \exp(z_{i,j})}$. To quantify calibration error, we partition samples into $B$ bins based on predicted confidence, compute average accuracy $\hat{a}_b$ and confidence $\hat{p}_b$ within each bin $b$, and measure their difference, where $I_b$ is the set of indices in bin $b$ and $N$ is the total number of samples:

$$\text{ECE} = \sum_{b=1}^{B} \frac{|I_b|}{N} |\hat{p}_b - \hat{a}_b|, \tag{1}$$

**Smooth Calibration Error (smCE).** Beyond binned ECE which suffers from discretization artifacts, we also consider the smooth calibration error (Blasiok et al., 2023), defined as the worst-case correlation between the calibration residual and 1-Lipschitz probes of predicted confidence:

$$\text{smCE}(f) := \sup_{\varphi \in \mathcal{H}} \left| \mathbb{E}\big[ (a(X) - p(X)) \varphi(p(X)) \big] \right|, \tag{2}$$

where $\mathcal{H} = \{\varphi : [0, 1] \to [-1, 1] \mid \text{Lip}(\varphi) \leq 1\}$ is the class of 1-Lipschitz continuous functions, $p(X)$ denotes the predicted confidence (maximum softmax probability), and $a(X) = \mathbb{I}\{\hat{y}(X) = y\}$ is the correctness indicator. This continuous metric avoids binning artifacts and provides theoretical foundation for our objective design in Section 3.4.

ECE and smCE/SoftECE are calibration-centric metrics: they directly measure confidence-reliability mismatch, but they are not proper scoring rules. Likelihood-based proper scores such as NLL evaluate overall probabilistic quality by combining calibration with sharpness and discrimination, so they can disagree with ECE. SMART therefore optimizes and reports calibration-oriented objectives, without claiming universal optimality for every probabilistic score.

**Temperature Scaling.** Temperature scaling (TS) (Guo et al., 2017) introduces positive scalar $T$ to adjust logit distribution before softmax: $p_{\theta,T}(y_i = k \mid \mathbf{x}_i) = \frac{\exp(z_{i,k}/T)}{\sum_{j=1}^{K} \exp(z_{i,j}/T)}$. Smaller temperature $T < 1$ sharpens the distribution, while larger $T > 1$ flattens it. Vanilla TS finds global $\hat{T} = \arg\min_{T>0} \mathcal{L}_{\text{NLL}}(\mathcal{D}_{val}, f_\theta, T)$ by minimizing NLL on a validation set.

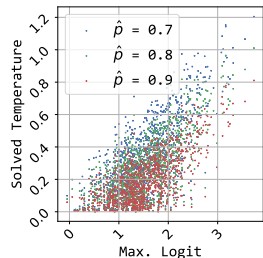 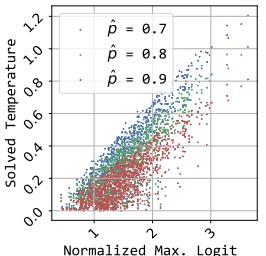 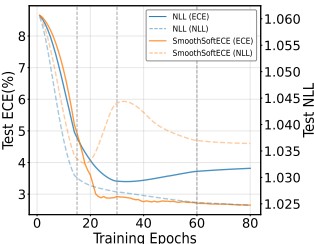

*Figure 2.* **Numerical study of temperature adjustment indicators.** The **left** three panels show a synthetic numerical validation with 1,000 random 10-class logit vectors, plotting solved temperature $T$ against candidate indicators. **Right:** Test ECE and NLL during training on ImageNet ViT-B/32.

### 3.2. Margin as a Principled Hardness Indicator

Effective post-hoc calibration requires distinguishing between easy and hard samples to apply appropriate confidence adjustments. While existing methods (Xiong et al., 2024; Yang et al., 2023) recognize this need, they rely on indirectly learned proxies such as feature-space proximity or semantic clustering. We propose using the logit margin $m = z_{\max} - z_{2\text{nd}}$ as a direct hardness indicator, where $z_{\max}$ and $z_{2\text{nd}}$ are the largest and second-largest logits.

As demonstrated in Figure 1, grouping samples by margin reveals different calibration behavior even at comparable confidence levels, with high-margin groups tending to be more under-confident in this setting. Margin also correlates strongly with adversarial robustness ($r = 0.68$), confirming it captures proximity to decision boundaries. We now establish theoretically why margin provides tighter temperature control compared to alternative indicators.

For a given logit vector $\mathbf{z} \in \mathbb{R}^K$ and target confidence $\hat{p} \in (0, 1)$, the temperature-confidence relationship $\frac{e^{z_{\max}/T}}{\sum_{k=1}^K e^{z_k/T}} = \hat{p}$ can be rearranged as $\sum_{k \neq M} e^{(z_k - z_{\max})/T} = S$ where $S := \frac{1}{\hat{p}} - 1$ and $M = \arg\max_k z_k$. Given only $z_{\max}$ and target $\hat{p} > 1/K$, we can construct configurations where all non-maximum logits equal $z_{\max} - \delta$ for varying $\delta > 0$, yielding $T = -\delta/\log(S/(K-1))$ which sweeps $(0, \infty)$ as $\delta$ varies. Thus maximum logit alone provides no bound on feasible $T$.

In contrast, margin provides tight constraints because it fixes the largest competitor's distance from the top logit, thereby bounding every non-maximum contribution to the softmax. For any non-maximum class $k$, we have $z_k \leq z_{2\text{nd}} = z_{\max} - m$, leading to bounds $e^{-m/T} \leq S \leq (K-1)e^{-m/T}$. Solving for $T$ yields: when $\hat{p} > 1/2$, $T \in \left[\frac{m}{-\log(S/(K-1))}, \frac{m}{-\log S}\right]$ (finite interval); when $1/K < \hat{p} \leq 1/2$, $T \in \left[\frac{m}{-\log(S/(K-1))}, +\infty\right)$ (finite lower bound). The interval width decreases as $m$ grows, and for binary classification the bounds coincide to uniquely determine $T$. Complete derivations appear in Appendix A.1.

Figure 2 (left three panels) validates these results empirically in a synthetic numerical setting. We sample 1,000

random 10-class logit vectors and numerically solve for temperatures achieving $\hat{p} = 0.8$. Margin exhibits clear functional structure with $T$ tightly constrained, while maximum logit and normalized maximum logit display scattered patterns spanning orders of magnitude. This supports margin as a more constrained scalar indicator for temperature-based calibration.

### 3.3. The NLL-Calibration Mismatch

Current post-hoc calibration methods optimize negative log-likelihood (NLL) under the assumption that minimizing NLL improves calibration. We demonstrate this assumption can fail. Figure 2 (rightmost panel) illustrates the phenomenon through a controlled experiment where we train SMART's margin-based temperature network on ImageNet ViT-B/32 using NLL as the training objective. While NLL decreases monotonically throughout 80 epochs, ECE begins increasing after epoch 30, creating clear divergence between objectives. By epoch 80, NLL has decreased by 15% while ECE has increased by 8% relative to epoch 30. This shows that following NLL gradients can actively worsen calibration despite improving likelihood.

We formalize conditions under which NLL and calibration objectives have opposing local preferences. Consider a baseline SMART map $h$ with temperature $T(x) = h(m(x))$, and a margin slice $G \subset [m_{\min}, m_{\max}]$ defining sample region $A := \{x : m(x) \in G\}$ where $m(x) = z_{(1)}(x) - z_{(2)}(x)$ is the margin between top two logits. For a local scale factor $s > 0$, let $h_s$ be the perturbed map that induces $T_s(x) = T(x)/s$ if $x \in A$ and $T_s(x) = T(x)$ otherwise. Since positive temperature scaling preserves logit ordering, $M_s(X) := \arg\max_k z_k(X)/T_s(X) = M(X)$; we write this shared top index as $M$. At baseline $s = 1$, analyzing how objectives change as $s$ varies reveals their directional preferences.

Define $t_k := z_k/T$ (scaled logit at $s = 1$), $q_k := \frac{e^{t_k}}{\sum_j e^{t_j}}$ (predicted probability), $p := q_M$, $p_s$ as the top-class confidence under $T_s$, $\langle t \rangle_q := \sum_k q_k t_k$ (expected scaled logit), and $r_X(X) := \mathbb{P}(Y = M(X) \mid X)$ (pointwise top-class probability). For NLL $L_{\text{nll}}(h_s)$ and calibration functional

$\mathcal{C}[\psi] = \mathbb{E}[(a(X) - p_s)\psi(p_s)]$ with smooth probe $\psi$, the directional derivatives at $s = 1$ are:

$$\frac{d}{ds}L_{\text{nll}}(h_s)\Big|_{s=1} = \mathbb{E}\big[\mathbb{I}_A(\langle t\rangle_q - t_Y)\big], \tag{3}$$

$$\frac{d}{ds}\mathcal{C}[\psi]\Big|_{s=1} = \mathbb{E}\Big[\mathbb{I}_A\, p(t_M - \langle t\rangle_q)$$
$$\cdot \big(\psi'(p)(r_X(X) - p) - \psi(p)\big)\Big], \tag{4}$$

where $\mathbb{I}_A$ indicates the margin slice, $t_M$ is the scaled logit of the top class, and $t_Y$ is that of the true class. The NLL gradient depends only on whether $t_Y$ falls below the softmax-averaged logit $\langle t\rangle_q$, while the calibration gradient is additionally sensitive to the calibration gap $r_X(X) - p$ through the probe $\psi$. These different sensitivities create potential for directional conflict.

Consider confidence slices $J_U, J_O \subset (p_0, 1)$ within the margin slice $A$, representing (on average) underconfident and overconfident regions. Define

$$\mu_U := \mathbb{P}(X \in A,\, p(X) \in J_U),$$
$$\mu_O := \mathbb{P}(X \in A,\, p(X) \in J_O),$$
$$\mu_{\text{gap}} := \mathbb{P}(X \in A,\, p(X) \notin J_U \cup J_O),$$

and the corresponding *average* calibration gaps on these slices

$$\rho_U := \mathbb{E}[\, r_X(X) - p(X) \mid X \in A,\, p(X) \in J_U\,] > 0,$$
$$\rho_O := \mathbb{E}[\, p(X) - r_X(X) \mid X \in A,\, p(X) \in J_O\,] > 0.$$

Let $\gamma_{\min}, \gamma_{\max}$ bound the runner-up gap $g(x) := m(x)/T(x)$ on $A$, and let $\Delta_G$ bound the non-top logit spread on $A$ (Appendix A.4). When

$$\rho_U\, \gamma_{\min}\, \mu_U \;>\; \gamma_{\max}\big(\rho_O\, \mu_O + \mu_{\text{gap}}\big) \;+\; \Delta_G\, \mu_A,$$

the sharpening direction $s \uparrow 1$ strictly *decreases* NLL, i.e., $\frac{d}{ds}L_{\text{nll}}(h_s)\big|_{s=1} < 0$. At the same time, under mild regularity (bounded density of $p$ on $A$ and a slice-bounded advantage), one can construct a 1-Lipschitz probe $\psi$ supported on $J_O$ such that $\frac{d}{ds}\mathcal{C}[\psi]\big|_{s=1} > 0$, meaning this same move *increases* a calibration witness functional. Since smCE is the supremum over such probe correlations (Eq. (2)), this establishes that improving NLL provides *no guarantee* of improving smCE; in particular, NLL optimization can move in directions that increase probe-based lower bounds on smCE. Detailed constructions and bounds appear in Appendix A.4.

This fundamental misalignment explains why NLL-based methods can achieve good likelihood while maintaining poor calibration. The mismatch occurs when calibration benefits from sharpening underconfident predictions are

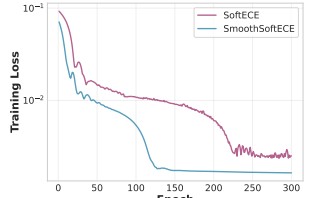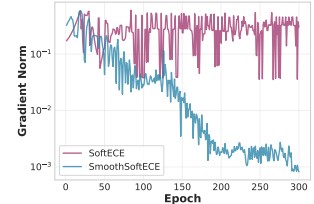

*Figure 3.* Post-hoc calibration on ImageNet ViT-B/16: training loss (left) and gradient norm (right) over training epochs for SoftECE and Charbonnier-SoftECE (ours).

outweighed by costs from further sharpening overconfident predictions, yet NLL gradients still favor overall sharpening due to their different sensitivity to confidence heterogeneity. This motivates developing objectives that directly target calibration error rather than likelihood.

### 3.4. Charbonnier-SoftECE Objective

Section 3.3 demonstrated that NLL optimization can conflict with calibration goals. We require a differentiable objective that directly targets calibration error while remaining statistically efficient with limited validation data. Current approaches face a bias-variance tradeoff: binned ECE has low variance but high bias from fixed binning, while point-wise losses have low bias but high variance from binary correctness indicators.

Following Karandikar et al. (2021), we adopt soft-binned ECE which balances this tradeoff through kernel smoothing. For sample $i$ with confidence $\hat{p}_i$ and bin centers $\{c_b\}_{b=1}^B$, soft membership weights

$$w_{i,b} = \frac{\exp(-\alpha(\hat{p}_i - c_b)^2)}{\sum_{b'}\exp(-\alpha(\hat{p}_i - c_{b'})^2)}$$

distribute each sample's contribution across neighboring bins, creating smooth gradients. In continuous formulation with Gaussian kernel $k_\lambda(t) = e^{-\lambda t^2}$ and a *uniform* reference density $\rho(u) \equiv 1$ on $[0, 1]$, this becomes:

$$\text{SoftECE}(f) := \mathbb{E}_X\left[\int_0^1 K_\lambda(p(X), u)|a(X) - u|\rho(u)du\right], \tag{5}$$

where

$$K_\lambda(p, u) = \frac{k_\lambda(p - u)}{\int_0^1 k_\lambda(p - v)\rho(v)dv}$$

is the normalized kernel, $p(X)$ is predicted confidence, and $a(X) = \mathbb{I}\{\hat{y}(X) = y\}$ is the correctness indicator.

We enhance SoftECE with Charbonnier smoothing to obtain a smooth objective with explicit calibration control. Replacing the absolute value with the Charbonnier function $\phi_\delta(r) = \sqrt{r^2 + \delta^2}$ yields:

$$\mathcal{H}_{\lambda,\delta}(f) := \mathbb{E}_X\left[\int_0^1 K_\lambda(p(X), u)\phi_\delta(a(X) - u)\rho(u)du\right]. \tag{6}$$

Since $\phi_\delta(r) \geq |r|$, $\mathcal{H}_{\lambda,\delta}$ is a smooth upper envelope of the absolute-value formulation. Our key theoretical contribution establishes that $\mathcal{H}_{\lambda,\delta}$ upper bounds the smooth calibration error up to a design-only kernel approximation term.

**Theorem 3.1** (Charbonnier-SoftECE Upper Bounds smCE). *Assume reference density $\rho$ satisfies $0 < \rho_{\min} \leq \rho(u) \leq \rho_{\max} < \infty$ for all $u \in [0,1]$ with condition number $\kappa := \rho_{\max}/\rho_{\min}$. Then for all classifiers $f$ and smoothing parameters $\delta \geq 0$:*

$$\text{smCE}(f) \leq \mathcal{H}_{\lambda,\delta}(f) + 2B_\lambda, \tag{7}$$

*where $B_\lambda := \sup_{p \in [0,1]} \int_0^1 |p-u| K_\lambda(p,u)\rho(u)du$ represents kernel approximation error. For Gaussian kernels with $\lambda \geq 1$, $B_\lambda \leq \frac{C_\kappa}{\sqrt{\lambda}}$ where $C_\kappa := \frac{2\kappa}{\sqrt{\pi}\,\text{erf}(1)} \approx 1.339\kappa$.*

The proof (Appendix A.2) decomposes smCE using mollification: for any 1-Lipschitz probe $\varphi$, we write $\mathbb{E}[(a-p)\varphi(p)]$ as a smooth term controlled by $\mathcal{H}_{\lambda,\delta}$ plus approximation error bounded by $B_\lambda$. The bound splits into a model-dependent term $\mathcal{H}_{\lambda,\delta}(f)$ that can be optimized and a design-only term $2B_\lambda$ that tightens as $O(1/\sqrt{\lambda})$. Thus minimizing $\mathcal{H}_{\lambda,\delta}$ directly minimizes an explicit upper bound on calibration error, addressing the NLL mismatch from Section 3.3.

Figure 3 demonstrates the practical benefits of Charbonnier smoothing. On ImageNet ViT-B/16, Charbonnier-SoftECE achieves faster training convergence (left panel) while maintaining stable gradient norms throughout optimization (right panel). Standard SoftECE exhibits oscillations in later training epochs. The Charbonnier enhancement thus provides theoretical control and improved optimization stability.

For computational efficiency, we discretize Equation (6) using $B=15$ soft bins with Gaussian kernel assignments (bandwidth $\sigma=0.05$, Charbonnier $\delta=10^{-3}$). The implemented bin-mean objective

$$\widetilde{\mathcal{H}}_{\sigma,\delta}^{\text{bin}}(f) := \sum_{b=1}^{B} \pi_b \Big( \phi_\delta(\hat{p}_b - \hat{a}_b) - \phi_\delta(0) \Big) \tag{8}$$

aggregates soft bin masses $\pi_b$ and bin-mean confidence/accuracy gaps. Under mild within-bin homogeneity conditions, this surrogate maintains the smCE upper bound guarantee of Theorem 3.1 (Corollary A.4 in Appendix A.3).

### 3.5. The SMART Framework

Building on the theoretical foundations established in Sections 3.2–3.4, we introduce SMART (Sample Margin-Aware Recalibration of Temperature), which learns a direct mapping from margin to temperature. Formally, SMART parameterizes a scalar temperature map $T(\cdot) : \mathbb{R}^+ \to \mathbb{R}^+$. The framework combines margin as the input indicator (Section 3.2) with Charbonnier-SoftECE as the training objective (Section 3.4).

SMART implements a lightweight two-layer MLP that maps logit margin $m = z_{\max} - z_{\text{2nd}}$ to sample-specific temperature $T(m)$: $h = \text{ReLU}(W_1 m + b_1)$ and $T(m) = \text{softplus}(W_2 h + b_2) + \epsilon$, where the hidden dimension is 16 and $\epsilon = 10^{-1}$ ensures numerical stability. The softplus activation guarantees positive temperatures. This architecture requires only 49 trainable parameters regardless of the number of classes $K$, substantially fewer than existing parametric approaches: vector scaling requires $2K$ parameters, matrix scaling $K^2 + K$, class-dependent temperature scaling (CTS) (Frenkel & Goldberger, 2021) requires $K$, and spline calibration (Gupta et al., 2021) requires $13K$. For ImageNet with $K = 1000$, these methods require thousands of parameters while SMART maintains minimal constant size.

Training minimizes the Charbonnier-SoftECE objective $\mathcal{H}_{\lambda,\delta}$ from Equation (6) on validation mini-batches using Adam optimizer with initial learning rate $5 \times 10^{-3}$. For each sample, we compute its margin, predict temperature via the network, apply temperature scaling to logits, and compute the soft-binned calibration loss with Charbonnier smoothing. At inference, SMART computes the margin for each test sample, predicts its temperature through the trained network, and applies temperature scaling to obtain calibrated predictions. Complete training and inference procedures are detailed in Algorithm 1 (Appendix D).

## 4. Experiments

### 4.1. Experimental Setup

**Datasets** We conduct experiments on several benchmark datasets, including CIFAR-10, CIFAR-100 (Krizhevsky et al., 2009), and ImageNet (Deng et al., 2009). To probe robustness under common corruptions and distribution shifts, we include *ImageNet-C* (all corruption types averaged, severity 5) (Hendrycks & Dietterich, 2019), *ImageNet-LT* (a long-tailed variant with power-law class imbalance) (Liu et al., 2019), and *ImageNet-Sketch* (ImageNet-S; sketch-based OOD variant) (Wang et al., 2019). Implementation details are provided in Appendix J.

**Model Architectures.** To demonstrate the generality of our calibration method, we evaluate across a diverse collection of convolutional and transformer-based networks. For CIFAR-10 and CIFAR-100, we employ ResNet-50 and ResNet-110 (He et al., 2016), Wide-ResNet (Zagoruyko & Komodakis, 2016), and DenseNet-121 (Huang et al., 2017), initialized with pretrained weights (Mukhoti et al., 2020) and trained following the recipe in Appendix J.1. ImageNet and its variants are evaluated on PyTorch's pretrained ResNet-50 and DenseNet-121 (Paszke et al., 2019), the transformer designs Swin-B (Liu et al., 2021), ViT-B/16 and ViT-B/32 (Dosovitskiy et al., 2021), and Wide-ResNet-

*Table 1.* **Comparison of Post-Hoc Calibration Methods in ECE ($\downarrow$, %, 15 bins) Across Various Datasets and Models** (mean across 5 runs). The best-performing method for each dataset-model combination is in bold, and our method is highlighted. Full mean±std results are in Table 7.

| Dataset | Model | Vanilla | TS 2017 | PTS 2022 | CTS 2021 | Spline 2021 | GC 2023 | ProCal 2024 | FC 2025 | SMART ours |
|---|---|---|---|---|---|---|---|---|---|---|
| CIFAR-10 | ResNet-50 | 4.34 | 1.38 | 1.10 | 0.83 | 1.52 | 1.37 | 4.17 | 1.66 | **0.76** |
| | Wide-ResNet | 3.24 | 0.93 | 0.90 | 0.81 | 1.74 | 0.89 | 2.81 | 1.12 | **0.43** |
| CIFAR-100 | ResNet-50 | 17.53 | 5.61 | 1.96 | 3.67 | 3.48 | 5.70 | 9.71 | 2.91 | **1.37** |
| | Wide-ResNet | 15.34 | 4.50 | 1.96 | 3.01 | 3.76 | 4.55 | 9.44 | 4.49 | **1.80** |
| ImageNet-1K | ResNet-50 | 3.65 | 2.17 | 0.95 | 2.17 | 0.62 | 2.44 | 1.08 | 1.71 | **0.52** |
| | DenseNet-121 | 2.53 | 1.85 | 1.02 | 1.86 | 0.81 | 2.20 | 1.52 | 1.35 | **0.57** |
| | Wide-ResNet | 5.43 | 2.89 | 1.14 | 3.27 | 0.66 | 3.66 | 1.57 | 1.62 | **0.52** |
| | Swin-B | 5.05 | 3.91 | 1.05 | 1.53 | 0.88 | 4.95 | 1.00 | 5.05 | **0.46** |
| | ViT-B/16 | 5.62 | 3.60 | 1.23 | 4.65 | 0.91 | 4.39 | 0.97 | 5.65 | **0.48** |
| | ViT-B/32 | 6.39 | 3.93 | 1.27 | 2.12 | 0.81 | 4.67 | 0.88 | 6.39 | **0.71** |
| ImageNet-C | ResNet-50 | 13.82 | 1.97 | 1.12 | 1.69 | 5.61 | 2.69 | 5.79 | 2.51 | **0.62** |
| | DenseNet-121 | 12.57 | 1.58 | 1.19 | 1.44 | 5.18 | 2.01 | 9.88 | 9.44 | **0.63** |
| | Swin-B | 12.03 | 5.82 | 1.53 | 3.05 | 2.58 | 6.92 | 2.53 | 5.18 | **1.23** |
| | ViT-B/16 | 8.28 | 5.24 | 1.27 | 2.76 | 1.71 | 5.95 | 1.96 | 5.37 | **1.06** |
| | ViT-B/32 | 7.69 | 5.10 | 1.07 | 2.97 | 1.43 | 6.40 | 1.55 | 5.50 | **0.96** |
| ImageNet-LT | ResNet-50 | 3.63 | 2.01 | 0.99 | 2.17 | 0.56 | 2.20 | 1.12 | 1.80 | **0.56** |
| | DenseNet-121 | 2.50 | 1.80 | 1.20 | 1.88 | **0.79** | 2.05 | 1.79 | 1.76 | 0.81 |
| | Wide-ResNet | 5.40 | 2.99 | 1.21 | 2.87 | 0.81 | 3.59 | 1.28 | 1.68 | **0.53** |
| | Swin-B | 4.69 | 3.98 | 1.21 | 1.50 | 0.79 | 4.79 | 0.95 | 4.82 | **0.58** |
| | ViT-B/16 | 5.58 | 3.73 | 1.14 | 1.43 | 0.66 | 4.34 | 0.77 | 5.72 | **0.56** |
| | ViT-B/32 | 6.28 | 3.98 | 1.35 | 2.12 | 0.72 | 4.76 | 0.83 | 6.26 | **0.60** |
| ImageNet-S | ResNet-50 | 22.32 | 2.06 | 1.69 | 1.48 | 9.76 | 1.99 | 9.52 | 12.58 | **0.92** |
| | DenseNet-121 | 20.13 | 1.67 | 1.93 | 1.16 | 9.20 | 1.77 | 12.93 | 22.67 | **0.59** |
| | Swin-B | 24.61 | 6.50 | 1.53 | 3.62 | 8.66 | 6.92 | 8.05 | 1.70 | **1.26** |
| | ViT-B/16 | 16.57 | 5.75 | 1.33 | 2.84 | 5.70 | 6.36 | 5.67 | 1.93 | **0.98** |
| | ViT-B/32 | 14.22 | 4.99 | 1.67 | 3.25 | 4.07 | 6.23 | 4.44 | 1.56 | **0.87** |

50. Calibration performance is primarily evaluated using ECE, with additional metrics including AdaECE and top-1 accuracy. Results are averaged over 5 seeds. For Table 1, Appendix E provides the complete mean±std results in Table 7. For the denser paired and ablation tables (Tables 2 and 4), fully reporting standard deviations would substantially expand already wide tables, so the main text reports five-run means and discusses stability qualitatively.

### 4.2. Calibration Performance

We evaluate SMART against leading post-hoc calibration approaches including TS (Guo et al., 2017), PTS (Tomani et al., 2022), CTS (Frenkel & Goldberger, 2021), spline-based calibration (Gupta et al., 2021), Group Calibration (Yang et al., 2023), ProCal (Xiong et al., 2024), and Feature Clipping (FC) (Tao et al., 2025), as well as uncalibrated (Vanilla) models across both standard settings and distribution shift scenarios.

SMART consistently outperforms these methods across CIFAR-10, CIFAR-100, and ImageNet-1K (Table 1), significantly reducing calibration error. The most notable improvement is seen in CIFAR-100, where SMART excels while Spline, despite its strong performance on other datasets, struggles. This highlights SMART's robustness across datasets with varying complexities. CNNs, which often suffer from overconfidence, are generally well-calibrated with TS-based methods. However, transformers see limited calibration improvements from TS-based methods, with SMART outperforming them substantially. On larger datasets like ImageNet-1K, SMART maintains its advantage with consistently lower ECE values. Because SMART applies a positive scalar temperature per sample, it preserves the logit ranking and therefore the top-1 prediction; Table 8 confirms no accuracy drop in practice. SMART works well on both CNNs and ViTs, including settings where GC and FC are less effective.

**Robustness under Class Imbalance and Distribution Shift** Across long-tailed (ImageNet-LT) and corrupted scenarios (ImageNet-S/ImageNet-Sketch, ImageNet-C), SMART's sample-wise temperature adaptation consistently outperforms global and class-wise scalers. Uniform approaches such as TS struggle to accommodate underrepresented classes or severe input degradations, leading to pronounced calibration drift. Spline, FC, and ProCal struggle on ImageNet-S with CNN backbones, whereas SMART remains robust.

**Calibration Performance on AdaECE** We also evaluate SMART using Adaptive Expected Calibration Error (AdaECE) to provide a comprehensive view of its performance, shown in Figure 4, with additional results available in Appendix F. SMART demonstrates strong performance on AdaECE compared to traditional calibration methods across diverse settings. AdaECE addresses limitations of

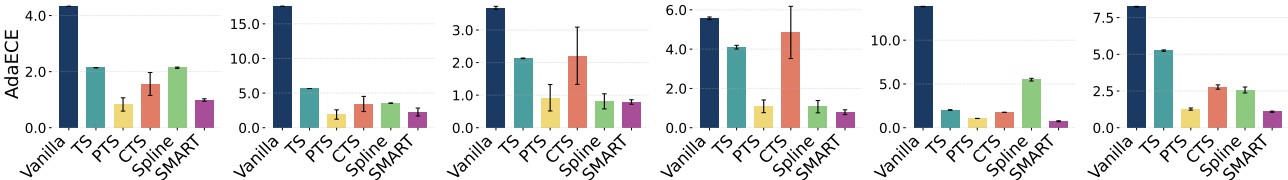

*Figure 4.* **Comparison of calibration methods using AdaECE (↓) across various datasets and models.** From left to right: CIFAR-10/100 (ResNet-50), ImageNet (ResNet-50, ViT-B/16), and ImageNet-C (ResNet-50, ViT-B/16), averaged over 5 runs.

*Table 2.* **Comparison of Training-Time Calibration Methods Using ECE (↓, %, 15 bins) Across Various Datasets and Models.** The best-performing method for each dataset-model combination is in bold, and our method is highlighted. Values are averaged over 5 runs.

| Dataset | Model | NLL | | Brier Loss | | MMCE | | LS-0.05 | | FLSD-53 | | FL-3 | |
|---|---|---|---|---|---|---|---|---|---|---|---|---|---|
| | | base | ours | base | ours | base | ours | base | ours | base | ours | base | ours |
| CIFAR-10 | ResNet-50 | 4.34 | **0.75** | 1.81 | **0.96** | 4.57 | **0.53** | 2.97 | **0.51** | 1.56 | **0.42** | 1.47 | **0.43** |
| | ResNet-110 | 4.41 | **0.44** | 2.56 | **0.60** | 5.07 | **0.38** | 2.09 | **0.28** | 1.87 | **0.45** | 1.54 | **0.54** |
| | DenseNet-121 | 4.51 | **0.53** | 1.52 | **0.31** | 5.10 | **0.66** | 1.89 | **0.51** | 1.23 | **0.62** | 1.31 | **1.02** |
| | Wide-ResNet | 3.24 | **0.30** | 1.25 | **0.38** | 3.30 | **0.34** | 4.25 | **0.36** | 1.58 | **0.39** | 1.68 | **0.54** |
| CIFAR-100 | ResNet-50 | 17.53 | **0.99** | 6.54 | **1.01** | 15.31 | **0.86** | 7.81 | **1.50** | 4.49 | **1.26** | 5.16 | **0.56** |
| | ResNet-110 | 19.06 | **0.98** | 7.87 | **0.87** | 19.13 | **1.42** | 11.03 | **1.01** | 8.54 | **0.85** | 8.65 | **0.73** |
| | DenseNet-121 | 20.99 | **1.86** | 5.22 | **0.59** | 19.10 | **1.34** | 12.87 | **1.02** | 3.70 | **0.91** | 4.14 | **0.98** |
| | Wide-ResNet | 15.34 | **1.38** | 4.35 | **1.00** | 13.17 | **0.98** | 4.88 | **1.24** | 3.02 | **0.79** | 2.14 | **1.12** |

standard ECE by accounting for uneven confidence distributions, providing a more reliable measure of calibration quality. SMART consistently achieves low AdaECE values and low variance across CNNs, ViTs, and datasets (CIFAR and ImageNet variants), demonstrating its robustness to dataset shifts and model architectures. Notably, SMART outperforms more complex methods like Spline calibration and CTS in calibration error and variance while requiring fewer parameters.

By leveraging instance-level temperature through logit margins, SMART yields stable calibration gains across diverse distribution shifts. Its lightweight per-sample inference preserves efficiency while delivering robustness that fixed or ensemble-based temperature schemes do not consistently match. In contrast, Spline degrades sharply on particularly challenging shifts such as ImageNet-S and ImageNet-C, whereas our method consistently sustains low and stable calibration error even under these adverse conditions.

### 4.3. Comparison With Training-Time Calibration Methods

We evaluate SMART alongside training-time calibration techniques in Table 2, including Brier Loss (Brier, 1950), Maximum Mean Calibration Error (MMCE) (Kumar et al., 2018), Label Smoothing (LS-0.05) (Szegedy et al., 2016), and Focal Loss variants (FLSD-53 and FL-3) (Mukhoti et al., 2020). Applying SMART on top of these trained models consistently reduces ECE across models and datasets. In addition, Table 3 compares SMART as a post-hoc method with recent training-time baselines (UWG, DFL, and MDCA); SMART attains lower ECE in all four settings while avoiding classifier retraining (training details in Appendix J.1).

*Table 3.* **Additional training-time baselines.** ECE (↓, %, 15 bins).

| Data/Model | Vanilla | UWG | DFL | MDCA | SMART |
|---|---|---|---|---|---|
| IN/R50 | 3.65 | 1.35 | 1.61 | 3.58 | **0.52** |
| IN/ViT-B/16 | 5.62 | 6.29 | 7.62 | 10.79 | **0.48** |
| Sketch/R50 | 22.32 | 6.00 | 3.30 | 1.06 | **0.92** |
| Sketch/ViT-B/16 | 16.57 | 3.65 | 5.83 | 9.25 | **0.98** |

The gains span CNNs and ViTs under standard and OOD shifts, while the training-time baselines vary more across architectures and datasets. SMART maintains low ECE without retraining the classifier or tuning training-time hyperparameters.

### 4.4. Scalability with Validation Data

SMART demonstrates a strong ability to leverage increasing validation sample sizes compared to competing calibration methods, as shown in Figure 5. While all approaches struggle with minimal validation data, SMART exhibits continuous performance improvement throughout the tested range and ultimately achieves the lowest calibration error. In contrast, the alternative methods display more limited utilization of additional validation samples. TS reaches a performance plateau at moderate validation sizes and fails to improve further, while PTS exhibits concerning instability in the mid-range validation sizes, implicitly reflecting the NLL mismatch. GC shows significant performance spikes that indicate poor robustness to varying validation set sizes. SMART's consistent improvement trajectory indicates that the margin provides a robust signal for more effective temperature estimation as additional validation samples become available. This sample utilization capability makes SMART

*Table 4.* **Different Calibration Objectives.** ECE (↓, %, 15 bins) on ImageNet averaged over 5 runs.

| Architecture | Method | NLL | LS | MSE | Brier | SoftECE | Charbonnier-SoftECE |
|---|---|---|---|---|---|---|---|
| ResNet-50 (Top-1 = 0.761) | TS | 2.04 | 14.33 | 3.69 | 2.31 | 3.16 | 2.12 |
| | PTS | 1.04 | 1.87 | 1.89 | 1.88 | 1.88 | 0.94 |
| | **SMART** | **0.93** | **1.09** | **1.39** | **1.38** | **0.65** | **0.52** |
| ViT-B/16 (Top-1 = 0.810) | TS | 3.73 | 6.05 | 5.58 | 3.11 | 3.10 | 3.08 |
| | PTS | 5.69 | 3.22 | 2.40 | 2.57 | 1.15 | 0.77 |
| | **SMART** | **3.62** | **3.11** | **0.84** | **0.80** | **0.89** | **0.48** |

*Table 5.* **Comparison of alternative off-the-shelf indicators.** ECE (↓, %, 15 bins) on ImageNet averaged over 5 runs.

| Model | Entropy | Conf. | All Logits | Max Logit | Max-Mean | Margin |
|---|---|---|---|---|---|---|
| ResNet-50 | 0.87 | 0.97 | 0.87 | 0.91 | 0.85 | **0.52** |
| DenseNet-121 | 0.62 | 0.89 | 0.79 | 0.80 | 0.84 | **0.57** |
| Wide-ResNet | 1.00 | 1.22 | 0.92 | 0.57 | 0.63 | **0.52** |
| Swin-B | 0.62 | 0.81 | 0.89 | 0.78 | 0.87 | **0.46** |
| ViT-B/16 | 0.90 | 0.75 | 0.97 | 0.91 | 1.20 | **0.48** |

valuable in practical applications where validation data availability may vary.

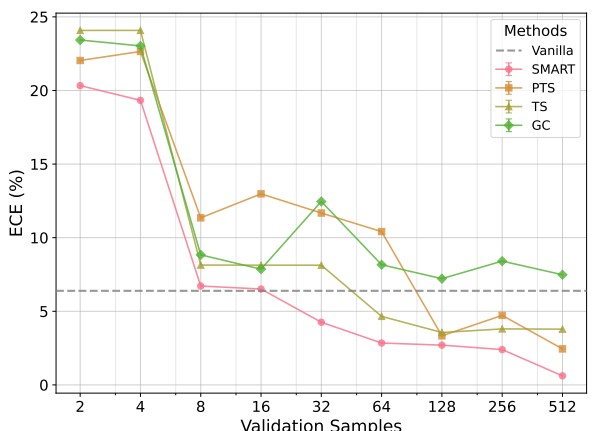

*Figure 5.* **ECE (↓, %, 15 bins) versus validation sample size.** Comparison of calibration methods on ImageNet (ViT-B/32), averaged over five runs.

### 4.5. Ablation Studies

**Choice of Calibration Objective**   We compare how various calibration objectives influence SMART's calibration performance (Table 4). While all tested objectives enable significant improvements over vanilla, they exhibit distinct behavior patterns across architectures. NLL is a useful proper-score diagnostic, and Appendix G reports NLL alongside ECE-style metrics, but proper scores combine reliability with sharpness and can therefore move differently from ECE/SoftECE. MSE and Brier objectives directly penalize squared probability errors, yet their effectiveness fluctuates between CNN and transformer architectures. Charbonnier-SoftECE emerges as the strongest calibration-

centric choice by directly optimizing a smooth reliability objective, achieving both the lowest average error and the smallest variance across diverse model architectures. This supports SMART as a reliability-focused recalibrator rather than a universal probability optimizer.

**Choice of Indicators**   We evaluated six candidate uncertainty signals as inputs to our temperature network on ImageNet-1K (Table 5): predictive entropy, predicted confidence, full logit vectors, maximum logit, mean-normalized logit deviation, and our proposed margin. The margin consistently achieves the lowest calibration error across all tested architectures, outperforming alternative indicators by substantial margins. While full logit vectors contain rich information, they introduce excessive noise that degrades performance in limited-data scenarios. Simpler scalar measures like maximum logit or predicted confidence fail to adequately capture the competitive dynamics between top classes that drive miscalibration. The margin's strong performance stems from its ability to distill prediction uncertainty into a compact, informative representation that directly reflects decision boundary proximity, enabling robust calibration across diverse model architectures.

## 5. Conclusion and Limitations

We introduced SMART, a lightweight recalibration method leveraging the logit margin as a principled calibration indicator for precise temperature adjustment. By capturing sample hardness through this signal, SMART achieves state-of-the-art calibration with minimal parameters and no change to the predicted class. Our Charbonnier-SoftECE objective enables stable calibration-centric optimization as validation data scales, and extensive experiments confirm robustness across architectures, datasets, and challenging distribution shifts.

The scope of this design is also worth clarifying. The top-two margin is not a universal sufficient statistic for calibration: gap-only conditioning is a deliberate bias-variance trade-off that stays robust with limited validation data, but it could be too coarse when ambiguity is dispersed across many non-top classes or in highly specialized domains with extreme class skew. Like other post-hoc methods, SMART can also degrade in zero-shot settings.

## Impact Statement

This paper presents work whose goal is to advance the field of Machine Learning. There are many potential societal consequences of our work, none that we feel must be specifically highlighted here.

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

# A. Theoretical Proofs

## A.1. Temperature–Confidence Relation and Margin Bounds

**Problem and observation.** To reach a target top-class confidence $\hat{p} \in (0, 1)$, how constrained is $T$? Empirically, fixing only $z_M$ leaves $T$ ill-determined; using the *Margin m* yields tight bounds. We now prove this, step by step.

**Target confidence equation.** Requiring $p_{\phi,\max} = \hat{p}$ is equivalent to

$$\frac{e^{z_M/T}}{\sum_{j=1}^{K} e^{z_j/T}} = \hat{p} \qquad \Longleftrightarrow \qquad \sum_{j \neq M} e^{(z_j - z_M)/T} = \frac{1}{\hat{p}} - 1 := S. \tag{9}$$

Because $\hat{p}$ is the *maximum* softmax probability, $\hat{p} \geq 1/K$, hence $S \leq K-1$; moreover $S > 0$ since $\hat{p} < 1$. Thus $S \in (0, K-1]$, and if we assume a strict top-1 margin $m > 0$ (no top-2 ties) then $\hat{p} > 1/K$ and $S \in (0, K-1)$.

**Unboundedness if only $z_M$ is known.** Assume $z_j - z_M = -\delta$ for all $j \neq M$ with $\delta > 0$. Then

$$\sum_{j \neq M} e^{(z_j - z_M)/T} = \sum_{j \neq M} e^{-\delta/T} = (K - 1)\, e^{-\delta/T} = S, \tag{10}$$

$$\Rightarrow \quad T = -\frac{\delta}{\log\left(\frac{S}{K-1}\right)}. \tag{11}$$

When $S < K-1$ (equivalently $\hat{p} > 1/K$), $\log(S/(K-1)) < 0$, so as $\delta \in (0, \infty)$ varies, (11) sweeps $T \in (0, \infty)$. Thus

$$\boxed{\text{Fixing } z_M \text{ alone leaves the feasible } T \text{ unbounded: } (0, \infty).}$$

**$m$-boundedness.** Let $\mu \in \arg\max_{j \neq M} z_j$ denote a runner-up index and define the (nonnegative) margin $m := z_M - z_\mu$. For any $j \notin \{M, \mu\}$,

$$z_j - z_M \leq z_\mu - z_M = -m \quad \Longrightarrow \quad e^{(z_j - z_M)/T} \leq e^{-m/T}.$$

Since $e^{(z_\mu - z_M)/T} = e^{-m/T}$, we have

$$\sum_{j \neq M} e^{(z_j - z_M)/T} = e^{-m/T} + \sum_{j \notin \{M, \mu\}} e^{(z_j - z_M)/T}$$

$$\geq e^{-m/T}, \tag{12}$$

$$\sum_{j \neq M} e^{(z_j - z_M)/T} \leq e^{-m/T} + (K - 2)\, e^{-m/T} = (K - 1)\, e^{-m/T}. \tag{13}$$

Combining (9), (12), and (13) yields

$$e^{-m/T} \leq S \leq (K - 1)\, e^{-m/T}, \qquad S = \tfrac{1}{\hat{p}} - 1 \in (0, K-1]. \tag{14}$$

Equivalently,

$$-\log\left(\frac{S}{K-1}\right) \geq \frac{m}{T} \geq -\log S. \tag{15}$$

Solving (15) for $T > 0$ gives the $m$-bounded feasible set:

$$\boxed{\begin{cases} T \in \left[ \dfrac{m}{-\log(S/(K-1))}, \dfrac{m}{-\log S} \right], & \text{if } 0 < S < 1 \ (\hat{p} > 1/2), \\[2ex] T \in \left[ \dfrac{m}{-\log(S/(K-1))}, \infty \right), & \text{if } 1 \leq S < K - 1 \ (1/K < \hat{p} \leq 1/2), \\[2ex] \text{feasible iff } m = 0, & \text{if } S = K - 1 \ (\hat{p} = 1/K). \end{cases}}$$

*Interpretation.* Knowing the margin $m$ pins down $T$ tightly. When the target confidence is above $1/2$, the feasible $T$ is a *finite* interval whose width shrinks as $m$ grows. For lower target confidences ($\hat{p} \leq 1/2$), one still gets a nontrivial *lower* bound on $T$; a finite upper bound appears exactly when $S < 1$ (i.e., $\hat{p} > 1/2$). In particular, for $K = 2$ the bounds coincide and $T = \frac{m}{-\log S}$ is uniquely determined.

## A.2. Charbonnier-SoftECE Upper-Bounds smCE

**Setup and goal.** Let $p(x) \in [0,1]$ denote the predicted probability of correctness (top-class confidence) and $a(x) := \mathbb{1}\{\hat{y}(x) = y\} \in \{0,1\}$ the correctness indicator. We measure calibration via the *smooth calibration error* (smCE), the worst-case correlation between the residual $a(X) - p(X)$ and any 1-Lipschitz probe of the prediction $p(X)$ (cf. forecasting((Kakade & Foster, 2004)) and the ML calibration view of (Blasiok et al., 2023)):

$$\mathrm{smCE}(f) := \sup_{\varphi \in \mathcal{H}} \left| \mathbb{E}\big[(a(X) - p(X))\, \varphi(p(X))\big] \right|,$$

$$\mathcal{H} := \big\{\varphi : [0,1] \to [-1,1] \text{ s.t. } \mathrm{Lip}(\varphi) \le 1\big\}.$$

We study the *Charbonnier-SoftECE* objective (a smoothed calibration error):

$$\mathcal{H}_{\lambda,\delta}(f) := \mathbb{E}_X\left[\int_0^1 K_\lambda(p(X), u)\, \phi_\delta\big(a(X) - u\big)\, \rho(u)\, du\right], \qquad \phi_\delta(r) := \sqrt{r^2 + \delta^2},$$

where $\rho$ is a reference density on $[0,1]$ and

$$K_\lambda(p, u) = \frac{k_\lambda(p - u)}{\int_0^1 k_\lambda(p - v)\, \rho(v)\, dv}, \qquad k_\lambda(t) := e^{-\lambda t^2}, \ \ \lambda > 0, \tag{16}$$

so that $\int_0^1 K_\lambda(p, u)\, \rho(u)\, du = 1$ for every $p \in [0,1]$. Assume *boundedness and bounded-away-from-zero* of $\rho$: there exist constants $0 < \rho_{\min} \le \rho(u) \le \rho_{\max} < \infty$ for all $u \in [0,1]$, and write $\kappa := \rho_{\max}/\rho_{\min}$.

### Main result.

**Theorem A.1** (Charbonnier-SoftECE upper-bounds smCE)**.** *Under the assumptions above, for all classifiers $f$ and all $\delta \ge 0$,*

$$\mathrm{smCE}(f) \le \mathcal{H}_{\lambda,\delta}(f) + 2\, B_\lambda, \qquad B_\lambda := \sup_{p \in [0,1]} \int_0^1 |p - u|\, K_\lambda(p, u)\, \rho(u)\, du. \tag{17}$$

*Moreover, for the Gaussian kernel $k_\lambda(t) = e^{-\lambda t^2}$,*

$$B_\lambda \le \min\left\{1,\ \frac{2\,\kappa}{\sqrt{\pi}} \cdot \frac{1}{\sqrt{\lambda}\, \mathrm{erf}(\sqrt{\lambda})}\right\}, \tag{18}$$

*and in particular for $\lambda \ge 1$,*

$$B_\lambda \le \frac{C_\kappa}{\sqrt{\lambda}}, \qquad C_\kappa := \frac{2\,\kappa}{\sqrt{\pi}\, \mathrm{erf}(1)} \approx 1.339\,\kappa. \tag{19}$$

**Proof.** For brevity write $p := p(X)$ and $a := a(X)$. Fix any $\varphi \in \mathcal{H}$ with $\|\varphi\|_\infty \le 1$. Introduce the (normalized) kernel smoothing operator

$$(\mathsf{T}_\lambda \varphi)(p) := \int_0^1 K_\lambda(p, u)\, \varphi(u)\, \rho(u)\, du.$$

*Decomposition.*

$$\mathbb{E}[(a - p)\, \varphi(p)] = \mathbb{E}[(a - p)\, (\mathsf{T}_\lambda \varphi)(p)] + \mathbb{E}\big[(a - p)\, \{\varphi(p) - (\mathsf{T}_\lambda \varphi)(p)\}\big]. \tag{20}$$

*Approximation (mollification) error.* By $\mathrm{Lip}(\varphi) \le 1$ and the triangle inequality,

$$\left|\varphi(p) - (\mathsf{T}_\lambda \varphi)(p)\right| = \left|\int_0^1 K_\lambda(p, u)\, \{\varphi(p) - \varphi(u)\}\, \rho(u)\, du\right| \le \int_0^1 K_\lambda(p, u)\, |p - u|\, \rho(u)\, du.$$

Taking the supremum over $p$ and $\varphi$ yields

$$\sup_{\varphi \in \mathcal{H}} \sup_{p \in [0,1]} \left|\varphi(p) - (\mathsf{T}_\lambda \varphi)(p)\right| \le B_\lambda. \tag{21}$$

Hence, using $|a - p| \le 1$,

$$\left| \mathbb{E}\big[(a - p)\{\varphi(p) - (\mathsf{T}_\lambda\varphi)(p)\}\big]\right| \le B_\lambda.$$

*Aligned main term.* By Fubini/Tonelli (bounded integrands) and normalization $\int K_\lambda(p, u)\rho(u)\, du = 1$,

$$\mathbb{E}[(a - p)\,(\mathsf{T}_\lambda\varphi)(p)] = \int_0^1 \varphi(u)\, \mathbb{E}\big[(a - p)\, K_\lambda(p, u)\big]\,\rho(u)\, du,$$

and since $|\varphi(u)| \le 1$,

$$\left| \mathbb{E}[(a - p)\,(\mathsf{T}_\lambda\varphi)(p)] \right| \le \int_0^1 \left| \mathbb{E}\big[(a - p)\, K_\lambda(p, u)\big] \right| \rho(u)\, du. \tag{22}$$

For each $u$, using $|x + y| \le |x| + |y|$ and $|a - p| \le |a - u| + |p - u|$, together with $\phi_\delta(r) \ge |r|$,

$$\begin{aligned}
\left| \mathbb{E}\big[(a - p)\, K_\lambda(p, u)\big] \right| &\le \mathbb{E}\big[\,|a - p|\, K_\lambda(p, u)\big] \\
&\le \mathbb{E}\big[\,\phi_\delta(a - u)\, K_\lambda(p, u)\big] + \mathbb{E}\big[\,|p - u|\, K_\lambda(p, u)\big].
\end{aligned} \tag{23}$$

Integrating (23) against $\rho(u)\, du$ and applying Fubini,

$$\begin{aligned}
\int_0^1 \left| \mathbb{E}\big[(a - p)\, K_\lambda(p, u)\big] \right| \rho(u)\, du &\le \mathbb{E}\left[ \int_0^1 K_\lambda(p, u)\, \phi_\delta(a - u)\, \rho(u)\, du \right] \\
&\quad + \sup_p \int_0^1 |p - u|\, K_\lambda(p, u)\, \rho(u)\, du,
\end{aligned}$$

i.e.,

$$\left| \mathbb{E}[(a - p)\,(\mathsf{T}_\lambda\varphi)(p)] \right| \le \mathcal{H}_{\lambda,\delta}(f) + B_\lambda. \tag{24}$$

*Conclusion.* Combining (21), (24) with (20) and taking the supremum over $\varphi \in \mathcal{H}$ gives $\mathrm{smCE}(f) \le \mathcal{H}_{\lambda,\delta}(f) + 2B_\lambda$.

*Explicit bounds for $B_\lambda$.* By definition,

$$B_\lambda = \sup_{p \in [0,1]} \frac{\int_0^1 |p - u|\, k_\lambda(p - u)\, \rho(u)\, du}{\int_0^1 k_\lambda(p - v)\, \rho(v)\, dv}.$$

Using $\rho(u) \le \rho_{\max}$ in the numerator and $\rho(v) \ge \rho_{\min}$ in the denominator, and changing variables $t = p - u$ or $t = p - v$, we obtain for all $p \in [0, 1]$:

$$B_\lambda \le \kappa \cdot \frac{\int_{\mathbb{R}} |t|\, e^{-\lambda t^2}\, dt}{\int_{p-1}^p e^{-\lambda t^2}\, dt}.$$

Since $p \in [0, 1]$, the denominator integrates over a length-1 interval contained in $[-1, 1]$; by symmetry and unimodality of $t \mapsto e^{-\lambda t^2}$, the minimum over such intervals is attained at an endpoint, e.g. $[0, 1]$. Hence

$$B_\lambda \le \kappa \cdot \frac{\int_{\mathbb{R}} |t|\, e^{-\lambda t^2}\, dt}{\int_0^1 e^{-\lambda t^2}\, dt} = \kappa \cdot \frac{\frac{1}{\lambda}}{\frac{\sqrt{\pi}}{2\sqrt{\lambda}}\, \mathrm{erf}(\sqrt{\lambda})} = \frac{2\kappa}{\sqrt{\pi}} \cdot \frac{1}{\sqrt{\lambda}\, \mathrm{erf}(\sqrt{\lambda})}.$$

Since $|p - u| \le 1$ and $\int K_\lambda \rho = 1$, we also have $B_\lambda \le 1$. For $\lambda \ge 1$, $\mathrm{erf}(\sqrt{\lambda}) \ge \mathrm{erf}(1)$, yielding (19). $\qquad\square$

**Interpretation and guidance.** The guarantee (17) decomposes into a *model-dependent* term $\mathcal{H}_{\lambda,\delta}(f)$ and a *design-only* kernel bias $B_\lambda$, the average *soft-bin radius* around $p$. The Charbonnier envelope obeys $\phi_\delta(r) \ge |r|$, so replacing $|a - u|$ with $\phi_\delta(a - u)$ never weakens control of smCE and yields smooth gradients near $r = 0$. For Gaussian kernels, $B_\lambda = \mathcal{O}(\kappa/\sqrt{\lambda})$ as in (19), so increasing $\lambda$ monotonically tightens the bound; the cap $B_\lambda \le 1$ ensures uniform validity for all $\lambda > 0$.

Section 3.4 optimizes a *bin-mean* Charbonnier-soft-binned objective computed from soft assignments. We formalize a simple additional mismatch term that makes the smCE control rigorous for this implemented surrogate.

Fix bin centers $\{c_b\}_{b=1}^B \subset (0,1)$ and bandwidth $\sigma > 0$, and define soft assignments

$$w_b(p) := \frac{\exp\left(-\frac{(p-c_b)^2}{2\sigma^2}\right)}{\sum_{b'} \exp\left(-\frac{(p-c_{b'})^2}{2\sigma^2}\right)}, \qquad \sum_{b=1}^B w_b(p) = 1.$$

Let $r(X) := a(X) - p(X)$ denote the calibration residual. Define soft bin masses and bin-conditional means

$$S_b := \mathbb{E}[w_b(p(X))], \qquad A_b := \frac{\mathbb{E}[w_b(p(X))\,a(X)]}{S_b}, \qquad C_b := \frac{\mathbb{E}[w_b(p(X))\,p(X)]}{S_b},$$

so that $A_b - C_b = \frac{\mathbb{E}[w_b(p(X))\,r(X)]}{S_b}$ is the (soft) bin-mean residual. Consider the corresponding (population) bin-mean surrogate

$$\overline{\mathcal{H}}_{\sigma,\delta}^{\text{bin}}(f) := \sum_{b=1}^B S_b\,\phi_\delta(C_b - A_b),$$

which differs from the implemented (8) by the constant shift $\phi_\delta(0)$ (since $\sum_b S_b = 1$), hence has the same minimizers.

Define the *bin-aggregation mismatch* (weighted within-bin residual dispersion)

$$\varepsilon_{\text{agg}}(\sigma, B) := \sum_{b=1}^B \mathbb{E}\big[w_b(p(X))\,\big|\,r(X) - (A_b - C_b)\,\big|\big]. \tag{25}$$

Note that $\phi_\delta$ is 1-Lipschitz since $|\phi_\delta'(t)| = \frac{|t|}{\sqrt{t^2+\delta^2}} \leq 1$, so for any random variable $R$, $\mathbb{E}[\phi_\delta(R)] \leq \phi_\delta(\mathbb{E}[R]) + \mathbb{E}[|R - \mathbb{E}[R]|]$.

**Proposition A.2** (smCE control for the bin-mean surrogate). *Let $\overline{\mathcal{H}}_{\sigma,\delta}^{\text{bin}}(f)$ and $\varepsilon_{\text{agg}}(\sigma, B)$ be as above. Then*

$$\mathcal{H}_{\lambda,\delta}(f) \leq \overline{\mathcal{H}}_{\sigma,\delta}^{\text{bin}}(f) + \varepsilon_{\text{agg}}(\sigma, B) + B_\lambda, \tag{26}$$

*and consequently*

$$\text{smCE}(f) \leq \overline{\mathcal{H}}_{\sigma,\delta}^{\text{bin}}(f) + \varepsilon_{\text{agg}}(\sigma, B) + 3B_\lambda. \tag{27}$$

*Equivalently, defining $\varepsilon_{\text{bin}}(\sigma, B) := \varepsilon_{\text{agg}}(\sigma, B) + B_\lambda$ and switching to the shifted implementation $\widetilde{\mathcal{H}}_{\sigma,\delta}^{\text{bin}} = \overline{\mathcal{H}}_{\sigma,\delta}^{\text{bin}} - \phi_\delta(0)$ yields the form stated in Corollary A.4.*

*Proof sketch.* Using the 1-Lipschitz property of $\phi_\delta$, $\phi_\delta(a - u) \leq \phi_\delta(a - p) + |p - u|$. Multiplying by $K_\lambda(p, u)\rho(u)$ and integrating in $u$ gives $\mathcal{H}_{\lambda,\delta}(f) \leq \mathbb{E}[\phi_\delta(a(X) - p(X))] + B_\lambda$. Next, use $\sum_b w_b(p) = 1$ to write $\mathbb{E}[\phi_\delta(r(X))] = \sum_b \mathbb{E}[w_b(p(X))\phi_\delta(r(X))]$. For each bin, let $\mu_b := A_b - C_b = \frac{\mathbb{E}[w_b r]}{S_b}$; by 1-Lipschitzness, $\mathbb{E}[w_b\phi_\delta(r)] \leq S_b\phi_\delta(\mu_b) + \mathbb{E}[w_b|r - \mu_b|]$. Summing over $b$ yields $\mathbb{E}[\phi_\delta(r(X))] \leq \overline{\mathcal{H}}_{\sigma,\delta}^{\text{bin}}(f) + \varepsilon_{\text{agg}}(\sigma, B)$. Combining the two displays gives (26), and then Theorem A.1 yields (27). $\square$

### A.3. Implementation Details of the Bin-Mean Objective

This section provides the complete implementation details for the bin-mean Charbonnier objective introduced in Section 3.4.

**Soft bin assignments.** On a validation set $\{(x_i, y_i)\}_{i=1}^N$, let $p_i := p(x_i) \in [0,1]$ denote top-class confidence and $a_i := \mathbb{I}\{\hat{y}(x_i) = y_i\} \in \{0,1\}$ correctness. Given $B$ uniformly spaced bin centers $\{c_b\}_{b=1}^B$ on $[0,1]$ and Gaussian kernel bandwidth $\sigma$, the soft assignment weights are:

$$w_{i,b} := \frac{\exp\left(-\frac{(p_i-c_b)^2}{2\sigma^2}\right)}{\sum_{b'} \exp\left(-\frac{(p_i-c_{b'})^2}{2\sigma^2}\right)}, \qquad \sum_{b=1}^B w_{i,b} = 1.$$

Each sample contributes to multiple bins proportionally to its proximity to each bin center, creating smooth gradients during optimization.

**Bin-wise statistics.** The soft bin masses and bin-conditional averages are computed as:

$$S_b := \sum_{i=1}^{N} w_{i,b}, \quad \hat{p}_b := \frac{\sum_{i=1}^{N} w_{i,b} p_i}{S_b}, \quad \hat{a}_b := \frac{\sum_{i=1}^{N} w_{i,b} a_i}{S_b}, \quad \pi_b := \frac{S_b}{N}.$$

Here $S_b$ is the effective sample count in bin $b$, $\hat{p}_b$ and $\hat{a}_b$ are the weighted mean confidence and accuracy within the bin, and $\pi_b$ is the normalized bin mass.

**Hyperparameter choices.** We use $B = 15$ soft bins with bandwidth $\sigma = 0.05$ and Charbonnier parameter $\delta = 10^{-3}$. The correspondence between discrete and continuous kernel parameterizations follows from matching exponents: the discrete Gaussian weight $\exp\left(-\frac{(p-c)^2}{2\sigma^2}\right)$ corresponds to the continuous kernel $k_\lambda(t) = e^{-\lambda t^2}$ with $\lambda = \frac{1}{2\sigma^2}$. Thus $\sigma = 0.05$ gives $\lambda = 200$, which by Theorem 3.1 yields kernel approximation error $B_\lambda \leq C_\kappa/\sqrt{200} \approx 0.095\kappa$.

The choice of $B = 15$ bins provides sufficient resolution across the confidence range while maintaining stable bin mass estimates. The bandwidth $\sigma = 0.05$ balances the bias-variance tradeoff: smaller values increase variance from sparse bin assignments, while larger values introduce bias from over-smoothing. The Charbonnier parameter $\delta = 10^{-3}$ provides near-linear behavior for calibration residuals above 0.001 while ensuring smooth gradients near perfect calibration.

See Section J.2 for sensitivity analysis demonstrating robustness across reasonable hyperparameter ranges.

**Theoretical guarantee for the bin-mean surrogate.**

**Assumption A.3** (Within-bin residual homogeneity)**.** The bin-mean surrogate (8) faithfully approximates the ideal objective (6) when the calibration residual $r(x) := a(x) - p(x)$ varies slowly within each soft bin, yielding small bin-aggregation mismatch $\varepsilon_{\mathrm{bin}}(\sigma, B)$ (bounded in Proposition A.2).

**Corollary A.4** (smCE control for the implemented bin-mean objective)**.** *Under the assumptions of Theorem 3.1 and Assumption A.3, the implemented surrogate (8) satisfies*

$$\mathrm{smCE}(f) \leq \widetilde{\mathcal{H}}_{\sigma,\delta}^{\mathrm{bin}}(f) + 2B_\lambda + \varepsilon_{\mathrm{bin}}(\sigma, B) + \phi_\delta(0), \tag{28}$$

*where $\varepsilon_{\mathrm{bin}}(\sigma, B)$ quantifies bin-level aggregation deviation (see Appendix A.2 for formal definition and bounds). The additive constant $\phi_\delta(0) = \delta$ comes from the implementation shift in (8) and does not affect optimization.*

## A.4. Charbonnier-SoftECE vs. NLL

We compare negative log-likelihood (NLL) with Charbonnier-SoftECE within the SMART family $T(x) = h(m(x))$ that scales by the *margin* $m(x) := z_{(1)}(x) - z_{(2)}(x) \in \mathbb{R}_{\geq 0}$. Throughout, assume (i) $T(x) \in [T_{\min}, T_{\max}]$ with $0 < T_{\min} \leq T_{\max} < \infty$, (ii) $\mathbb{E}\|z(X)\|_\infty < \infty$, and (iii) $\mathbb{P}(z_{(1)} = z_{(2)}) = 0$ (no top-2 ties a.s.). Write $t(x) := z(x)/T(x)$, $q(x) := \mathrm{softmax}(t(x))$, $M(x) := \arg\max_k t_k(x)$, $p(x) := q_{M(x)}(x) \in (0, 1)$, and $Y^\top(x) := \mathbf{1}\{Y(x) = M(x)\}$. Define the pointwise top-class probability $r_X(x) := \mathbb{P}(Y = M(x) \mid X = x)$ and the *reliability* curve $r(p) := \mathbb{E}[r_X(X) \mid p(X) = p]$. For any local perturbation below, $h_s$ denotes the perturbed margin-to-temperature map and $M_s(x) := \arg\max_k z_k(x)/h_s(m(x))$ denotes the resulting top index. Since $h_s(m) > 0$ only rescales logits by a positive scalar, $M_s(x) = M(x)$, so the derivatives below keep the top index fixed and write it simply as $M$. We measure calibration by the smooth calibration error (smCE)

$$\mathrm{smCE}(f) := \sup_{\substack{\varphi:[0,1]\to[-1,1] \\ \mathrm{Lip}(\varphi)\leq 1}} \left| \mathbb{E}[(Y^\top - p)\varphi(p)] \right|.$$

**Charbonnier-SoftECE and its smCE control.** Charbonnier-SoftECE is the objective

$$\mathcal{H}_{\lambda,\delta}(f) := \mathbb{E}_X\left[ \int_0^1 K_\lambda(p(X), u)\, \phi_\delta(Y^\top(X) - u)\, \rho(u)\, du \right], \qquad \phi_\delta(r) := \sqrt{r^2 + \delta^2},$$

with normalized kernel $K_\lambda$ as in (16) and a reference density $\rho$ on $[0, 1]$. We *use* (proved in Appendix A.2) the smCE control

$$\mathrm{smCE}(f) \leq \mathcal{H}_{\lambda,\delta}(f) + 2B_\lambda, \qquad B_\lambda = \sup_{p \in [0,1]} \int_0^1 |p - u|\, K_\lambda(p, u)\, \rho(u)\, du, \tag{29}$$

with $B_\lambda = \mathcal{O}(\kappa/\sqrt{\lambda})$ for Gaussian kernels.

**A SMART-feasible local scaling path.** Fix a Borel margin slice $G \subset \mathbb{R}_{\geq 0}$ and $A := \{x : m(x) \in G\}$. For $s > 0$, define the local scaling

$$T_s(x) := \begin{cases} T(x)/s, & x \in A, \\ T(x), & x \notin A, \end{cases}$$

$$t_s(x) := \frac{z(x)}{T_s(x)} = \begin{cases} s\, t(x), & x \in A, \\ t(x), & x \notin A, \end{cases}$$

$$q^{(s)} := \mathrm{softmax}(t_s), \quad p_s := q_{M_s}^{(s)} = q_M^{(s)}.$$

Because uniform multiplication by $s > 0$ preserves coordinate ordering, $M$ is unchanged for all $s > 0$; (iii) rules out measure-zero ties at the boundary.

**Lemma A.5** (Directional derivatives under local margin-dependent scaling). *Let $L_{\mathrm{nll}}(h) := \mathbb{E}[-\log q_Y(X)]$. For any $C^1$ probe $\psi : [0, 1] \to \mathbb{R}$ with $\mathrm{Lip}(\psi) \leq 1$ and $\|\psi\|_\infty \leq 1$, the Gâteaux derivatives at $s = 1$ exist and*

$$\left. \frac{d}{ds} L_{\mathrm{nll}}(h_s) \right|_{s=1} = \mathbb{E}\big[\mathbf{1}_A \left(\langle t\rangle_q - t_Y\right)\big], \tag{30}$$

$$\left. \frac{d}{ds} \mathbb{E}\big[(Y^\top - p_s)\psi(p_s)\big] \right|_{s=1} = \mathbb{E}\big[\mathbf{1}_A\, p\, \left(t_M - \langle t\rangle_q\right) \left(\psi'(p)\, (r_X - p) - \psi(p)\right)\big], \tag{31}$$

*where $\langle t\rangle_q := \sum_k q_k\, t_k$ and $r_X := r_X(X)$.*

*Proof.* On $A$, $\partial_s q_k^{(s)} = q_k^{(s)}(t_k - \langle t\rangle_{q^{(s)}})$, hence $\partial_s(-\log q_Y^{(s)}) = \langle t\rangle_{q^{(s)}} - t_Y$. Outside $A$ the derivative vanishes. Dominated convergence applies since $\big|\partial_s(-\log q_Y^{(s)})\big| \leq 2\|t\|_\infty$ and $\mathbb{E}\|t\|_\infty \leq \mathbb{E}\|z\|_\infty/T_{\min} < \infty$, yielding (30). For $F_\psi(s) := \mathbb{E}[(Y^\top - p_s)\psi(p_s)]$, with $M$ fixed, $\partial_s p_s = \partial_s q_M^{(s)} = q_M^{(s)}(t_M - \langle t\rangle_{q^{(s)}}) = p_s(t_M - \langle t\rangle_{q^{(s)}})$. Thus

$$\partial_s\big((Y^\top - p_s)\psi(p_s)\big) = \big(-\psi(p_s) + (Y^\top - p_s)\psi'(p_s)\big)\partial_s p_s.$$

Conditioning on $X$ replaces $Y^\top$ by $r_X(X)$, whence (31) at $s = 1$ after integration; dominated convergence holds because $p\,|t_M - \langle t\rangle_q| \leq 2\|t\|_\infty$ and $|\psi'| \leq 1$, $|\psi| \leq 1$. $\qquad\square$

**Lemma A.6** (Margin lower bound for the top-logit advantage). *On $\{M = \arg\max t\}$,*

$$t_M - \langle t\rangle_q \geq (1 - p)\left(t_M - t_{(2)}\right) = (1 - p)\frac{m}{T}. \tag{32}$$

*Proof.* $\langle t\rangle_q = p\, t_M + \sum_{j \neq M} q_j t_j \leq p\, t_M + (1 - p)\, t_{(2)}$; rearrange. $\qquad\square$

**A correct NLL directional upper bound (multi-class).** Define the *runner-up gap* $g(x) := t_{(1)}(x) - t_{(2)}(x) = m(x)/T(x) \geq 0$ and the *non-top spread* $\Delta(x) := t_{(2)}(x) - t_{(K)}(x) \geq 0$. For any $x$ with predicted index $M$ and confidence $p = q_M(x)$,

$$\mathbb{E}[\langle t\rangle_q - t_Y \mid X = x] \leq \left(p - r_X(x)\right)g(x) + \left(1 - r_X(x)\right)\Delta(x). \tag{33}$$

In particular, for binary classification ($K = 2$) one has $\Delta \equiv 0$ and (33) reduces to $\mathbb{E}[\langle t\rangle_q - t_Y \mid X] = (p - r_X)\, g$ (exact).

*Derivation of* (33). With $\eta_k(x) := \mathbb{P}(Y = k \mid X = x)$,

$$\mathbb{E}[\langle t\rangle_q - t_Y \mid X = x] = \sum_k (q_k - \eta_k)\, t_k = (p - r_X)\left(t_M - t_{(2)}\right) + \sum_{j \neq M}(q_j - \eta_j)\left(t_j - t_{(2)}\right).$$

Since $t_j - t_{(2)} \leq 0$ and $\sum_{j \neq M}(\eta_j - q_j)_+ \leq \sum_{j \neq M}\eta_j = 1 - r_X$, the last sum is $\leq (1 - r_X)\left(t_{(2)} - t_{(K)}\right) = (1 - r_X)\Delta$. $\qquad\square$

**Consequences and a mild spread control.** On a slice $A = \{m \in G\}$, assume the empirically checkable *spread control*

$$\Delta(x) \leq \Delta_G < \infty \qquad \text{for all } x \in A. \tag{34}$$

Then, combining (33) with Lemma A.5,

$$\frac{d}{ds} L_{\mathrm{nll}}(h_s)\Big|_{s=1} = \mathbb{E}\big[\mathbf{1}_A(\langle t\rangle_q - t_Y)\big] \leq \mathbb{E}\big[\mathbf{1}_A\,(p - r_X)\,g\big] + \Delta_G\,\mu_A, \tag{35}$$

where $\mu_A := \mathbb{P}\{X \in A\}$. In the binary case $\Delta \equiv 0$ and (35) holds with equality.

**Two-slice *mismatch* under mild, empirically observed heterogeneity.** We next give conditions under which a single SMART-feasible local move reduces NLL yet worsens a Lipschitz calibration witness (hence provides an explicit lower-bound increase for smCE).

*Assumptions (empirically checkable).* Fix a *compact* margin slice $G \subset [m_{\min}, m_{\max}]$ and set $A := \{x : m(x) \in G\}$. Let $\gamma_{\min} := \inf_{x \in A} \frac{m(x)}{T(x)}$ and $\gamma_{\max} := \sup_{x \in A} \frac{m(x)}{T(x)}$ (finite and positive by $G$ compact and $T \in [T_{\min}, T_{\max}]$). Assume there exist disjoint compact intervals $J_{\mathrm{U}}, J_{\mathrm{O}} \subset (p_0, 1)$ with gap $\Delta > 0$ and constants $\rho_{\mathrm{U}}, \rho_{\mathrm{O}} > 0$ such that

$$r(p) - p \geq \rho_{\mathrm{U}} \quad \text{for } p \in J_{\mathrm{U}}, \qquad r(p) - p \leq -\rho_{\mathrm{O}} \quad \text{for } p \in J_{\mathrm{O}}.$$

Write $\mu_{\mathrm{U}} := \mathbb{P}\{p \in J_{\mathrm{U}},\ x \in A\}$, $\mu_{\mathrm{O}} := \mathbb{P}\{p \in J_{\mathrm{O}},\ x \in A\}$, $\mu_{\mathrm{gap}} := \mathbb{P}\{x \in A,\ p \notin J_{\mathrm{U}} \cup J_{\mathrm{O}}\}$, and assume additionally:

(a) (*bounded conditional density of $p$ on $A$*) the conditional distribution of $p$ given $X \in A$ has a density $f_{p|A}$ on $(0,1)$ with $\|f_{p|A}\|_\infty \leq D_G < \infty$. In particular, for any interval $I \subset (0,1)$, $\mathbb{P}\{p \in I,\ X \in A\} \leq D_G\,|I|$.

(b) (*slice-bounded advantage*) there exists $C_G < \infty$ with $p(x)\big(t_M(x) - \langle t(x)\rangle_{q(x)}\big) \leq C_G$ for $x \in A$ (e.g., it holds with $C_G := 2\,\mathrm{ess\,sup}_{x \in A} \|t(x)\|_\infty$ whenever $t$ is essentially bounded on $A$).

(c) the spread control (34) holds on $A$ with constant $\Delta_G$.

**Proposition A.7** (Two-slice mismatch: NLL $\downarrow$ but a smCE witness $\uparrow$ (multi-class))**.** *Consider the* sharpening *direction $s \uparrow 1$ applied on the SMART-feasible set $A = \{m \in G\}$. If*

$$\rho_{\mathrm{U}}\,\gamma_{\min}\,\mu_{\mathrm{U}} > \gamma_{\max}\big(\rho_{\mathrm{O}}\,\mu_{\mathrm{O}} + \mu_{\mathrm{gap}}\big) + \Delta_G\,\mu_A, \tag{36}$$

*then $\frac{d}{ds} L_{\mathrm{nll}}(h_s)\big|_{s=1} < 0$ (NLL strictly decreases). Moreover, for any $c \in (0, \min\{1, \Delta\})$ there exists a 1-Lipschitz probe $\psi$ with $\psi \equiv 0$ on $J_{\mathrm{U}}$, $\psi \equiv -c$ on $J_{\mathrm{O}}$, and with transitions confined to a band whose $A$-mass is at most $\varepsilon > 0$, such that*

$$\frac{d}{ds}\mathbb{E}[(Y^\top - p_s)\psi(p_s)]\Big|_{s=1} \geq c\,\underline{p}_{\mathrm{O}}\,(1 - \overline{p}_{\mathrm{O}})\,\gamma_{\min}\,\mu_{\mathrm{O}} - (1 + c)\,C_G\,\varepsilon, \tag{37}$$

*where $\underline{p}_{\mathrm{O}} := \inf J_{\mathrm{O}}$ and $\overline{p}_{\mathrm{O}} := \sup J_{\mathrm{O}}$. Choosing $\varepsilon < \frac{c\,\underline{p}_{\mathrm{O}}\,(1 - \overline{p}_{\mathrm{O}})\,\gamma_{\min}}{(1 + c)\,C_G}\,\mu_{\mathrm{O}}$ makes the right-hand side strictly positive. Because on $J_{\mathrm{O}}$ one has $(r(p) - p)\psi(p) \geq c\,\rho_{\mathrm{O}}$ while $\psi \equiv 0$ on $J_{\mathrm{U}}$, the signed functional at $s = 1$ obeys*

$$\mathbb{E}[(Y^\top - p)\psi(p)] = \mathbb{E}[(r(p) - p)\psi(p)] \geq c\,\rho_{\mathrm{O}}\,\mu_{\mathrm{O}} - c\,\varepsilon > 0, \tag{38}$$

*so a positive derivative implies a strict increase of its absolute value:*

$$\frac{d}{ds}\Big|\mathbb{E}[(Y^\top - p_s)\psi(p_s)]\Big|\Big|_{s=1} > 0.$$

*Consequently, along this SMART-feasible move, NLL strictly decreases while the magnitude of a 1-Lipschitz calibration witness strictly increases. Since $\mathrm{smCE}(f_{h_s}) \geq \big|\mathbb{E}[(Y^\top - p_s)\psi(p_s)]\big|$, this provides an explicit increasing lower bound for smCE (i.e., a concrete direction in which calibration can deteriorate even as NLL improves).*

*Proof.* By (35) and splitting $A$ into the three regions,

$$\frac{d}{ds}L_{\mathrm{nll}}(h_s)\Big|_{s=1} \leq \mathbb{E}\big[\mathbf{1}_{A\cap\{p\in J_{\mathrm{U}}\}}(p-r_X)g\big]$$
$$+ \mathbb{E}\big[\mathbf{1}_{A\cap\{p\in J_{\mathrm{O}}\}}(p-r_X)g\big]$$
$$+ \mathbb{E}\big[\mathbf{1}_{A\cap\{p\notin J_{\mathrm{U}}\cup J_{\mathrm{O}}\}}(p-r_X)g\big]$$
$$+ \Delta_G\,\mu_A.$$

On $A\cap\{p\in J_{\mathrm{U}}\}$, $g\geq\gamma_{\min}$ and $\mathbb{E}[p-r_X\mid p]=p-r(p)\leq-\rho_{\mathrm{U}}$, hence the contribution is $\leq -\rho_{\mathrm{U}}\gamma_{\min}\mu_{\mathrm{U}}$. On $A\cap\{p\in J_{\mathrm{O}}\}$, $g\leq\gamma_{\max}$ and $\mathbb{E}[p-r_X\mid p]\geq\rho_{\mathrm{O}}$, giving at most $\gamma_{\max}\rho_{\mathrm{O}}\mu_{\mathrm{O}}$. On the gap region, $|p-r_X|\leq 1$ and $g\leq\gamma_{\max}$, giving at most $\gamma_{\max}\mu_{\mathrm{gap}}$. This yields strict negativity under (36). For the probe, on $J_{\mathrm{O}}$ we have $\psi'(p)=0$ and $-\psi(p)=c$, so by (31) and Lemma A.6,

$$\frac{d}{ds}\mathbb{E}[(Y^\top-p_s)\psi(p_s)]\Big|_{s=1,\,p\in J_{\mathrm{O}}} \geq c\,\underline{p}_{\mathrm{O}}\,(1-\overline{p}_{\mathrm{O}})\,\gamma_{\min}.$$

On $J_{\mathrm{U}}$ the contribution is $0$ since $\psi\equiv 0$. On the transition band (of $A$-mass $\varepsilon$), $|\psi'|\leq 1$ and $|\psi|\leq c$, hence $|\psi'(p)(r_X-p)-\psi(p)|\leq(1+c)$ while $p(t_M-\langle t\rangle_q)\leq C_G$ on $A$ by (b). Thus the transition contribution is at most $(1+c)\,C_G\,\varepsilon$ in magnitude, giving (37). Finally, (38) holds since $\psi$ depends only on $p$ and $\mathbb{E}[Y^\top-p\mid p]=r(p)-p$. By (a), we can realize the 1-Lipschitz $\psi$ with linear ramps of total width at most $2c$, whence $\varepsilon\leq 2D_Gc$; shrinking $c$ if needed makes the stated choice of $\varepsilon$ feasible. $\square$

**Lemma A.8** (Small-$s$ realization for the witness mismatch). *Under Proposition A.7, there exists $s^\uparrow>1$ arbitrarily close to 1 such that*
$$L_{\mathrm{nll}}(h_{s^\uparrow}) < L_{\mathrm{nll}}(h) \qquad \text{and} \qquad \big|\mathbb{E}[(Y^\top-p_{s^\uparrow})\psi(p_{s^\uparrow})]\big| > \big|\mathbb{E}[(Y^\top-p)\psi(p)]\big|,$$
*where $\psi$ is the 1-Lipschitz probe constructed in Proposition A.7.*

*Proof.* $L_{\mathrm{nll}}(h_s)$ is $C^1$ at $s=1$ by Lemma A.5, with strictly negative derivative; hence $L_{\mathrm{nll}}(h_{s^\uparrow})<L_{\mathrm{nll}}(h)$ for all $s^\uparrow>1$ sufficiently close to 1. For the witness, fix the $\psi$ from Proposition A.7; then $F_\psi(s):=\mathbb{E}[(Y^\top-p_s)\psi(p_s)]$ is $C^1$ with $F_\psi(1)>0$ and $F'_\psi(1)>0$, so $|F_\psi(s^\uparrow)|>|F_\psi(1)|$ for all $s^\uparrow>1$ close enough to 1. $\square$

**Takeaway.** Along SMART-feasible local scalings of the temperature map $T(x)=h(m(x))$, Charbonnier-SoftECE continues to control smCE via (29), whereas NLL can be *locally improved* (decreased) while the magnitude of a 1-Lipschitz calibration witness *increases* under mild, empirically checkable heterogeneity of confidence slices (Proposition A.7). Since smCE is the supremum over such witnesses, this shows that NLL improvement does not guarantee calibration improvement and can directly conflict with calibration objectives.

## B. The Use of Large Language Models

During the preparation of this work, we utilized a Large Language Model (LLM) to assist with editorial refinement of the manuscript. The model's application was limited exclusively to improving textual quality and presentation, not for generating substantive research content. The LLM's contributions included:

- Enhancing sentence structure and paragraph organization to improve clarity, brevity, and scholarly tone.

- Identifying and correcting errors in grammar, spelling, and punctuation.

- Strengthening coherence and smoothing transitions throughout the text.

## C. Runtime Efficiency

To verify the runtime efficiency of our method, we compare its runtime with baseline methods. The results are reported in Table 6. TS optimizes a single scalar temperature via a few gradient steps or closed-form updates, then applies this same factor to every logit, resulting in negligible overhead (2.42 s). SMART yields a small per-sample inference cost and

hence a modest total runtime (23.03 s). PTS feeds logits into a small neural network for each sample to predict a bespoke temperature, incurring a larger computational cost than SMART. CTS is the most expensive at more than 1 hour with the highest variance, as it conducts an exhaustive grid search for 5 epochs over a dense temperature grid (e.g., 0.1–10) for each of the 1,000 classes, leading to $O(C \times G \times N)$ evaluations (classes $\times$ grid points $\times$ samples). The spline-based calibrator precomputes a monotonic mapping on the validation set and then applies a fast piecewise-linear transform at test time, yielding intermediate overhead. These differences illustrate the trade-off between expressive power and efficiency: TS is almost instantaneous, SMART adds only a small network-forward cost per sample, PTS trades per-sample flexibility for moderate cost, and CTS's brute-force search becomes prohibitive at scale.

*Table 6.* **Average Runtime (s) on ImageNet** over 10 runs on a ResNet-50 model.

| Method | TS | Spline | PTS | CTS | SMART |
|---|---|---|---|---|---|
| Runtime (s) | $2.42 \pm 0.1$ | $28.51 \pm 0.9$ | $1050.44 \pm 37.8$ | $5457.55 \pm 125.5$ | $23.03 \pm 0.41$ |
| Runtime / SMART | $0.11\times$ | $1.24\times$ | $45.6\times$ | $237.0\times$ | $1.00\times$ |

## D. The Proposed SMART Framework

This section presents the detailed algorithmic implementation of SMART, providing a step-by-step procedure for applying margin-based temperature scaling with soft-binned ECE optimization.

---

**Algorithm 1** SMART: Sample Margin-Aware Recalibration of Temperature

---

1: **Input:** Validation logits and labels $\{\mathbf{z}_i, y_i\}_{i=1}^{N_{\text{val}}}$, temperature network $h_\phi(\cdot)$, mini-batch size $M$
2: Compute margins: $m_i = z_{i,\text{max}} - z_{i,\text{2nd}}$ for each $i \in \{1, \ldots, N_{\text{val}}\}$
3: **for** epoch $= 1, \ldots, N_{\text{epochs}}$ **do**
4:     Shuffle validation indices and split them into mini-batches $\{\mathcal{B}_t\}$ of size $M$
5:     **for** each mini-batch $\mathcal{B}_t$ **do**
6:         Predict temperatures: $T_i = h_\phi(m_i)$ for each $i \in \mathcal{B}_t$
7:         Scale logits: $\tilde{\mathbf{z}}_i = \mathbf{z}_i / T_i$ for each $i \in \mathcal{B}_t$
8:         Compute mini-batch loss: $\mathcal{L}_{\mathcal{B}_t} = \text{CharbonnierSoftECE}(\{(\tilde{\mathbf{z}}_i, y_i)\}_{i \in \mathcal{B}_t})$ (Equation 8)
9:         Update $\phi$ with Adam using $\nabla_\phi \mathcal{L}_{\mathcal{B}_t}$
10:     **end for**
11: **end for**
12: **Return:** Trained temperature network $h_\phi$

---

## E. Full Calibration Performance

Full calibration performance for Table 1 is reported in Table 7.

*Table 7.* **Comparison of Post-Hoc Calibration Methods Using ECE (↓, %, 15 bins) Across Various Datasets and Models** (mean ± std across 5 seeds, where applicable). The best-performing method for each dataset-model combination is in bold, and our method is highlighted.

| Dataset | Model | Vanilla | TS | PTS | CTS | Spline | GC | ProCal | FC | SMART (ours) |
|---|---|---|---|---|---|---|---|---|---|---|
| CIFAR-10 | ResNet-50 | 4.34 | 1.38 ± 0.26 | 1.10 ± 0.21 | 0.83 ± 0.15 | 1.52 ± 0.03 | 1.37 ± 0.08 | 4.17 ± 0.12 | 1.66 ± 0.09 | **0.76 ± 0.02** |
| | Wide-ResNet | 3.24 | 0.93 ± 0.20 | 0.90 ± 0.19 | 0.81 ± 0.17 | 1.74 ± 0.01 | 0.89 ± 0.06 | 2.81 ± 0.11 | 1.12 ± 0.07 | **0.43 ± 0.05** |
| CIFAR-100 | ResNet-50 | 17.53 | 5.61 ± 1.39 | 1.96 ± 0.48 | 3.67 ± 0.88 | 3.48 ± 0.00 | 5.70 ± 0.15 | 9.71 ± 0.18 | 2.91 ± 0.12 | **1.37 ± 0.27** |
| | Wide-ResNet | 15.34 | 4.50 ± 0.62 | 1.96 ± 0.27 | 3.01 ± 0.42 | 3.76 ± 0.00 | 4.55 ± 0.13 | 9.44 ± 0.16 | 4.49 ± 0.14 | **1.80 ± 0.10** |
| ImageNet-1K | ResNet-50 | 3.65 | 2.17 ± 0.03 | 0.95 ± 0.36 | 2.17 ± 0.78 | 0.62 ± 0.18 | 2.44 ± 0.12 | 1.08 ± 0.14 | 1.71 ± 0.08 | **0.52 ± 0.12** |
| | DenseNet-121 | 2.53 | 1.85 ± 0.04 | 1.02 ± 0.46 | 1.86 ± 0.81 | 0.81 ± 0.35 | 2.20 ± 0.25 | 1.52 ± 0.21 | 1.35 ± 0.29 | **0.57 ± 0.03** |
| | Wide-ResNet | 5.43 | 2.89 ± 0.11 | 1.14 ± 0.24 | 3.27 ± 0.69 | 0.66 ± 0.10 | 3.66 ± 0.16 | 1.57 ± 0.10 | 1.62 ± 0.09 | **0.52 ± 0.07** |
| | Swin-B | 5.05 | 3.91 ± 0.07 | 1.05 ± 0.05 | 1.53 ± 0.08 | 0.88 ± 0.14 | 4.95 ± 0.17 | 1.00 ± 0.15 | 5.05 ± 0.06 | **0.46 ± 0.03** |
| | ViT-B/16 | 5.62 | 3.60 ± 0.19 | 1.23 ± 0.29 | 4.65 ± 1.02 | 0.91 ± 0.31 | 4.39 ± 0.25 | 0.97 ± 0.30 | 5.65 ± 0.06 | **0.48 ± 0.13** |
| | ViT-B/32 | 6.39 | 3.93 ± 0.02 | 1.27 ± 0.97 | 2.12 ± 1.59 | 0.81 ± 0.12 | 4.67 ± 0.13 | 0.88 ± 0.32 | 6.39 ± 0.06 | **0.71 ± 0.18** |
| ImageNet-C | ResNet-50 | 13.82 | 1.97 ± 0.02 | 1.12 ± 0.13 | 1.69 ± 0.20 | 5.61 ± 0.15 | 2.69 ± 0.11 | 5.79 ± 0.19 | 2.51 ± 0.13 | **0.62 ± 0.03** |
| | DenseNet-121 | 12.57 | 1.58 ± 0.00 | 1.19 ± 0.15 | 1.44 ± 0.19 | 5.18 ± 0.13 | 2.01 ± 0.09 | 9.88 ± 0.24 | 9.44 ± 0.31 | **0.63 ± 0.01** |
| | Swin-B | 12.03 | 5.82 ± 0.05 | 1.53 ± 0.00 | 3.05 ± 0.01 | 2.58 ± 0.21 | 6.92 ± 0.18 | 2.53 ± 0.12 | 5.18 ± 0.17 | **1.23 ± 0.04** |
| | ViT-B/16 | 8.28 | 5.24 ± 0.01 | 1.27 ± 0.05 | 2.76 ± 0.10 | 1.71 ± 0.22 | 5.95 ± 0.15 | 1.96 ± 0.14 | 5.37 ± 0.20 | **1.06 ± 0.02** |
| | ViT-B/32 | 7.69 | 5.10 ± 0.00 | 1.07 ± 0.08 | 2.97 ± 0.24 | 1.43 ± 0.24 | 6.40 ± 0.16 | 1.55 ± 0.11 | 5.50 ± 0.18 | **0.96 ± 0.01** |
| ImageNet-LT | ResNet-50 | 3.63 | 2.01 ± 0.02 | 0.99 ± 0.32 | 2.17 ± 0.68 | 0.56 ± 0.10 | 2.20 ± 0.17 | 1.12 ± 0.20 | 1.80 ± 0.23 | **0.56 ± 0.04** |
| | DenseNet-121 | 2.50 | 1.80 ± 0.06 | 1.20 ± 0.26 | 1.88 ± 0.41 | **0.79 ± 0.07** | 2.05 ± 0.11 | 1.79 ± 0.09 | 1.76 ± 0.50 | 0.81 ± 0.01 |
| | Wide-ResNet | 5.40 | 2.99 ± 0.05 | 1.21 ± 0.77 | 2.87 ± 1.79 | 0.81 ± 0.24 | 3.59 ± 0.18 | 1.28 ± 0.06 | 1.68 ± 0.10 | **0.53 ± 0.02** |
| | Swin-B | 4.69 | 3.98 ± 0.12 | 1.21 ± 0.45 | 1.50 ± 0.56 | 0.79 ± 0.17 | 4.79 ± 0.27 | 0.95 ± 0.16 | 4.82 ± 0.10 | **0.58 ± 0.01** |
| | ViT-B/16 | 5.58 | 3.73 ± 0.13 | 1.14 ± 0.47 | 1.43 ± 0.58 | 0.66 ± 0.05 | 4.34 ± 0.14 | 0.77 ± 0.14 | 5.72 ± 0.08 | **0.56 ± 0.14** |
| | ViT-B/32 | 6.28 | 3.98 ± 0.06 | 1.35 ± 0.41 | 2.12 ± 0.63 | 0.72 ± 0.23 | 4.76 ± 0.08 | 0.83 ± 0.12 | 6.26 ± 0.03 | **0.60 ± 0.11** |
| ImageNet-S | ResNet-50 | 22.32 | 2.06 ± 0.06 | 1.69 ± 0.27 | 1.48 ± 0.23 | 9.76 ± 0.22 | 1.99 ± 0.16 | 9.52 ± 0.31 | 12.58 ± 1.35 | **0.92 ± 0.09** |
| | DenseNet-121 | 20.13 | 1.67 ± 0.28 | 1.93 ± 0.19 | 1.16 ± 0.11 | 9.20 ± 0.32 | 1.77 ± 0.15 | 12.93 ± 0.23 | 22.67 ± 1.07 | **0.59 ± 0.25** |
| | Swin-B | 24.61 | 6.50 ± 0.05 | 1.53 ± 0.19 | 3.62 ± 0.45 | 8.66 ± 0.15 | 6.92 ± 0.35 | 8.05 ± 0.30 | 1.70 ± 0.06 | **1.26 ± 0.05** |
| | ViT-B/16 | 16.57 | 5.75 ± 0.08 | 1.33 ± 0.21 | 2.84 ± 0.43 | 5.70 ± 0.19 | 6.36 ± 0.29 | 5.67 ± 0.38 | 1.93 ± 0.18 | **0.98 ± 0.08** |
| | ViT-B/32 | 14.22 | 4.99 ± 0.15 | 1.67 ± 0.27 | 3.25 ± 0.50 | 4.07 ± 0.21 | 6.23 ± 0.16 | 4.44 ± 0.23 | 1.56 ± 0.09 | **0.87 ± 0.18** |

# F. Calibration Performance on Other Metrics

## F.1. Accuracy Performance

*Table 8.* **Comparison of Classification Accuracy (%) Across Calibration Methods** (averaged over seeds 1–5).

| Dataset | Model | Vanilla | TS | PTS | CTS | Spline | SMART |
|---|---|---|---|---|---|---|---|
| CIFAR-10 | ResNet-50 | 95.05% | 95.05% | 95.05% | 94.88% | 95.05% | 95.05% |
| | Wide-ResNet | 96.13% | 96.13% | 96.13% | 96.09% | 96.13% | 96.13% |
| CIFAR-100 | ResNet-50 | 76.69% | 76.69% | 76.69% | 76.38% | 76.69% | 76.69% |
| | Wide-ResNet | 79.29% | 79.29% | 79.29% | 79.28% | 79.29% | 79.29% |
| ImageNet-1K | ResNet-50 | 76.16% | 76.16% | 76.16% | 75.32% | 76.17% | 76.16% |
| | DenseNet-121 | 74.44% | 74.44% | 74.44% | 73.71% | 74.43% | 74.44% |
| | Wide-ResNet | 78.46% | 78.46% | 78.46% | 77.70% | 78.46% | 78.46% |
| | Swin-B | 83.17% | 83.17% | 83.17% | 82.80% | 83.17% | 83.17% |
| | ViT-B/16 | 81.12% | 81.12% | 81.12% | 79.64% | 80.86% | 81.12% |
| | ViT-B/32 | 75.95% | 75.95% | 75.95% | 75.14% | 75.94% | 75.95% |
| ImageNet-C | ResNet-50 | 19.16% | 19.16% | 19.16% | 19.34% | 19.16% | 19.16% |
| | DenseNet-121 | 21.25% | 21.25% | 21.25% | 21.36% | 21.25% | 21.25% |
| | Swin-B | 40.83% | 40.83% | 40.83% | 41.22% | 40.83% | 40.83% |
| | ViT-B/16 | 41.07% | 41.07% | 41.07% | 41.28% | 41.07% | 41.07% |
| | ViT-B/32 | 37.82% | 37.82% | 24.56% | 37.96% | 37.85% | 37.82% |
| ImageNet-LT | ResNet-50 | 76.04% | 76.04% | 76.04% | 75.43% | 76.04% | 76.04% |
| | DenseNet-121 | 74.34% | 74.34% | 74.34% | 73.88% | 74.40% | 74.34% |
| | Wide-ResNet | 78.39% | 78.39% | 78.39% | 77.67% | 78.40% | 78.39% |
| | Swin-B | 82.95% | 82.95% | 82.95% | 82.55% | 82.94% | 82.95% |
| | ViT-B/16 | 80.95% | 80.95% | 80.95% | 80.58% | 81.00% | 80.95% |
| | ViT-B/32 | 75.89% | 75.89% | 75.89% | 75.14% | 75.92% | 75.89% |
| ImageNet-S | ResNet-50 | 24.09% | 24.09% | 24.09% | 23.88% | 24.09% | 24.09% |
| | DenseNet-121 | 24.30% | 24.30% | 24.30% | 23.87% | 31.55% | 24.30% |
| | Swin-B | 31.54% | 31.54% | 31.54% | 31.65% | 31.55% | 31.54% |
| | ViT-B/16 | 29.37% | 29.37% | 29.37% | 29.51% | 29.39% | 29.37% |
| | ViT-B/32 | 27.77% | 27.77% | 27.77% | 27.76% | 27.75% | 27.77% |

**Accuracy Preservation Analysis** Table 8 confirms that SMART achieves strong calibration while preserving classification accuracy, an important advantage of post-hoc methods. Unlike CTS, which suffers accuracy drops up to 1.48% due to class-specific boundary alterations, or Spline's variable impacts on transformers, SMART's design keeps the top-1 prediction unchanged. By operating through positive temperature scaling, SMART adjusts confidence without disturbing the relative logit ordering that determines predictions. This preservation holds even under severe distribution shifts like ImageNet-C and ImageNet-Sketch, where SMART maintains base model accuracy while improving calibration. This property makes SMART suitable for applications requiring both correct predictions and reliable uncertainty estimates.

## F.2. AdaECE Performance

This section provides an in-depth analysis of calibration performance using AdaECE across different datasets and model architectures, complementing the results presented in Section 4.2. Adaptive-ECE is a measure of calibration performance that addresses the bias of equal-width binning scheme of ECE. It adapts the bin-size to the number of samples and ensures that each bin is evenly distributed with samples. The formula for Adaptive-ECE is as follows:

$$\text{Adaptive-ECE} = \sum_{i=1}^{\mathbb{B}} \frac{|B_i|}{N}|I_i - C_i| \text{ s.t. } \forall i, j \cdot |B_i| = |B_j| \tag{39}$$

AdaECE offers a more rigorous assessment of calibration quality than standard ECE by adapting bin boundaries to ensure uniform sample distribution, preventing calibration errors from being masked in sparsely populated confidence regions. Table 9 presents comprehensive AdaECE results across all evaluated datasets and architectures. SMART consistently outperforms competing methods under this metric, achieving the lowest AdaECE on 24 of 26 dataset-architecture combinations.

**CIFAR Performance Analysis.** SMART demonstrates strong calibration on CIFAR datasets in Figure 6, achieving the lowest AdaECE with notably stable variance compared to competitors. The key insight emerges when comparing CIFAR-10 to CIFAR-100: while global methods like TS degrade as class count increases, SMART maintains robust performance. PTS shows competitive results but with substantially higher variance, indicating reliability issues. Spline

*Table 9.* **Comparison of AdaECE Calibration Methods Using AdaECE (↓, %, 15 bins) Across Various Datasets and Models** (averaged over 5 seeds).

| Dataset | Model | Vanilla | TS | PTS | CTS | Spline | SMART |
|---------|-------|---------|-----|-----|-----|--------|-------|
| CIFAR-10 | ResNet-50 | $4.33 \pm 0.0\%$ | $2.14 \pm 0.0\%$ | $0.83 \pm 28.6\%$ | $1.56 \pm 26.2\%$ | $2.14 \pm 1.1\%$ | $\mathbf{0.99 \pm 4.3\%}$ |
| | Wide-ResNet | $3.24 \pm 0.0\%$ | $1.71 \pm 0.0\%$ | $0.89 \pm 21.9\%$ | $1.47 \pm 19.7\%$ | $2.30 \pm 0.4\%$ | $\mathbf{0.50 \pm 12.2\%}$ |
| CIFAR-100 | ResNet-50 | $17.53 \pm 0.0\%$ | $5.66 \pm 0.0\%$ | $1.91 \pm 35.3\%$ | $3.43 \pm 32.0\%$ | $3.55 \pm 0.0\%$ | $\mathbf{2.27 \pm 25.2\%}$ |
| | Wide-ResNet | $15.34 \pm 0.0\%$ | $4.41 \pm 0.0\%$ | $1.69 \pm 13.0\%$ | $2.95 \pm 11.6\%$ | $3.95 \pm 0.1\%$ | $\mathbf{1.83 \pm 2.1\%}$ |
| ImageNet-1K | ResNet-50 | $3.68 \pm 1.3\%$ | $2.13 \pm 0.5\%$ | $0.92 \pm 44.1\%$ | $2.21 \pm 39.8\%$ | $0.81 \pm 28.7\%$ | $\mathbf{0.79 \pm 8.7\%}$ |
| | DenseNet-121 | $2.52 \pm 1.4\%$ | $1.74 \pm 1.8\%$ | $1.05 \pm 41.3\%$ | $1.78 \pm 38.0\%$ | $0.77 \pm 28.0\%$ | $\mathbf{0.65 \pm 10.2\%}$ |
| | Wide-ResNet | $5.31 \pm 0.3\%$ | $2.87 \pm 2.8\%$ | $1.04 \pm 20.6\%$ | $3.24 \pm 18.0\%$ | $\mathbf{0.83 \pm 36.3\%}$ | $0.87 \pm 14.3\%$ |
| | Swin-B | $4.86 \pm 0.6\%$ | $4.50 \pm 1.0\%$ | $1.05 \pm 4.6\%$ | $1.59 \pm 5.1\%$ | $1.04 \pm 5.3\%$ | $\mathbf{0.74 \pm 12.2\%}$ |
| | ViT-B/16 | $5.57 \pm 1.2\%$ | $4.10 \pm 2.3\%$ | $1.09 \pm 29.7\%$ | $4.85 \pm 27.4\%$ | $1.07 \pm 29.2\%$ | $\mathbf{0.79 \pm 15.4\%}$ |
| | ViT-B/32 | $6.41 \pm 0.4\%$ | $3.92 \pm 1.7\%$ | $1.27 \pm 71.9\%$ | $1.90 \pm 66.4\%$ | $0.96 \pm 15.3\%$ | $\mathbf{0.78 \pm 3.6\%}$ |
| ImageNet-C | ResNet-50 | $13.84 \pm 0.2\%$ | $2.02 \pm 1.7\%$ | $1.06 \pm 0.7\%$ | $1.76 \pm 0.6\%$ | $5.49 \pm 2.8\%$ | $\mathbf{0.74 \pm 8.0\%}$ |
| | DenseNet-121 | $12.57 \pm 0.1\%$ | $1.64 \pm 0.7\%$ | $1.17 \pm 9.9\%$ | $1.48 \pm 8.2\%$ | $2.57 \pm 7.9\%$ | $\mathbf{0.70 \pm 3.6\%}$ |
| | Swin-B | $11.98 \pm 0.1\%$ | $5.83 \pm 1.0\%$ | $1.58 \pm 0.0\%$ | $3.07 \pm 0.2\%$ | $5.13 \pm 2.3\%$ | $\mathbf{1.31 \pm 2.9\%}$ |
| | ViT-B/16 | $8.24 \pm 0.3\%$ | $5.25 \pm 0.9\%$ | $1.27 \pm 5.9\%$ | $2.77 \pm 5.3\%$ | $2.57 \pm 7.9\%$ | $\mathbf{1.09 \pm 4.0\%}$ |
| | ViT-B/32 | $7.66 \pm 0.2\%$ | $5.11 \pm 0.0\%$ | $1.07 \pm 4.3\%$ | $2.97 \pm 3.7\%$ | $1.45 \pm 16.8\%$ | $\mathbf{1.01 \pm 4.2\%}$ |
| ImageNet-LT | ResNet-50 | $3.54 \pm 0.9\%$ | $2.02 \pm 1.2\%$ | $0.92 \pm 35.5\%$ | $2.17 \pm 33.0\%$ | $0.71 \pm 20.7\%$ | $\mathbf{0.67 \pm 3.3\%}$ |
| | DenseNet-121 | $2.37 \pm 3.4\%$ | $1.74 \pm 2.1\%$ | $1.17 \pm 23.6\%$ | $1.86 \pm 21.3\%$ | $\mathbf{0.73 \pm 26.4\%}$ | $0.76 \pm 0.7\%$ |
| | Wide-ResNet | $5.22 \pm 0.4\%$ | $2.98 \pm 0.9\%$ | $1.22 \pm 62.4\%$ | $2.83 \pm 58.1\%$ | $0.79 \pm 18.1\%$ | $\mathbf{0.98 \pm 4.4\%}$ |
| | Swin-B | $4.69 \pm 0.6\%$ | $4.48 \pm 1.2\%$ | $1.43 \pm 19.1\%$ | $1.23 \pm 18.0\%$ | $0.95 \pm 6.7\%$ | $\mathbf{0.74 \pm 31.3\%}$ |
| | ViT-B/16 | $5.57 \pm 0.8\%$ | $4.18 \pm 2.9\%$ | $1.13 \pm 43.4\%$ | $1.06 \pm 40.1\%$ | $0.95 \pm 12.9\%$ | $\mathbf{0.85 \pm 15.1\%}$ |
| | ViT-B/32 | $6.26 \pm 0.6\%$ | $3.97 \pm 1.6\%$ | $1.30 \pm 31.1\%$ | $2.04 \pm 28.2\%$ | $0.86 \pm 26.5\%$ | $\mathbf{0.84 \pm 10.1\%}$ |
| ImageNet-S | ResNet-50 | $22.31 \pm 0.3\%$ | $2.01 \pm 2.9\%$ | $1.64 \pm 16.4\%$ | $1.51 \pm 14.7\%$ | $9.51 \pm 2.4\%$ | $\mathbf{0.90 \pm 15.8\%}$ |
| | DenseNet-121 | $20.15 \pm 0.5\%$ | $1.67 \pm 17.0\%$ | $1.93 \pm 9.6\%$ | $1.16 \pm 8.3\%$ | $8.7 \pm 1.92\%$ | $\mathbf{0.76 \pm 32.3\%}$ |
| | Swin-B | $24.62 \pm 0.0\%$ | $6.40 \pm 0.5\%$ | $1.53 \pm 12.2\%$ | $3.57 \pm 11.1\%$ | $9.06 \pm 4.2\%$ | $\mathbf{1.53 \pm 3.8\%}$ |
| | ViT-B/16 | $16.57 \pm 0.2\%$ | $5.62 \pm 0.7\%$ | $1.33 \pm 8.7\%$ | $2.98 \pm 7.3\%$ | $8.66 \pm 1.9\%$ | $\mathbf{1.08 \pm 4.3\%}$ |
| | ViT-B/32 | $14.19 \pm 0.3\%$ | $4.98 \pm 2.9\%$ | $1.66 \pm 16.0\%$ | $3.23 \pm 14.1\%$ | $5.64 \pm 3.3\%$ | $\mathbf{1.07 \pm 19.9\%}$ |

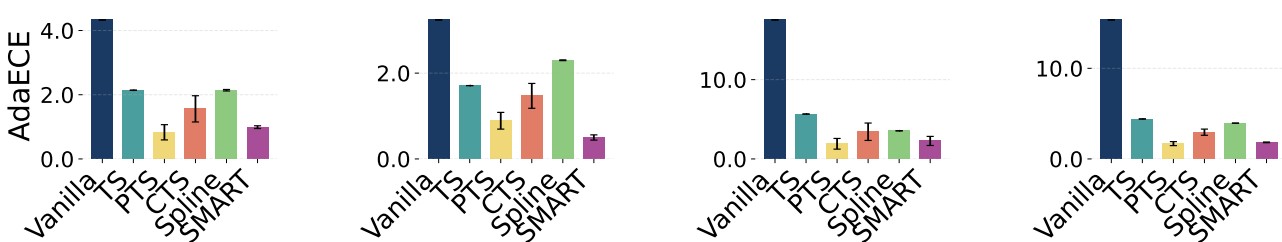

*Figure 6.* **AdaECE comparison on CIFAR datasets.** SMART consistently achieves strong calibration on both CIFAR-10 and CIFAR-100 across multiple architectures. From left to right are CIFAR-10 ResNet-50/Wide-ResNet and CIFAR-100 ResNet-50/Wide-ResNet.

struggles particularly with CIFAR-100's complex confidence landscape, revealing how non-parametric methods become less effective as classification complexity increases.

**Large-Scale Classification on ImageNet.** In Figure 7, the ImageNet results reveal a crucial architectural insight: SMART maintains consistent performance across both CNN and transformer designs, while traditional methods like TS and CTS show pronounced degradation on transformers. This architectural robustness highlights SMART's ability to capture uncertainty signals through the margin regardless of model inductive biases. PTS exhibits high variance, confirming that high-dimensional parameterizations struggle with reliability when learning complex temperature mappings, particularly on large-scale datasets.

**Robustness to Input Corruption.** As shown in Figure 8, SMART's resilience under corruption provides evidence for the stability of decision boundary information. While Spline performs competitively on clean ImageNet, it deteriorates under corruption with values 5–7× higher than SMART. This degradation reveals a limitation of non-parametric methods: they can overfit to validation distributions and fail when input characteristics change. SMART's focus on decision boundary uncertainty via the margin remains informative even when input distributions shift substantially.

**Long-Tailed Distribution Calibration.** As shown in Figure 9, the ImageNet-LT results reveal that class imbalance presents a different calibration challenge than input corruption. Interestingly, Spline performs competitively here, suggesting

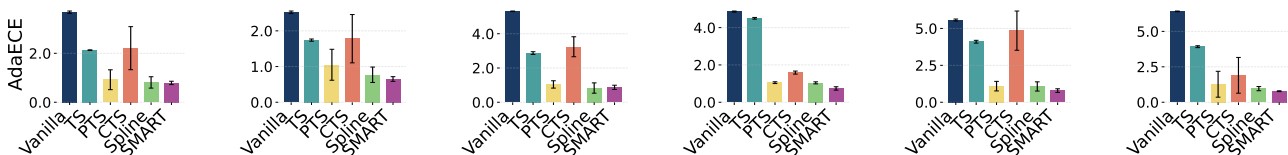

*Figure 7.* **AdaECE (↓, %, 15 bins) comparison on ImageNet-1K.** SMART delivers consistent calibration across diverse architectures, from CNNs to vision transformers. From left to right are ResNet-50, DenseNet-121, Wide-ResNet, Swin-B, ViT-B/16, ViT-B/32.



*Figure 8.* **AdaECE (↓, %, 15 bins) comparison on ImageNet-C.** SMART maintains strong calibration under corruption, while Spline and TS-based methods demonstrate significant degradation. From left to right are ResNet-50, DenseNet-121, Swin-B, ViT-B/16, ViT-B/32.

non-parametric methods can handle statistical imbalances better than distributional shifts. However, CTS underperforms despite being explicitly designed for per-class variations, demonstrating that simply applying different temperatures per class is insufficient for complex imbalanced scenarios.

**Extreme Domain Shift Calibration.** The sketch-based domain shift represents the most challenging calibration scenario in Figure 10, where SMART demonstrates its largest advantage. Spline's degradation here reinforces the brittleness of non-parametric methods under distribution shifts, while SMART's consistent performance across architectures suggests that margin information captures robust uncertainty signals across input characteristics and domains.

## G. Comparison of Various Training-Time Calibration Methods on Other Metrics

This section presents a comprehensive evaluation of SMART when combined with various training-time calibration methods across multiple metrics, extending the ECE analysis provided in Section 4.3. We examine SMART's performance using AdaECE, Classwise ECE (CECE), Negative Log-Likelihood (NLL), and classification accuracy.

### G.1. Accuracy Preservation

*Table 10.* **Comparison of Training-Time Calibration Methods Using Accuracy (↑, %) Across Various Datasets and Models.** Results demonstrate that SMART preserves the original model accuracy across all training methods. Values follow the same five-seed protocol as the main text.

| Dataset | Model | NLL | | Brier Loss | | MMCE | | LS-0.05 | | FLSD-53 | | FL-3 | |
|---|---|---|---|---|---|---|---|---|---|---|---|---|---|
| | | base | ours | base | ours | base | ours | base | ours | base | ours | base | ours |
| CIFAR-10 | ResNet-50 | 95.1 | 95.1 | 95.0 | 95.0 | 95.0 | 95.0 | 94.7 | 94.7 | 95.0 | 95.0 | 94.8 | 94.8 |
| | ResNet-110 | 95.1 | 95.1 | 94.5 | 94.5 | 94.6 | 94.6 | 94.5 | 94.5 | 94.6 | 94.6 | 94.9 | 94.9 |
| | DenseNet-121 | 95.0 | 95.0 | 94.9 | 94.9 | 94.6 | 94.6 | 94.9 | 94.9 | 94.6 | 94.6 | 94.7 | 94.7 |
| | Wide-ResNet | 96.1 | 96.1 | 95.9 | 95.9 | 96.1 | 96.1 | 95.8 | 95.8 | 96.0 | 96.0 | 95.9 | 95.9 |
| CIFAR-100 | ResNet-50 | 76.7 | 76.7 | 76.6 | 76.6 | 76.8 | 76.8 | 76.6 | 76.6 | 76.8 | 76.8 | 77.3 | 77.3 |
| | ResNet-110 | 77.3 | 77.3 | 74.9 | 74.9 | 76.9 | 76.9 | 76.6 | 76.6 | 77.5 | 77.5 | 77.1 | 77.1 |
| | DenseNet-121 | 75.5 | 75.5 | 76.3 | 76.3 | 76.0 | 76.0 | 75.9 | 75.9 | 77.3 | 77.3 | 76.8 | 76.8 |
| | Wide-ResNet | 79.3 | 79.3 | 79.4 | 79.4 | 79.3 | 79.3 | 78.8 | 78.8 | 79.9 | 79.9 | 80.3 | 80.3 |

**Accuracy Analysis** As shown in Table 10, SMART consistently preserves the classification accuracy of all base models across all training-time calibration methods. This is a critical property of post-hoc calibration methods, as improving confidence estimates should not come at the cost of predictive performance. Accuracy preservation is by design, as SMART's positive temperature scaling does not alter the relative logit ordering and therefore maintains the same class predictions. This contrasts with some training-time methods that may involve trade-offs between accuracy and calibration quality during model optimization. The preservation of accuracy across diverse architectures and datasets further supports SMART's

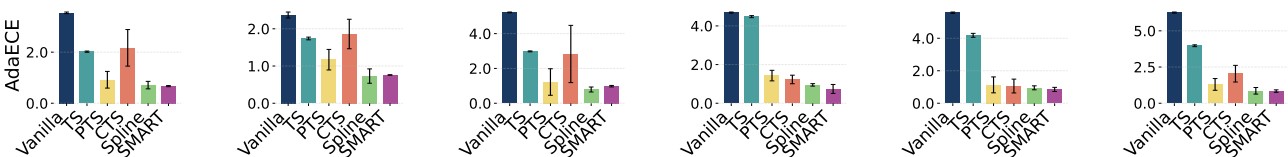

*Figure 9.* **AdaECE (↓, %, 15 bins) comparison on ImageNet-LT.** SMART maintains strong calibration under long-tailed class distributions, particularly on CNN architectures. From left to right are ResNet-50, DenseNet-121, Wide-ResNet, Swin-B, ViT-B/16, ViT-B/32.



*Figure 10.* **AdaECE (↓, %, 15 bins) comparison on ImageNet-Sketch.** SMART maintains strong calibration under extreme domain shift, while Spline struggles significantly. From left to right are ResNet-50, DenseNet-121, Swin-B, ViT-B/16, ViT-B/32.

practical utility when maintaining predictive performance is essential.

## G.2. AdaECE Performance

*Table 11.* **Comparison of Training-Time Calibration Methods Using AdaECE (↓, %, 15 bins) Across Various Datasets and Models.** The best-performing method for each dataset-model combination is in bold, and our method (SMART) is highlighted. Values follow the same five-seed protocol as the main text.

| Dataset | Model | NLL | | Brier Loss | | MMCE | | LS-0.05 | | FLSD-53 | | FL-3 | |
|---|---|---|---|---|---|---|---|---|---|---|---|---|---|
| | | base | ours | base | ours | base | ours | base | ours | base | ours | base | ours |
| CIFAR-10 | ResNet-50 | 4.33 | **0.80** | 1.74 | **1.01** | 4.55 | **0.67** | 3.88 | **2.18** | 1.56 | **0.45** | 1.95 | **0.48** |
| | ResNet-110 | 4.40 | **1.22** | 2.61 | **0.56** | 5.07 | **0.93** | 4.46 | **3.66** | 2.07 | **0.40** | 1.64 | **0.52** |
| | DenseNet-121 | 4.49 | **0.61** | 2.01 | **0.51** | 5.10 | **0.96** | 4.40 | **2.95** | 1.38 | **0.62** | 1.23 | **0.83** |
| | Wide-ResNet | 3.24 | **0.44** | 1.70 | **0.44** | 3.29 | **0.53** | 4.27 | **0.97** | 1.52 | **0.44** | 1.84 | **0.59** |
| CIFAR-100 | ResNet-50 | 17.53 | **1.00** | 6.54 | **1.41** | 15.31 | **1.08** | 7.63 | **1.75** | 4.40 | **1.35** | 5.08 | **0.95** |
| | ResNet-110 | 19.06 | **1.67** | 7.73 | **0.93** | 19.13 | **1.98** | 11.07 | **2.72** | 8.54 | **0.93** | 8.65 | **1.22** |
| | DenseNet-121 | 20.99 | **2.23** | 5.04 | **1.02** | 19.10 | **1.73** | 12.83 | **1.96** | 3.54 | **0.93** | 4.14 | **0.97** |
| | Wide-ResNet | 15.34 | **1.55** | 4.28 | **0.97** | 13.16 | **1.12** | 5.13 | **2.11** | 2.77 | **0.75** | 2.07 | **1.15** |

**AdaECE Analysis** The adaptive ECE results in Table 11 provide further validation of SMART's effectiveness when combined with various training-time calibration methods. AdaECE, which uses adaptive binning to ensure equal sample counts in each bin, offers a more robust calibration measure than standard ECE by reducing potential biases from uneven confidence distributions. SMART consistently improves AdaECE across all training methods, with particularly large improvements for models trained with NLL and MMCE, where we observe reductions of up to 18× (17.53% to 1.00% for CIFAR-100 ResNet-50).

The most substantial AdaECE improvements occur on CIFAR-100, which has ten times more classes than CIFAR-10 and thus represents a more challenging calibration scenario. This suggests that SMART's effectiveness scales favorably with task complexity. Even for models already trained with calibration-oriented objectives like Focal Loss or FLSD, SMART provides further substantial improvements, indicating that its margin-based temperature adjustment captures complementary information to these training-time approaches. Notably, the combination of SMART with FLSD-53 achieves some of the lowest overall AdaECE values (e.g., 0.40% on CIFAR-10 ResNet-110), suggesting a particularly effective synergy between these methods.

## G.3. Classwise ECE Performance

**CECE Analysis** Classwise ECE (CECE) provides insights into calibration performance at the individual class level rather than aggregated across all classes. The formula for classwise ECE is:

*Table 12.* **Comparison of Training-Time Calibration Methods Using Classwise ECE (↓, %, 15 bins) Across Various Datasets and Models.** The best-performing method for each dataset-model combination is in bold, and our method (SMART) is highlighted. Values follow the same five-seed protocol as the main text.

| Dataset | Model | NLL | | Brier Loss | | MMCE | | LS-0.05 | | FLSD-53 | | FL-3 | |
|---|---|---|---|---|---|---|---|---|---|---|---|---|---|
| | | base | ours | base | ours | base | ours | base | ours | base | ours | base | ours |
| CIFAR-10 | ResNet-50 | 0.91 | **0.43** | 0.46 | **0.40** | 0.94 | **0.51** | 0.71 | **0.51** | 0.42 | **0.37** | 0.43 | **0.38** |
| | ResNet-110 | 0.92 | **0.49** | 0.59 | **0.45** | 1.04 | **0.54** | 0.66 | **0.54** | 0.47 | **0.41** | 0.44 | **0.38** |
| | DenseNet-121 | 0.92 | **0.45** | 0.46 | **0.41** | 1.04 | **0.59** | 0.60 | **0.50** | 0.41 | **0.38** | 0.42 | **0.35** |
| | Wide-ResNet | 0.68 | **0.37** | 0.44 | **0.39** | 0.70 | **0.38** | 0.79 | **0.40** | 0.41 | **0.29** | 0.44 | **0.34** |
| CIFAR-100 | ResNet-50 | 0.38 | **0.21** | 0.22 | **0.20** | 0.34 | **0.20** | 0.23 | **0.21** | 0.20 | **0.20** | 0.20 | **0.20** |
| | ResNet-110 | 0.41 | **0.20** | 0.24 | **0.21** | 0.42 | **0.21** | 0.26 | **0.20** | 0.24 | **0.20** | 0.24 | **0.21** |
| | DenseNet-121 | 0.45 | **0.23** | 0.20 | **0.20** | 0.42 | **0.23** | 0.29 | **0.21** | **0.19** | 0.20 | 0.20 | **0.20** |
| | Wide-ResNet | 0.34 | **0.19** | 0.19 | **0.19** | 0.30 | **0.19** | 0.21 | **0.20** | 0.18 | **0.18** | 0.18 | **0.18** |

$$\text{Classwise-ECE} = \frac{1}{\mathcal{K}} \sum_{i=1}^{B} \sum_{j=1}^{\mathcal{K}} \frac{|B_{i,j}|}{N} |I_{i,j} - C_{i,j}| \tag{40}$$

where the calibration error is computed separately for each class $j$ across all bins $i$, then averaged across all $\mathcal{K}$ classes. This metric is particularly valuable for understanding whether calibration improvements are uniformly distributed across classes or concentrated in specific categories.

Table 12 demonstrates SMART's ability to improve per-class calibration across almost all training methods and architectures. The improvements are particularly prominent for models trained with NLL and MMCE, where CECE values are typically reduced by 50% or more after applying SMART (e.g., from 0.91% to 0.43% for CIFAR-10 ResNet-50). This substantial improvement suggests that SMART's margin-based temperature scaling effectively addresses class-specific miscalibration patterns that may arise during training with these standard objectives.

Interestingly, CECE values are consistently lower on CIFAR-100 compared to CIFAR-10 despite the higher class count, which contrasts with the pattern observed for ECE and AdaECE. This phenomenon occurs because CECE averages calibration errors across classes, and with 100 classes, individual class miscalibrations tend to average out more effectively than with only 10 classes. Additionally, the higher granularity of class divisions in CIFAR-100 may lead to more balanced per-class confidence distributions, making the averaging effect more pronounced.

For models already trained with calibration-oriented losses like FLSD-53 and FL-3, SMART provides more modest improvements in CECE, and in a few cases maintains the same level of performance. This suggests that these training-time methods are already effective at addressing per-class calibration issues through their specialized loss formulations that inherently consider class-wise balance. However, SMART can still provide complementary benefits in most scenarios, particularly for classes that may remain poorly calibrated even after specialized training procedures.

### G.4. Negative Log-Likelihood Performance

*Table 13.* **Comparison of Training-Time Calibration Methods Using NLL (↓, values multiplied by 100) Across Various Datasets and Models.** The best-performing method for each dataset-model combination is in bold, and our method (SMART) is highlighted. Values follow the same five-seed protocol as the main text.

| Dataset | Model | NLL | | Brier Loss | | MMCE | | LS-0.05 | | FLSD-53 | | FL-3 | |
|---|---|---|---|---|---|---|---|---|---|---|---|---|---|
| | | Base | Ours | Base | Ours | Base | Ours | Base | Ours | Base | Ours | Base | Ours |
| CIFAR-10 | ResNet-50 | 41.2 | **19.7** | 18.7 | **18.4** | 44.8 | **21.0** | 27.7 | **27.7** | 17.6 | **17.1** | 18.4 | **17.9** |
| | ResNet-110 | 47.5 | **22.5** | 20.4 | **19.4** | 55.7 | **23.6** | 29.9 | **29.4** | 18.5 | **17.9** | 17.8 | **17.3** |
| | DenseNet-121 | 42.9 | **20.8** | 19.1 | **18.6** | 52.1 | **24.1** | 28.7 | **28.7** | 18.4 | **18.1** | 18.0 | **17.9** |
| | Wide-ResNet | 26.8 | **14.9** | 15.9 | **15.4** | 28.5 | **15.9** | 21.7 | **19.9** | 14.6 | **13.7** | 15.2 | **14.9** |
| CIFAR-100 | ResNet-50 | 153.7 | **105.3** | 99.6 | **99.5** | 125.3 | **100.7** | 121.0 | **120.1** | 88.0 | 88.4 | 87.5 | 88.1 |
| | ResNet-110 | 179.2 | **104.0** | 110.7 | **110.0** | 180.6 | **106.1** | 133.1 | **128.8** | 89.9 | **88.3** | 90.9 | **90.0** |
| | DenseNet-121 | 205.6 | **119.1** | 98.3 | **98.9** | 166.6 | **112.6** | 142.0 | **134.3** | 85.5 | 86.5 | 87.1 | 87.3 |
| | Wide-ResNet | 140.1 | **95.2** | 84.6 | 84.9 | 119.6 | **94.1** | 108.1 | **106.5** | 76.9 | 77.4 | 74.7 | 75.8 |

**NLL Analysis**   NLL is a likelihood-based proper score that reflects calibration together with sharpness and discrimination. Table 13 shows that SMART improves NLL for most models, with the largest gains observed for NLL, MMCE, and LS-0.05

trained models. The improvements are particularly clear for CIFAR-10, where NLL is reduced by up to 60% after applying SMART (e.g., 41.22 to 19.70 for ResNet-50 with NLL).

However, a different pattern emerges for models trained with specialized losses like FLSD-53 and FL-3 on CIFAR-100, where SMART sometimes leads to slight increases in NLL despite improvements in calibration metrics like ECE and AdaECE. This reflects the trade-off discussed in the main text: a calibration-centric rescaling can move differently from likelihood when reliability and sharpness prefer different adjustments. Nevertheless, the overall trend across metrics indicates that SMART maintains or improves model performance in most cases.

## H. Calibration Performance Under Specific Corruption Types

To provide deeper insights into SMART's robustness across different corruption scenarios, we examine the calibration error reduction achieved by various methods on individual corruption types in ImageNet-C. We analyze performance across two architectures (ResNet-50 and ViT-B/16) and two metrics (ECE and AdaECE), providing a comprehensive view of how different calibration approaches respond to specific distribution shifts. This granular analysis helps understand which corruption types pose the greatest calibration challenges and how architectural differences influence calibration robustness.

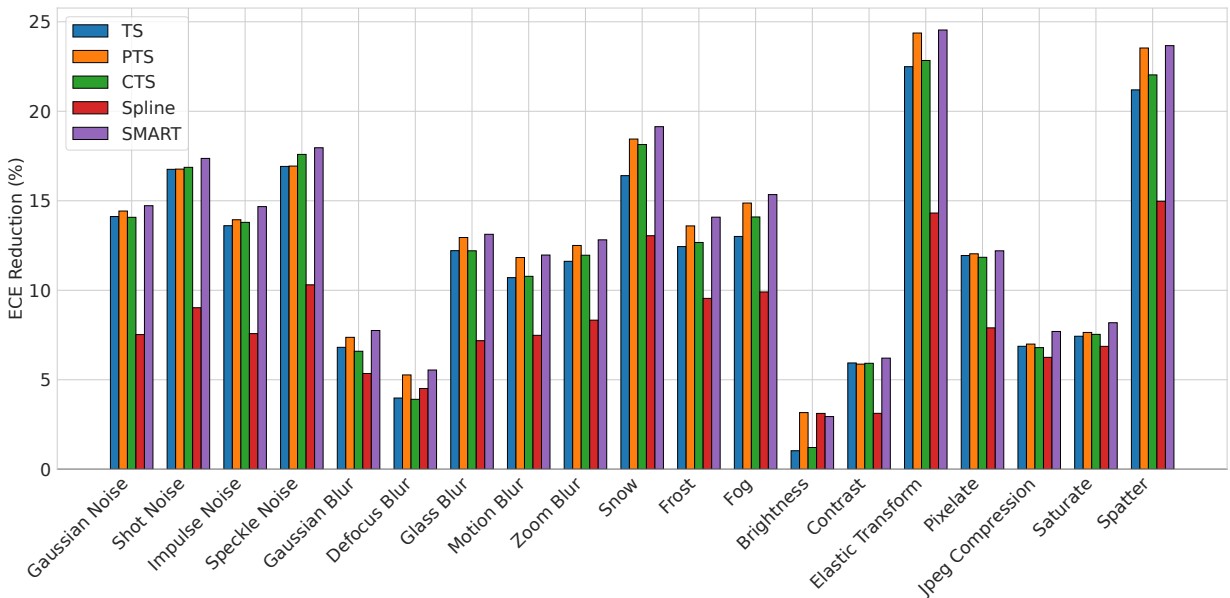

*Figure 11.* **ECE reduction (↑, %, 15 bins) across corruption types for ResNet-50.** SMART consistently achieves strong calibration improvements across diverse corruption scenarios, demonstrating robustness to distribution shifts.

**ResNet-50 ECE Analysis** The corruption-specific analysis reveals that SMART demonstrates strong consistency, achieving the highest ECE reduction across most corruption categories with improvements often exceeding 20%. The inclusion of Spline calibration exposes a limitation of non-parametric methods: brittleness under distribution shifts. While Spline achieves competitive results on certain corruptions like Snow, it degrades on others such as Brightness and Contrast, highlighting how non-parametric approaches can overfit to validation characteristics and break down when faced with novel corruptions.

This contrasts with SMART's robust performance across all corruption types. The key insight is that SMART's margin indicator captures decision boundary information that remains meaningful across input degradation types, including geometric distortions, noise, and digital artifacts. Temperature Scaling and other global methods show predictable limitations on uniform corruptions, while parametric methods like PTS exhibit moderate consistency but still significant variability. SMART's sample-specific adaptation based on decision boundary information provides reliable calibration improvements for scenarios where corruption characteristics are unpredictable.

**ResNet-50 AdaECE Analysis** The AdaECE results closely mirror the ECE patterns, suggesting that SMART's calibration improvements are not artifacts of fixed-width binning. SMART achieves the highest reduction rates across most corruptions,

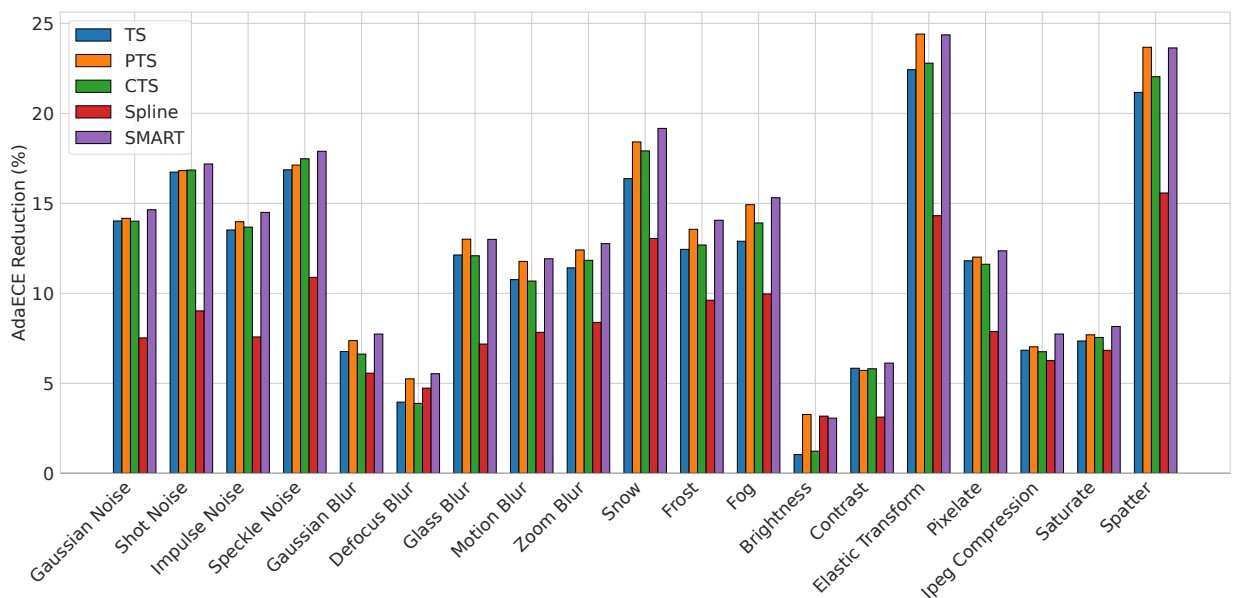

*Figure 12.* **AdaECE reduction (↑, %, 15 bins) across corruption types for ResNet-50.** SMART maintains consistent improvements across corruption types under adaptive binning, confirming robust calibration trends under a second metric.

with particularly strong performance on geometric distortions approaching 25% improvement. Spline's brittleness persists under adaptive binning, performing reasonably on weather corruptions but failing on uniform transforms, which suggests that its limitations stem from overfitting rather than evaluation methodology. The similar performance rankings across both metrics indicate that SMART's margin approach captures robust calibration signals under both evaluation protocols.

**ViT-B/16 ECE Analysis**    The transformer results reveal striking architectural differences in calibration behavior under corruption. Most notably, Temperature Scaling frequently worsens calibration, showing negative improvements on multiple corruption types including Shot Noise, Speckle Noise, Snow, Brightness, Pixelate, Jpeg Compression, Saturate, and Spatter. This suggests that transformers' attention mechanisms and different inductive biases can make global temperature adjustments unreliable under distribution shifts.

SMART maintains consistent positive improvements across all corruption types, though generally more modest than with ResNet-50. This architectural difference suggests that while transformers are inherently better calibrated, they also present unique challenges that require more sophisticated approaches than global scaling. The convergence of all methods on Fog corruption (around 25% improvement) indicates that certain atmospheric corruptions create calibration conditions where architectural differences become less relevant.

A key insight emerges: the margin's decision boundary information remains meaningful across architectures, while global statistics become unreliable for transformers under corruption. PTS and CTS show more consistent improvements than TS, but SMART's sample-specific adaptation consistently outperforms all alternatives, confirming its architectural robustness.

**ViT-B/16 AdaECE Analysis**    The AdaECE results closely replicate the ECE patterns, suggesting that the transformer calibration trends are not artifacts of the binning protocol. Temperature Scaling's negative performance persists under adaptive binning, while SMART maintains consistent positive improvements across all corruption types. This metric-level agreement indicates that SMART's margin approach captures robust decision boundary information under both ECE and AdaECE.

# I. Margin Perspective on Calibration

Traditional calibration analysis evaluates models from an overall perspective, potentially masking important sample-specific miscalibration patterns. By examining calibration behavior across margin values, we uncover how neural networks distribute confidence and validate our method visually.

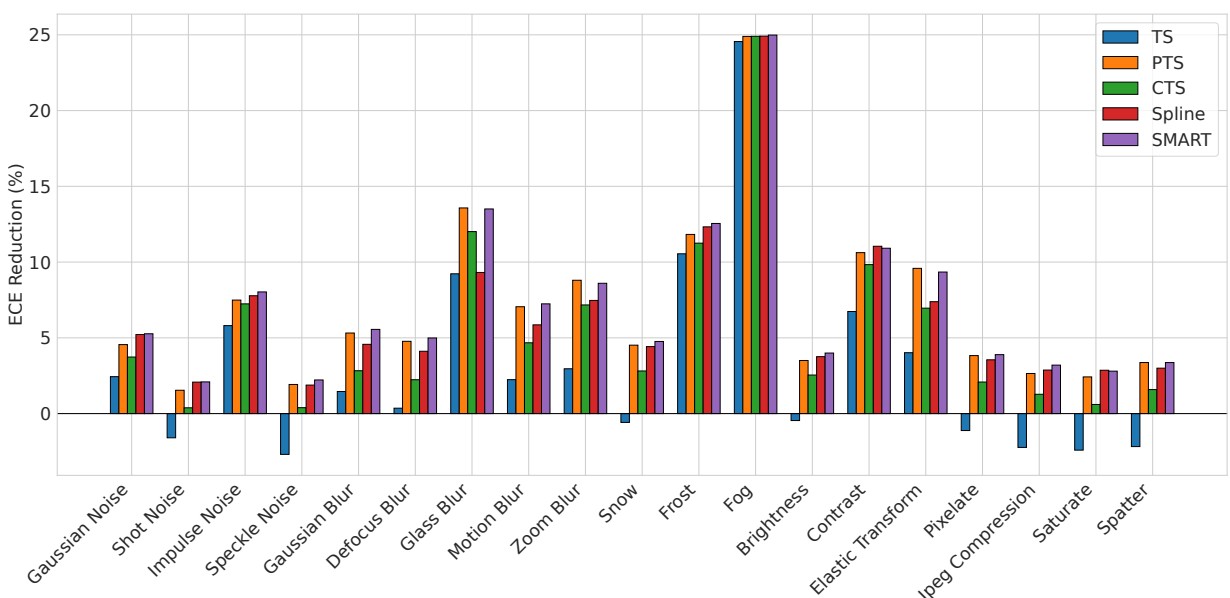

*Figure 13.* **ECE reduction (↑, %, 15 bins) across corruption types for ViT-B/16.** Transformer architectures exhibit distinct calibration challenges under corruption, with global methods often failing while SMART maintains consistent improvements.

Figure 15 demonstrates heterogeneity across margin groups. For ImageNet with ViT-B/16, while overall calibration appears near-perfect (Figure 15a), decomposing by margin reveals distinct patterns: low margin samples achieve good calibration (Figure 15c), while high margin samples show systematic under-confidence (Figure 15b). This pattern persists across different conditions, as shown in CIFAR-100 with ResNet-50 (Figures 15d and 15e), indicating that margin-based groupings reveal consistent calibration characteristics beyond dataset-specific or architecture-specific behavior.

**The Under-Confidence Paradox in High Margin Samples**   Perhaps the most counterintuitive finding emerges from examining high margin samples. Despite representing easy classifications with substantial separation between top predictions, these samples consistently exhibit under-confidence rather than expected over-confidence. High margin samples from ImageNet ViT-B/16 show systematic under-confidence, with predicted confidence consistently lower than empirical accuracy (Figure 15b). This pattern persists even when overall model behavior differs dramatically, as CIFAR-100 ResNet-50 maintains under-confidence in high margin samples despite overall over-confidence (Figure 15e).

**Method-Specific Failures from the Margin Perspective**   The confidence adjustment patterns in Figure 15f expose limitations in existing approaches. Temperature Scaling's uniform adjustment ignores heterogeneous calibration needs across margin groups, applying identical modifications regardless of sample characteristics. More critically, PTS makes substantial adjustments to low margin samples that already achieve good calibration and require minimal intervention. This unnecessary manipulation exemplifies how increased dimensionality introduces noise for precise temperature parameterization. In contrast, SMART provides minimal adjustments to low margin samples that are already well-calibrated, while delivering targeted confidence increases to high margin samples suffering from under-confidence. This adaptive behavior emerges naturally from our lightweight margin-to-temperature mapping, demonstrating how principled architectural choices translate into appropriate calibration strategies.

## J. Implementation Details

All experiments employ a training-time batch size of 1024 and run on a single NVIDIA RTX 3090 GPU. CIFAR-10 and CIFAR-100 contain 60,000 images of size $32 \times 32$ pixels, with 10 and 100 classes, respectively, split into 45,000 training, 5,000 validation, and 10,000 test images. For ImageNet-related datasets, we use 20% of the official validation set as the new validation set, with the remainder used as the test set. The testing batch size for all datasets is set to 128.

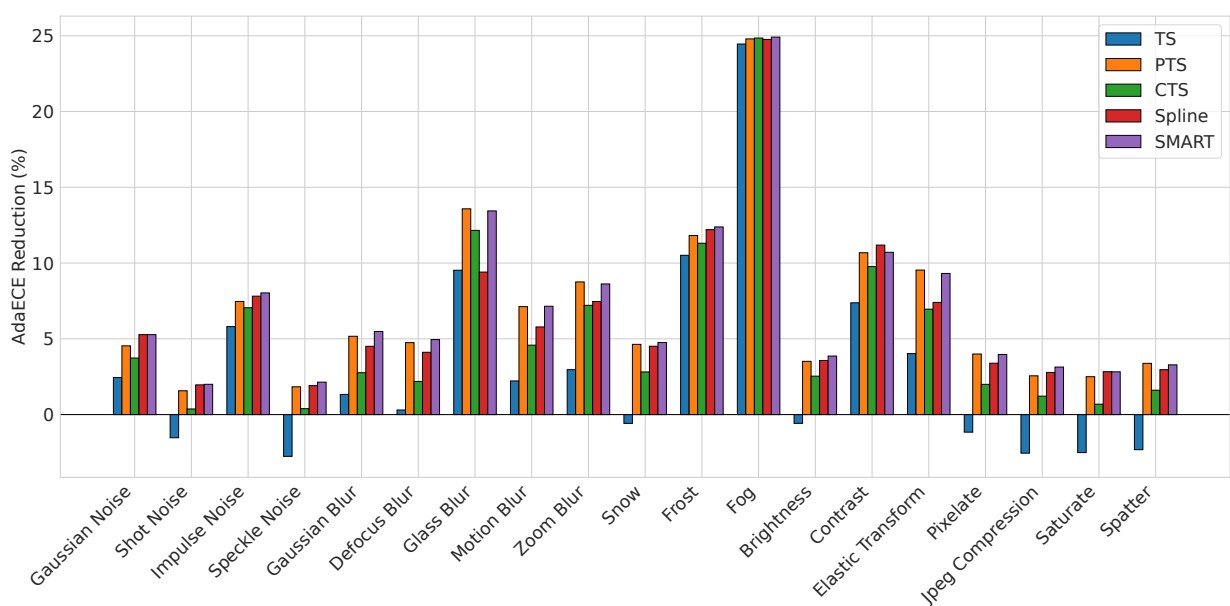

*Figure 14.* **AdaECE reduction (↑, %, 15 bins) across corruption types for ViT-B/16.** Transformer calibration patterns remain consistent under adaptive binning, confirming architectural-specific calibration challenges and SMART's robustness.

### J.1. Training Recipe for Training-Time Calibration Methods

For CIFAR-10 and CIFAR-100 networks, we use the pretrained weights provided by Mukhoti et al. (2020) and follow their training recipe. We use SGD with momentum 0.9 and train for 350 epochs, with learning rate 0.1 for the first 150 epochs, 0.01 for the next 100 epochs, and 0.001 for the last 100 epochs. We use a training batch size of 128 and augment training images with random crops and random horizontal flips.

### J.2. Sensitivity to Hyperparameters $\sigma$ and $\delta$

We examine the sensitivity of SMART's performance to the bandwidth parameter $\sigma$ and Charbonnier smoothing parameter $\delta$ in Equation (6). Tables 14 and 15 report ECE (15 bins) on ImageNet for ResNet-50 and ViT-B/16 across different $(\sigma, \delta)$ combinations.

*Table 14.* ECE ($\downarrow$, %, 15 bins) on ImageNet ResNet-50 for different $(\sigma, \delta)$ combinations.

| $\sigma \backslash \delta$ | 0.001 | 0.010 | 0.100 | 1.000 |
|---|---|---|---|---|
| 0.01 | 0.66 | 0.83 | 1.02 | 0.67 |
| 0.05 | 0.61 | 0.66 | 0.67 | 0.66 |
| 0.10 | 0.66 | 3.11 | 0.71 | 0.72 |
| 0.50 | 0.85 | 0.95 | 1.29 | 1.26 |
| 1.00 | 0.79 | 1.15 | 2.49 | 2.51 |

*Table 15.* ECE ($\downarrow$, %, 15 bins) on ImageNet ViT-B/16 for different $(\sigma, \delta)$ combinations.

| $\sigma \backslash \delta$ | 0.001 | 0.010 | 0.100 | 1.000 |
|---|---|---|---|---|
| 0.01 | 1.32 | 0.84 | 0.78 | 0.85 |
| 0.05 | 0.84 | 0.80 | 0.89 | 0.86 |
| 0.10 | 0.99 | 0.97 | 0.81 | 2.26 |
| 0.50 | 2.09 | 2.02 | 2.06 | 2.05 |
| 1.00 | 2.04 | 2.48 | 2.56 | 2.56 |

The results demonstrate that performance remains stable within a reasonable range of values. For $\sigma \in \{0.01, 0.05, 0.10\}$, ECE varies by less than 0.2% across different $\delta$ choices, indicating robustness to the Charbonnier smoothing parameter.

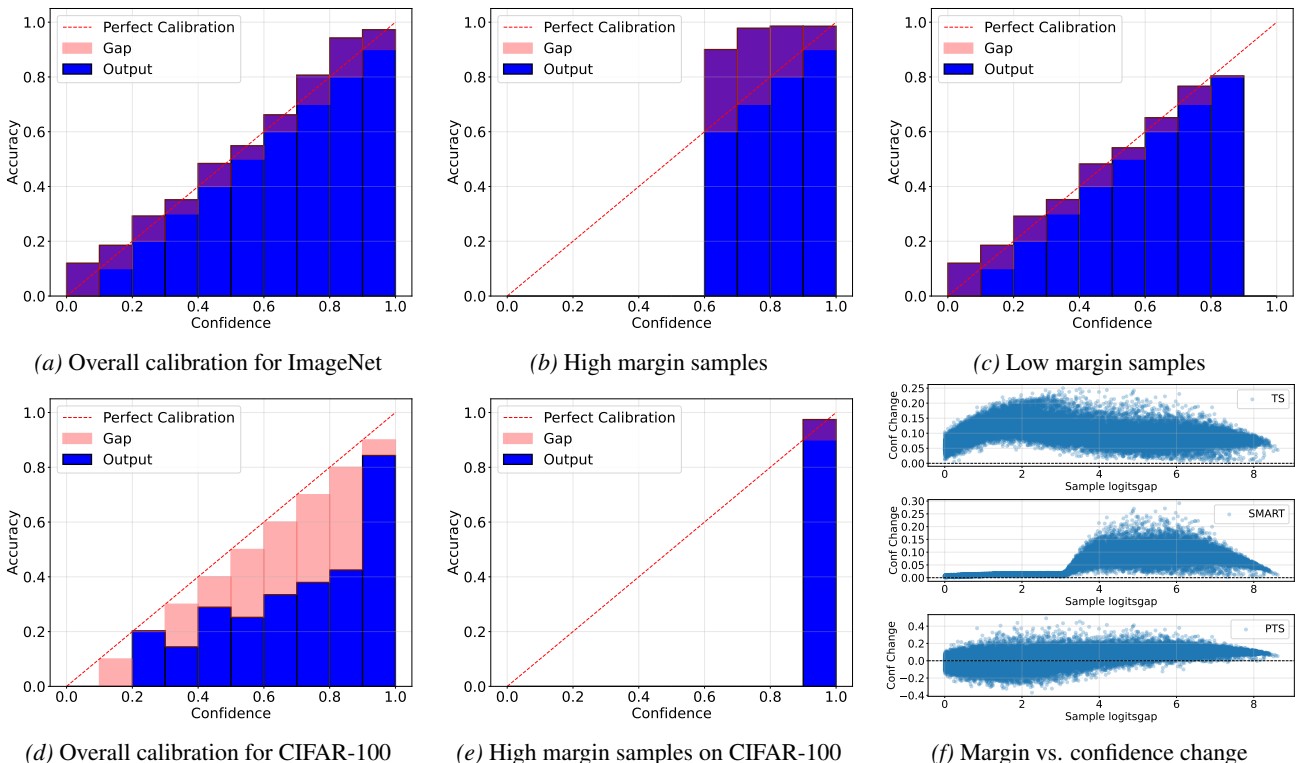

*(a)* Overall calibration for ImageNet     *(b)* High margin samples     *(c)* Low margin samples

*(d)* Overall calibration for CIFAR-100     *(e)* High margin samples on CIFAR-100     *(f)* Margin vs. confidence change

*Figure 15.* **Margin reveals hidden calibration patterns across the confidence spectrum.** ImageNet ViT-B/16 shows near-perfect overall calibration (a) but reveals systematic under-confidence in high margin samples (b) and well-calibrated low margin samples (c). CIFAR-100 ResNet-50 demonstrates that even with overall over-confidence (d), high margin samples remain under-confident (e). Panel (f) shows that SMART provides targeted adjustments while TS and PTS show suboptimal patterns.

Larger values ($\sigma \geq 0.50$) lead to degraded performance due to over-localization of kernel weights, creating high variance in calibration estimates. Our choice of $\sigma = 0.05$ and $\delta = 0.001$ (highlighted rows) provides consistent performance across both CNN and transformer architectures, though the method is not particularly sensitive to $\delta$ within the range $[0.001, 0.100]$ when $\sigma$ is appropriately chosen.

## K. Margin–Temperature Relationship

Figure 16 illustrates that the learned margin–temperature mapping is not constrained to be monotonic. For ImageNet ResNet-50, the mapping closely follows an increasing linear trend: samples with larger logit margins receive higher temperatures (softer probabilities), while low-margin samples are assigned temperatures closer to one. In contrast, on ImageNet with ViT-B/16 the mapping is clearly non-monotonic, with an approximately U-shaped dependence on the margin. This behavior indicates that the relationship between margin and miscalibration is architecture- and dataset-dependent; SMART adapts to these differences rather than enforcing a fixed monotone form, and understanding the underlying theoretical reasons is left for future work.

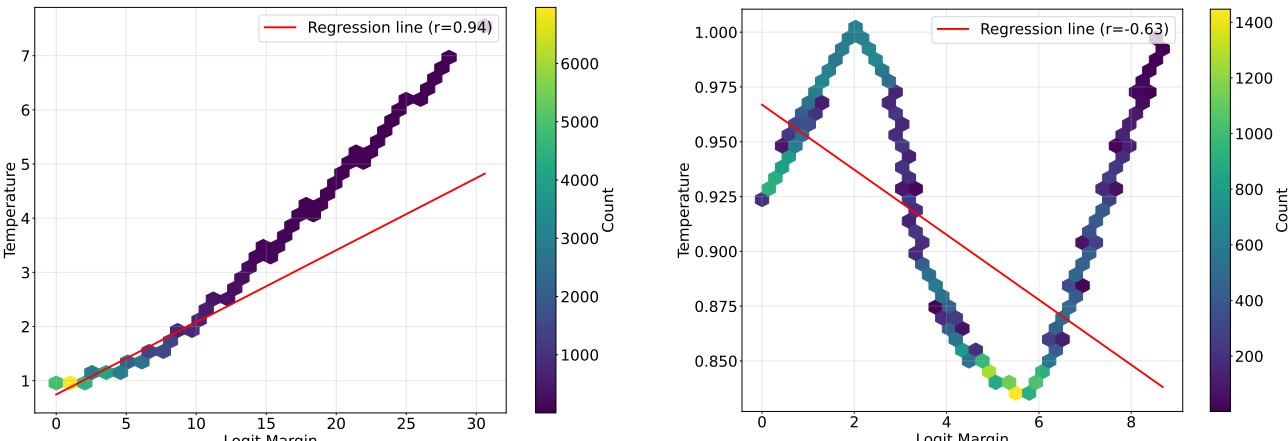

*Figure 16.* Empirical margin–temperature relationship learned by SMART. Left: ImageNet with a ResNet-50 backbone, where the mapping is approximately linear and monotone increasing (Pearson $r = 0.94$). Right: ImageNet with a ViT-B/16 backbone, where the mapping becomes non-monotonic with a pronounced U-shaped pattern (Pearson $r = -0.63$ for the best linear fit).

