# OpenReview forum: "Sample Margin-Aware Recalibration of Temperature Scaling"
_ICML.cc/2026/Conference — ICML 2026 regular_

### Official Review · Reviewer_MZzk · 2026-02-18

**Soundness:** 2
**Presentation:** 2
**Significance:** 1
**Originality:** 1
**Overall Recommendation:** 3
**Confidence:** 4

**Summary:**

This work proposes method of SMART, a post-hoc, sample-wise temperature scaling that uses a single scalar per sample by the logit gap. It also points out that minimizing NLL only can worsen calibration and introduce SoftECE to solve it. Experiemnts have been completed on C10/100 and imageNet.

**Compliance With Llm Reviewing Policy:**

Affirmed.

**Ethical Review Concerns:**

N/A.

**Final Justification:**

Maintain my score as the last comment replied.

**Key Questions For Authors:**

1. A technical bet: “logit gap is a sufficient statistic for how much to scale”? (Sufficiency)
2. SMART is very much in the same family as Relaxed Softmax: [1] “add/learn one extra scalar per sample to control softmax confidence.” The main distinction is where it’s trained (in-training vs post-hoc) and what drives T (learned signal vs gap-only). How does SMART differ empirically and conceptually from prior sample-wise temperature approaches?
3. SoftECE used Charbonnier, can other types of kernels be ajusted into this loss and what are the impacts? an ablation for different kernels can be considered.
4. Is the learned proxies monotone? if not, can you provide full theoretical explanation?
5. Since [2] already provides a well-defined proof that upper-bounds smCE, the paper should make explicit whether its proof differs in any substantive way (assumptions, smooth surrogate definition, kernel choice, or tightness). Otherwise, this theoretical component should be framed as prior art rather than a core novelty claim.

[1] Relaxed Softmax: Efficient Confidence Auto-Calibration for Safe Pedestrian Detection, 2018 NIPS Workshop on Machine Learning for Intelligent Transportation Systems.

[2] Błasiok, J. and Nakkiran, P. Smooth ece: Principled reliability diagrams via kernel smoothing. arXiv preprint arXiv:2309.12236, 2023.

**Limitations:**

SMART is not a fundamentally new calibration family; it’s a carefully regularized instance-wise TS where the major technical choice is the conditioning statistic (logit gap) + metric-aligned training (SoftECE). Its strength is statistical: low-dimensional conditioning reduces variance under scarce calibration data. Its weakness is expressivity: gap-only cannot represent class-conditional or multi-modal uncertainty effects, and ECE-like optimization can be metric-specific unless validated with proper scores.
see Weakness.

**Strengths And Weaknesses:**

Strengths

1. Sample-wise temperature scaling. The method calibrates predictions using per-sample temperatures rather than a single global temperature, which can better accommodate instance-level difficulty and heterogeneity.

2. Margin-based uncertainty proxy with good scalability. Using the logit margin (gap) as an uncertainty signal avoids feeding a potentially noisy, high-dimensional logit vector making the approach more scalable to large-class (large-K) settings. The paper also provides a boundedness-based theoretical motivation linking the margin to temperature adjustment.

3. Smooth calibration surrogate (SoftECE). The paper introduces a differentiable surrogate of calibration error using the Charbonnier kernel, enabling direct optimization during training (or calibration) rather than relying solely on non-differentiable binning-based objectives.

Weaknesses / Concerns

1. Reproducibility gap. The paper reports experiments on ViT and Swin models, but the anonymous code release does not include the corresponding implementations in the “models” directory. This discrepancy makes it difficult to verify the results and raises concerns about reproducibility.

2. Baseline missing for the core mechanism. The central operation in SMART—modifying the logit gap—appears closely related to ideas explored in earlier work (e.g., [1]). A direct comparison with such methods is necessary to clarify what is genuinely new in SMART beyond this gap adjustment, particularly in the results reported in Table 2.

3. Limited comparison with prior sample-wise calibration methods. Predicting a temperature on a per-sample basis has been studied previously. For example, [3] proposes local temperature scaling, and [2] predicts a sample-dependent temperature using a small post-hoc module. Because SMART also relies on post-hoc, sample-wise temperature prediction, the overlap with these approaches is substantial. The paper would benefit from a clearer discussion of these connections and stronger empirical comparisons with methods such as [2] and [3].

4. Objective-level concern: ECE is not a proper scoring rule. The SoftECE objective is an interesting design choice but introduces additional hyperparameters. More importantly, ECE-based objectives are diagnostic rather than proper scoring rules: minimizing them does not guarantee recovery of the true conditional probabilities. By contrast, NLL and Brier score are proper. As a result, a method may improve ECE while degrading likelihood or overall probability quality in some settings. It would therefore be helpful to report NLL and Brier alongside ECE and to discuss cases in which optimizing SoftECE might lead to undesirable behavior.

5. Broader calibration baselines are missing. Since the work focuses on uncertainty calibration, the empirical evaluation would be stronger if it included comparisons with other commonly used approaches, such as ensemble methods, Bayesian or variational techniques, or other uncertainty-aware calibration strategies, rather than focusing mainly on temperature-scaling variants.

6. Insufficient coverage of recent related work. Several recent and relevant baselines (e.g., [4–10]) are not included in the evaluation, making it difficult to assess how SMART compares with current approaches. In particular, methods such as [4] and [9], which are also based on post-hoc scaling, appear closely related and would be important points of comparison.
References

[1] Tao, L., Dong, M., and Xu, C. Dual focal loss for calibration. In International Conference on Machine Learning, pp. 33833–33849. PMLR, 2023.

[2] Joy, Tom, et al. "Sample-dependent adaptive temperature scaling for improved calibration." Proceedings of the AAAI Conference on Artificial Intelligence, Vol. 37, No. 12, 2023.

[3] Ding, Z., Han, X., Liu, P., and Niethammer, M. Local temperature scaling for probability calibration. In International Conference on Computer Vision, 2021.

[4] Zhang, S. and Xie, L., 2025, April. Parametric ρ-Norm Scaling Calibration. In Proceedings of the AAAI Conference on Artificial Intelligence (Vol. 39, No. 21, pp. 22551-22559).

[5] Kull, M., Silva Filho, T.M. and Flach, P., 2017. Beyond sigmoids: How to obtain well-calibrated probabilities from binary classifiers with beta calibration.

[6] Lin, J., Tao, L., Dong, M. and Xu, C., 2025. Uncertainty Weighted Gradients for Model Calibration. In Proceedings of the Computer Vision and Pattern Recognition Conference (pp. 15497-15507).

[7] Liu, B., Ben Ayed, I., Galdran, A. and Dolz, J., 2022. The devil is in the margin: Margin-based label smoothing for network calibration. In Proceedings of the IEEE/CVF Conference on Computer Vision and Pattern Recognition (pp. 80-88).

[8] Hebbalaguppe, R., Prakash, J., Madan, N. and Arora, C., 2022. A stitch in time saves nine: A train-time regularizing loss for improved neural network calibration. In Proceedings of the IEEE/CVF Conference on Computer Vision and Pattern Recognition (pp. 16081-16090).

[9] Dabah, L. and Tirer, T., 2024. On temperature scaling and conformal prediction of deep classifiers. arXiv preprint arXiv:2402.05806.

[10] Arad, A. and Rosset, S., 2025. Improving Multi-Class Calibration through Normalization-Aware Isotonic Techniques. arXiv preprint arXiv:2512.09054.

---

> ### Author Rebuttal · Authors · 2026-03-31
>
> Thank you for the detailed feedback.
>
> Added tables are in the paper's anonymous link.
>
> Re:W1 We updated the anonymous repository to include all models used in the paper, including ViT and Swin, with configs.
>
> Re:W2 DualFocal/AdaDualFocal are train-time methods that retrain the classifier, whereas SMART is post-hoc and preserves it. The novelty is not “gap adjustment,” but margin as a hardness signal, the NLL/calibration-mismatch analysis, and a calibration-aligned Charbonnier-SoftECE objective; see Re:W4 and Table 3 (ViT-B/16: NLL 3.62 ECE vs Char-SoftECE 0.48).
>
> Re:W3 LTS [3] is a semantic-segmentation method, not a classifier-calibration baseline. AdaTS [2] is closer, but it uses a class-conditional VAE and higher-dimensional pseudo-likelihood vector optimized with CE/NLL, whereas SMART uses a single scalar margin and a calibration-aligned objective.
>
> Re:W4[See added table] ECE/SoftECE is not a proper scoring rule; NLL and Brier are proper complementary diagnostics. But for top-confidence calibration, ECE-family metrics remain the primary target because NLL/Brier mix reliability with accuracy/resolution. this is standard in recent calibration work (Spline, Group Calibration, ProCal, Feature Clipping, DualFocal/UWG/MBLS/MDCA). SMART’s claim is calibration-centric: in our post-hoc family, NLL can move against calibration (Sec. 3.3), so we optimize a calibration-aligned surrogate. Appendix G.4 already reports NLL for train-time methods, and Table 3 shows on ViT-B/16: NLL 3.62 ECE, Brier 0.80, SoftECE 0.89, Char-SoftECE 0.48.
>
> The excel added shows the trade-off directly: on ImageNet / ResNet-50, SMART greatly improves ECE but not NLL/Brier; on ViT-B/16, it improves all three, which means SMART achieves comparative bs compared to others, not worse.
>
> Re:W5 [See added table] The paper is not TS-only, it already compares TS, PTS, CTS, Spline, Group Calibration, ProCal, Feature Clipping, and several train-time methods. We added ETS as an ensemble-style baseline.
>
> ETS (Zhang et al., 2020) is competitive on CIFAR-100 / ResNet-50, but SMART is stronger on CIFAR-10 / ResNet-50 and ImageNet / ViT-B/16. AdaTS already provides a variational sample-wise baseline.
>
> Re:W6 The added table already covers several requested recent methods, including TS4CP [9], which is still worse than SMART on ImageNet / ViT-B/16 (3.44 vs 0.83 ECE); LTS [3] is used for segmentation, PRS [4] and normalization-aware isotonic [10] have no public code, and Beta calibration [5] is binary-only.
>
> Re:Q1 We do not claim that the logit gap is a sufficient statistic in general multiclass settings. The claim is weaker: margin is a principled low-dimensional hardness signal; in the binary case it exactly determines T for a fixed target confidence, while in the multiclass case Sec. 3.2 gives bounds, not exact recovery.
>
> Re:Q2 Relaxed Softmax is not close to SMART. It is an end-to-end detector modification for pedestrian detection with an extra scalar per region; SMART is a post-hoc multiclass calibrator for fixed classifiers, driven by margin and a calibration-aligned objective.
>
> Re:Q3 [See added table] Other smooth kernels can be used. Original SoftECE uses |a-u|; Charbonnier replaces this by sqrt(r^2+delta^2), which is smooth and upper-bounds |r|. Huber,
> phi_Huber(r)=0.5 r^2/delta for |r|<=delta and |r|-0.5 delta otherwise,
> is smooth but does not upper-bound |r|.
>
> So Huber helps, but Charbonnier is best and also preserves the upper-bound interpretation.
>
> Re:Q4 We do not impose monotonicity on learned T(m), nor claim a universal monotonicity theorem. Sec. 3.2 only gives monotone relations/bounds for fixed target confidence; the learned proxy adapts to empirical calibration patterns, and Appendix K / Fig. 16 shows two examples.
>
> Re:Q5 SmoothECE (Błasiok & Nakkiran, 2023) and the related “When Does Optimizing a Proper Loss Yield Calibration?” line are important prior art, but not the same result: the latter studies when optimizing a proper loss implies smooth/weak calibration, while SmoothECE develops a measurement/reliability-diagram framework. SMART’s novelty is different: (i) margin-based feasible-temperature bounds (Sec. 3.2), (ii) an explicit NLL/calibration mismatch result inside the sample-wise TS family (Sec. 3.3), and (iii) a trainable Charbonnier-smoothed upper-bound objective (Sec. 3.4). We will credit the smooth-calibration foundation more explicitly while clarifying that SMART’s novelty is the margin-based post-hoc calibration framework and its empirical findings.
>
> Re:Limitation SMART is intentionally inside the temperature-scaling family because the paper targets lightweight post-hoc recalibration for frozen large-K classifiers. The real limitation is a deliberate bias-variance trade-off: gap-only conditioning sacrifices expressivity for robustness under scarce validation data; Table 4 shows richer indicators underperform margin, and Appendix I / Fig. 15(f) shows higher-dimensional PTS yields noisier confidence changes.

---

> > ### Author Rebuttal · Reviewer_MZzk · 2026-04-02
> >
> > Thank you for the detailed rebuttal and for adding the extra tables and repository updates. The response is helpful in several respects. In particular, the reproducibility concern is partially alleviated that the anonymous repository includes the ViT/Swin implementations and configs. The added tables also improve the empirical coverage. However, they still do not provide trained models that would allow others to directly verify the reported calibration metrics. For example, releasing a trained ViT model on ImageNet-1K would make it much easier to reproduce and validate the reported ECE and related results.
> >
> > The main reason is that my core concerns were not only about missing experiments or presentation clarity, but about the paper’s methodological positioning and novelty. On this point, I still do not think the rebuttal fully resolves the issue. The central mechanism of SMART remains a sample-wise, margin-driven scheme, and while the authors argue that this should not be viewed as “gap adjustment,” the response still does not clearly establish how far the method is from prior sample-dependent / local temperature scaling approaches in substance, rather than in implementation details or framing.
> >
> > I also still have reservations about the objective design. I appreciate that the authors now explicitly distinguish calibration-centric metrics from proper scoring rules, and that they report NLL/Brier as complementary diagnostics. However, this clarification does not fully address the underlying concern: optimizing an ECE/SoftECE-aligned surrogate may improve calibration error while leaving open the question of whether the resulting probabilities are better justified as probability estimates in a broader sense. The rebuttal provides empirical support, but not a principled resolution of this issue.
> >
> > Similarly, while more baselines are discussed, I am still not convinced that the comparisons are sufficient to isolate what is genuinely new in SMART relative to the closest sample-wise or local post-hoc calibration methods. Some of the response explains why certain methods are not directly comparable, which is fair, but the novelty gap remains not fully clarified for me.
> >
> > On the theory side, the rebuttal also makes the positioning more careful and gives more credit to prior work. I appreciate this. At the same time, this makes me feel that the theoretical component is less clearly a core novelty than the original presentation suggested.
> >
> > Last but not least, comparing the best post-hoc results in Table 1 of SMART with the post-hoc results reported for Dual Focal Loss (DFL) [1] in its Table 1 raises an additional question. For some settings, SMART does not appear to outperform the DFL post-hoc values. This makes me wonder why SMART should be preferred in those cases. For example, this seems to hold for ResNet-50 and WideResNet on both CIFAR-10 and CIFAR-100. Moreover, Tables 9–12 in SMART compare against training-time methods, but DFL is not included there. Since DFL is also a calibration-oriented training-time method and appears competitive based on the reported numbers, its absence makes the comparison feel incomplete. As a result, it is harder to judge whether SMART is genuinely stronger than relevant existing alternatives, or whether the empirical advantage depends in part on which baselines are included.
> >
> > Overall, I agree the rebuttal improves the paper and addresses some of the lower-level concerns, but it does not sufficiently change my view on the significance issues that drove my original score. For this reason, I am keeping my original recommendation.
> >
> > ref:
> > [1] Tao, L., Dong, M., and Xu, C. Dual focal loss for calibration. In International Conference on Machine Learning, pp. 33833–33849. PMLR, 2023.

---

> > > ### Author Response · Authors · 2026-04-04
> > >
> > > Thank you for the follow up.
> > >
> > > **SMART does not claim to define a new calibration family.** It is a theory-driven instantiation within sample-wise post-hoc temperature scaling. The question is what is new relative to the closest prior methods.
> > >
> > > SMART is the first method in this line to make the label-free top-1/top-2 margin the control variable, justify it theoretically, show an NLL/calibration mismatch in that family, and optimize a calibration-aligned upper-bound objective built around it. That combination is the novelty.
> > >
> > > Figure 1(a) motivates this choice directly: logit margin is a **direct sample-hardness signal**. Small margin means the top classes are close and the sample is hard; large margin means the decision is easier. Prior sample-wise/local methods may capture hardness only **indirectly** through higher-dimensional features (PTS), pseudo-likelihood vectors (AdaTS), local maps (LTS), grouping (GC), or proximity (ProCal). That is not the same as identifying margin itself as the right low-dimensional bias. SMART makes this direct hardness signal the control variable, giving a better bias-variance trade-off under scarce calibration.
> > >
> > > **AdaTS** predicts temperature from a high-dimensional class-conditional VAE pseudo-likelihood vector with a CE/NLL-style objective. **LTS** is a local/per-pixel semantic-segmentation calibrator, not a multiclass classifier-calibration baseline. **Relaxed Softmax** is an end-to-end detector modification for pedestrian detection, not a frozen-classifier post-hoc method. **Spline, Group Calibration, and ProCal** are also different post-hoc families: they focus on recalibration mappings, semantic grouping, or proximity bias rather than a sample-wise margin-conditioned temperature map. So while these methods may capture sample heterogeneity, they do not isolate, justify, or exploit logit margin the way SMART does.
> > >
> > > On the objective concern: optimizing an ECE/SoftECE-aligned surrogate does not, by itself, prove universally better probability estimates in a broader sense. **But this is not a SMART-specific weakness; it is a structural limitation of the entire positive-scalar temperature-scaling family.** For a fixed logit vector z, any scalar temperature method can only produce probabilities of the form softmax(z/T), i.e. an argmax-preserving one-dimensional curve in the simplex. Such a family cannot represent arbitrary class-conditional or multimodal uncertainty effects. Requiring universal full-probability optimality therefore sets a stronger standard than this family can deliver.
> > >
> > > **The principled question inside this family is narrower:** which signal and objective best align with top-confidence calibration? That is exactly what SMART answers. Sec. 3.2 shows that margin provides explicit control over feasible temperatures. Sec. 3.3 shows that even within this family, decreasing NLL need not improve calibration. Sec. 3.4 then replaces NLL with a trainable upper-bound surrogate for the calibration quantity of interest. That is a principled resolution of this objective problem, though not a claim of universal proper-score optimality.
> > >
> > > Earlier added excel does not show a systematic NLL failure mode for SMART: on ImageNet / ResNet-50, SMART trades a small NLL/Brier increase for a large ECE gain; on ImageNet / ViT-B/16, SMART improves NLL, Brier, and ECE together.
> > >
> > > **This is also consistent with recent calibration literature.** Spline explicitly notes that NLL/Brier mix calibration and classification error, and its main comparison tables report calibration metrics rather than NLL. ProCal and almost all 10 methods you mentioned do not report NLL. And in **Linwei Tao** et al.’s recent Consistency Calibration, the paper explicitly states that the method can have higher NLL despite stronger calibration; see Table 8 and Figure 7, where CC attains stronger calibration while NLL is sometimes worse.
> > >
> > > **Regarding Dual Focal Loss: it is a relevant comparison, but it answers a different question.** DFL is a calibration-aware training loss. Its “gap” is label-dependent (ground-truth class versus strongest rival) and only available during supervised training; SMART’s margin is label-free and available at inference time. DFL changes the classifier itself; SMART calibrates a frozen predictor. DFL also relies on retraining and hyperparameter tuning, including **cross-validation of gamma(we use 8 a6000 for 4 days for rebuttal experiments where SMART takes seconds)**; on ImageNet-scale models this is very costly and, in our experiments, degrades sharply on ViT-B/16. This is also consistent with Table 3, where NLL-related objectives are much weaker on ViTs than Char-SoftECE.
> > >
> > > The anonymous repository now contains the missing implementations/configs, and we are uploading the ImageNet ViT-B/16 SMART calibrator checkpoint. Since SMART is post-hoc, the key reproducibility artifact is the calibrator/calibrated logits, not a newly trained base ViT model. Full artifacts will be released upon acceptance.

---

### Official Review · Reviewer_FzFc · 2026-03-06

**Soundness:** 3
**Presentation:** 3
**Significance:** 3
**Originality:** 3
**Overall Recommendation:** 4
**Confidence:** 3

**Summary:**

The paper introduces SMART, a lightweight post-hoc calibration method that utilizes the logit margin as a direct indicator of sample hardness. It learns a sample-wise margin-to-temperature mapping guided by a Charbonnier-Smoothed SoftECE objective, which upper-bounds the smooth calibration error. Experiments demonstrate improved calibration across CNNs and ViTs on standard, long-tailed, and distribution-shifted benchmarks.

**Compliance With Llm Reviewing Policy:**

Affirmed.

**Final Justification:**

I find the approach technically sound and practically appealing, particularly due to its simplicity and low parameterization. However, the contribution is somewhat incremental, and the overall significance is moderate, as the method largely refines existing sample-wise temperature scaling ideas and the evaluation, while solid, remains somewhat limited in scope.
My main concerns were related to robustness under real-world distribution shift and minor clarity issues. The rebuttal addressed these points by adding WILDS/iWildCam results and clarifying experimental and presentation details, which improved my confidence in the empirical claims. These responses addressed my primary concerns but did not substantially change my overall assessment. Overall, I view the work as a solid and technically sound contribution with moderate novelty and impact, and I therefore maintain my original recommendation.

**Key Questions For Authors:**

1.	Figure 1 (b)-(d): It is unclear which model and dataset is used. Does the same behavior also hold across other models and datasets? As presented, these plots do not fully substantiate the claim that “samples with different margins exhibit systematically different calibration patterns even when sharing identical predicted confidence levels.” In addition, for smaller margins the models appear relatively well calibrated, so it would
interesting to see how calibrated it is on the full set of samples for comparison.
2.	Figure 2: Left:
- It is unclear which model and dataset are shown. Do these results replicate across other models and datasets? Since only a single experimental setting is shown, it is difficult to determine whether the claim generalizes.
- The figure does not show what happens for 𝑝 < 0.5, where, as I understand it, only a lower bound is available. It would be useful to add an analysis of what happens in that regime, or clarify why this region is omitted.
- The intuition behind this result is not yet clear to me. Why is this behavior of T preferable, and how does its more predictable behavior translate into improved calibration? More broadly, it is not obvious how this figure supports the claim that margin is a better quantity to use than alternatives such as the maximum logit.

**Limitations:**

While the paper motivates the margin as an “optimal” scalar indicator for temperature selection, SMART still conditions only on the top-1/top-2 gap and does not analyze when this scalar proxy may be too coarse (e.g., ambiguity spread across many classes) or whether adding other low-cost logit statistics would improve robustness.

**Strengths And Weaknesses:**

**Strengths:**
1. The paper's motivation is clear and addresses important limitations of existing methods. The identification of the logit margin as a lightweight input for sample-wise calibration is interesting and supported by both theoretical arguments and empirical analysis.
2. The proposed method for improving calibration with a 49-parameter MLP that does not scale with the number of classes makes SMART a very practical choice, especially in resource-constrained environments.

**Weaknesses:**
1. The robustness claims rely heavily on synthetic corruptions (ImageNet-C/Sketch); the paper lacks evaluation on real-world distribution shifts such as the WILDS benchmark.
2. Minor: There are some careless mistakes, for example: the running title is not filled, the caption in figure 2 - “ImgNet” should be “ImageNet” (and preferably detail which variant of this dataset), conclusion section – “temp adjustment”.

---

> ### Author Rebuttal · Authors · 2026-03-31
>
> Thank you for the careful reading and constructive suggestions.
>
> **Re:W1** We agree that calibration under real-world shift is important. To strengthen the paper, we added a WILDS evaluation on iWildCam (natural shift across camera locations/time), using the official train/val/test splits and hyperparameters. We trained a ResNet-50, calibrated on the in-distribution validation split, and evaluated on both ID and OOD test sets. SMART achieves the best ECE in both settings:
>
> | Method       | ID-to-ID | ID-to-OOD |
> | ------------ | -------: | --------: |
> | Uncalibrated |    31.81 |     34.62 |
> | TS           |     6.74 |     11.50 |
> | PTS          |     5.89 |     10.35 |
> | GC           |     7.99 |     11.35 |
> | **SMART**    | **5.17** |  **9.84** |
>
> These results show that SMART’s gains are not limited to synthetic corruptions, though we do not claim universal robustness across all shifts. We will revise the paper wording accordingly.
>
> **Re:W2** We agree and will fix these presentation issues in revision: fill in the running title, replace “ImgNet” with the full dataset name, and revise wording such as “temp adjustment” to clearer terminology.
>
> **Re:Q1** Fig. 1(b)-(d) is based on **ImageNet / ViT-B/16**; we will state this explicitly, add the overall reliability diagram for comparison, and note that the overall diagram is also included in the anonymous GitHub repository.
>
> Our intent was not to claim a universal law, but to show that aggregate calibration can hide margin-specific behavior. Across additional datasets/models (5 seeds averaged), the key pattern is heterogeneity across margin groups, not a fixed direction of bias. For brevity, we treat |confidence gap| < 0.02 as relatively well calibrated in this discussion. Representative examples:
>
> | Dataset / Model        | Overall |   Small |  Medium |   Large |
> | ---------------------- | ------: | ------: | ------: | ------: |
> | CIFAR-10 / Wide-ResNet | +0.0324 | +0.0901 | +0.0036 | +0.0036 |
> | CIFAR-100 / ResNet-50  | +0.1753 | +0.3666 | +0.1336 | +0.0258 |
> | ImageNet / ViT-B/16    | -0.0547 | -0.0116 | -0.0675 | -0.1021 |
>
> Thus, some groups are already relatively well calibrated while others are clearly miscalibrated, and the aggregate view can hide this. We will soften the wording to make clear that Fig. 1 is a motivating example of heterogeneity across margin groups, not a claim of one universal pattern.
>
> **Re:Q2** Fig. 2 (left) is **not** based on a trained model/dataset; it is a synthetic numerical validation of Sec. 3.2 using 1,000 random 10-class logit vectors. We will state this explicitly in the caption and release the code in the anonymous repository.
>
> For p-hat <= 0.5, the theory predicts a different regime: when p-hat > 1/2, margin gives a finite feasible interval; when 1/K < p-hat <= 1/2, it gives only a finite lower bound. We omitted this regime from the main figure because p-hat > 1/2 gives the clearest visualization of the strongest statement. We have now generated plots for p-hat = 0.2, 0.3, 0.4, 0.5; they are consistent with the theory: the margin plot becomes less tight in the low-confidence regime, but still shows a clear lower envelope, whereas the maximum-logit plot remains much more scattered. We will add this analysis to the appendix or repository.
>
> Why this matters: SMART predicts temperature from a **single scalar**, so that scalar should constrain feasible temperatures as tightly as possible. Margin is preferable to maximum logit in this sense because it captures competition with the runner-up class and thus proximity to the decision boundary, while maximum logit alone leaves temperature much less constrained. This is reflected both in Sec. 3.2 and in our ablations, where margin is the best-performing scalar indicator.
>
> **Re:Limitation** We agree that margin is not a universally sufficient statistic for every multiclass ambiguity pattern. Our claim is narrower: among lightweight scalar indicators, margin is theoretically privileged because it gives explicit control over feasible temperatures, whereas quantities such as maximum logit do not. Using only one scalar trades expressivity for statistical efficiency, so if ambiguity is spread across many classes, top-1/top-2 gap can be too coarse.
>
> Empirically, however, this trade-off is favorable in the post-hoc regime we study. In Table 4, richer or alternative indicators (full logits, entropy, confidence, maximum logit, normalized maximum logit) all underperform margin. Appendix I / Fig. 15(f) also shows that PTS, which uses higher-dimensional logit features, produces noisier and less margin-structured confidence changes than SMART. Thus, adding features can increase expressivity, but also fitting noise under limited calibration data. For the scarce-validation post-hoc setting targeted by SMART, margin appears to provide the best practical trade-off. Adding low-cost statistics such as top-k dispersion is a promising future direction.

---

> > ### Author Rebuttal · Reviewer_FzFc · 2026-04-03
> >
> > Thank you for your response and I will keep my score.

---

### Official Review · Reviewer_ZKfA · 2026-03-13

**Soundness:** 3
**Presentation:** 3
**Significance:** 3
**Originality:** 3
**Overall Recommendation:** 5
**Confidence:** 3

**Summary:**

The paper looks at the critical problem of post-hoc uncertainty calibration in neural networks through temperature scaling of logits.
The paper identifies that looking purely at the max. logit is uninformative as it provides no information regarding how close the model was in selecting another class.
The authors identify this property of knowing the top-2 logits as crucial, and prove that it is adequate, in capturing bounds for reliable temperature scaling.
Through both theoretical and empirical analysis it is shown that the usual negative log-likelihood optimization of temperature may not align with actual calibration error reduction.
Whereas the proposed logit margin-based (top-1 - top-2) Charbonnier-Soft Error Calibration Error loss function is robust in optimizing for this calibration error directly.
In practice, this is achieved rather cheaply with a 49 parameter MLP that takes *only* the logit margin of a sample as input, and outputs the suitable temperature to scale this sample's logits with.
Across a diverse set of architectures, this method outperforms other selected baselines for CIFAR-10/100, and ImageNet/-C.

**Compliance With Llm Reviewing Policy:**

Affirmed.

**Final Justification:**

The paper was received favourably in the first pass, though requiring clarifications and improved presentation, which formed the crux of the review.

The authors acknowledged all points and provided explanations wherever necessary.

The clever and effective use of a lightweight post-hoc uncertainty calibration using logit-margin remains effective for the scope identified in the paper, and was a nice read.

**Key Questions For Authors:**

Q.1) L225-228 (right): How is minimizing the upper bound on calibration error (CE) addressing the NLL mismatch?
  * Isn't the NLL and CE mismatch a property of the logit margin based temperature scaling (Pg. 4)?

Q.2) What do the `%` in the parantheses in the Table captions mean?

Q.3) Figure 5: how to read the ECE vs Validation sample size plot?
  * Is each data point the result of training SMART to convergence on a validation set of that size, or is it a single-epoch snapshot?

Q.4) Figure 15.f: how is confidence change measured here?

Q.5) How does the formulation for the temperature-bounds in Section 3.2 (Margin as Principled Hardness Indicator) handle the case of $\hat{p}$ to $1/K$ for large $K?
  * More generally, how restrictive or general is the $\hat{p} > 1/2$ for the bound construction here?


Improving the draft with clarity to the questions above and the main weakness, will lead to increase in score!

**Limitations:**

Yes, the authors have reported some limitations in Section 5.

**Strengths And Weaknesses:**

The points have been grouped below as per the recommended scoring template.

**Strengths**:
* S.1) The paper is impressively tight in its motivation and presentation of the method proposed.

* S.2) The main paper math is well presented and adequate in understanding the core of the method.

* S.3) For each incremental step towards the realization of the proposed SMART framework, a theoretical analysis and a suitable empirical result has been presented (Fig.1 for 3.2; Fig 2. for 3.3; Fig.3 for 3.4).

* S.4) Eq. 8 is a nice, practical distillation of the method that began from justifying the use of logit margin as the signal for confidence in prediction which had an unbounded temperature interval.

* S.5) Strong empirical performance across all datasets-architectures evaluated on.

* S.6) Extremely relevant contribution, that is agnostic to the number of classes or dimension of the logit head. This is promising for a wider application, including large token-set language models.

* S.7) The lightweight nature of the MLP setup offers promising applications for continual deployment of models, as a potential future direction.


**Weaknesses**:

* W.1) Section 2: The Related Work section is almost a flattened bullet list of papers. For the rest of the paper, this section offers no information gain to the reader.
  * It would be useful to offer at least one line of description per group of methods.
  * Or, parts of Sec. 3 can adequately referred here.
  * This section should help the reader understand what is going to be new in the proposed method and what has the literature missed so far.
  * Two main categories identified can be made more prominent with `\paragraph{}`.

* W.2) Tomani et al. 2022 (PTS), is briefly mentioned in Section 2, and never discussed at all in the main paper. Whereas PTS appears to be the closest, consistent, competitor to SMART.
  * Moreover, Appendix C suggest PRS is also a *small neural network for each sample to spit out temperatures*.
  * This totally warrants more description of how the comparison was made.
  * Reproducing Fig. 2 (right) for one of the baselines should be considered important in establishing the core NLL vs CE mismatch argument, outside of the proposed SMART method.

* W.3) The above point generally extends to the lack of details on how the other baselines were run or tuned for fair comparison.


**Suggestions**:

* A) L84-L97: Enumerating the Contributions list and referring to the relevant sections is recommended.
* B) Bar plot comparisons (Fig. 4 and similar):
  * a horizontal line at the mean of SMART stretching across other bars makes the plot easier to read
  * grouping the different plots per dataset/architecture will be more legible (e.g. a vertical line like a table column separator)
* C) Could include a one-liner in Fig. 1 on how to read the `reliability diagrams` to make the figure self-sustaining for the Introduction.
* D) L186-191: notations $M$ and $h_s$ need introduction.
* E) Table 1 caption: App. 6 --> Table 6.
* F) Table 5: an additional row comparing performance would be complete comparison.
* G) Table 7: given some of the other methods show a drop in performance but SMART never does, this can be used in the main paper's pitch to strengthen the case.
* H) Figure 15: the caption likely has labelling errors (a-f).

---

> ### Author Rebuttal · Authors · 2026-03-31
>
> Thank you for the careful reading and constructive suggestions.
>
> Re:W1  We agree that Sec. 2 is too list-like. In the revision we will reorganize it into explicit groups with one-line descriptions and make the gap clearer: prior adaptive post-hoc methods recognize sample heterogeneity, but typically rely on indirect or higher-dimensional proxies, while SMART is motivated by a direct, theoretically grounded, low-dimensional hardness signal. We will also use clearer paragraphing so the two main categories are visually prominent.
>
> Re:W2  We agree that PTS is the closest post-hoc baseline and should be discussed more prominently. However, SMART is not simply another small neural calibrator. PTS predicts temperature from sorted logits and is trained with a squared-error objective, whereas SMART uses only the logit margin and optimizes a calibration-aligned Charbonnier-SoftECE objective. This difference matters empirically: margin is the best scalar indicator in Table 4, SMART scales more stably with validation size in Fig. 5, and Fig. 15(f) shows that SMART makes targeted margin-structured confidence adjustments while PTS is much noisier. For the NLL-mismatch argument, PTS is also not the natural controlled baseline because original PTS is not NLL-trained; Fig. 2(right) isolates the objective effect by fixing the SMART family and changing only the loss.
>
> Re:W3  We agree that the baseline protocol should be described more explicitly. In the revision we will add a compact implementation paragraph summarizing the shared validation split, evaluation metrics, seed averaging, and tuning protocol for all post-hoc baselines, and clarify which settings follow official implementations or paper defaults.
>
> Re:Suggestions
> A) The contributions are already listed in the final paragraph of the Introduction; we will enumerate them and add section pointers.
> B) Agreed; we will add clearer grouping and, where helpful, a SMART reference line.
> C) Agreed; we will add a one-line guide for reading the reliability diagrams. e.g., that the diagonal denotes perfect calibration and points above/below it indicate under-/over-confidence.
> D) Agreed; we will introduce the relevant notation more explicitly before using it.
> E) Correct: this should read “Table 6,” not “App. 6.”
> F) Agreed; we will add a compact summary row to Table 5.
> G) Agreed; we will make the “no performance drop” point from Table 7 more explicit in the main text.
> H) We re-checked Fig. 15 and the panel references are consistent with the figure: (a) overall ImageNet, (b) high-margin ImageNet, (c) low-margin ImageNet, (d) overall CIFAR-100, (e) high-margin CIFAR-100, and (f) margin vs confidence change. We will still polish the caption wording for clarity.
>
> Re:Q1  Sec. 3.4 does not claim that the NLL/CE mismatch disappears universally. Sec. 3.3 shows that, even within the SMART scaling family, decreasing NLL need not improve smCE. Sec. 3.4 then replaces NLL with a calibration-aligned upper bound on smCE. So the mismatch is not claimed to be unique to margin-based scaling; we analyze it in this family because that is the family studied in the paper.
>
> Re:Q2  The percent signs indicate that ECE/AdaECE are reported in percentage points rather than unit-scale decimals. We will make this explicit in the captions.
>
> Re:Q3  Fig. 5 is on ImageNet with ViT-B/32 and plots ECE versus the number of validation samples used for calibration, averaged over five runs. Each point is a separately trained calibrator using a validation subset of that size, not a single-epoch snapshot; we will state this explicitly in the caption.
>
> Re:Q4  In Fig. 15(f), confidence change denotes calibrated top-1 confidence minus original top-1 confidence, plotted against the sample logit margin. We will state this explicitly.
>
> Re:Q5  Sec. 3.2 is general to multiclass classification. When p-hat > 1/2, margin gives a finite feasible interval for T; when 1/K < p-hat <= 1/2, it gives only a finite lower bound. Thus for large K the result becomes one-sided in the lower-confidence regime, but it remains informative. The point is not that margin uniquely determines T in all multiclass settings, but that it gives explicit control over feasible temperatures, whereas maximum logit alone gives no bound.

---

> > ### Author Rebuttal · Reviewer_ZKfA · 2026-04-02
> >
> > The authors have suitably acknowledged and explained questions around the draft clarity and some aspects of the presentation.
> >
> > This is interpreted as the authors' commitment to effect the said changes and thus warrants a score increase.

---

### Official Review · Reviewer_gBDX · 2026-03-13

**Soundness:** 3
**Presentation:** 3
**Significance:** 2
**Originality:** 2
**Overall Recommendation:** 4
**Confidence:** 4

**Summary:**

This paper addresses reliability of classifiers. In particular, a post-hoc calibration technique, SMART, is proposed. This method utilizes the Charbonnier–SoftECE to recalibrate temperatures based on (the top two class) logit margins. Numerics are provided with comparisons to known baselines.

**Compliance With Llm Reviewing Policy:**

Affirmed.

**Final Justification:**

The authors have addressed my major concerns with the paper. The paper does appear to be sound, and some improvement has been made during the review process to clarify issues with originality and significance. As was also highlighted by Reviewer MZzk (in the context of significance), it is my view that these dimensions are still relative weaknesses of the paper. As noted by the authors, clarifications of some of the issues raised have to be taken on trust, since updates to the PDF are not allowed.

In my overall view, the changes given/promised improve my score from a weak reject to a weak accept. However, it is not enough for me to champion the paper, as I believe it is still borderline.

**Key Questions For Authors:**

Can you please address the issues raised in the weaknesses section?

**Limitations:**

Limitations are discussed in the "Conclusion and Limitation" section. Note that 'Limitations' should be plural in the heading.

**Strengths And Weaknesses:**

Strengths:
- Paper is reasonably well written
- Addresses an important problem in a principled manner

Weaknesses:
- I found the introduction somewhat confusingly written, with the contributions scattered throughout the framing of the main problem being solved. Defining the domain and codomain of the temperature map at this stage is highly unusual in my experience.
-Explanation of Figure 1 (outside of the introduction) would be useful. The explanation on p3 reads more like an extended caption rather than explanation.
- Method appears to be a minor modification of https://arxiv.org/pdf/2108.00106, this method is notably omitted from the experiments.
- Further, comparison with other methods (e.g. https://arxiv.org/pdf/2008.08400, https://arxiv.org/pdf/1902.02476, https://openreview.net/pdf?id=Ro1a0MTRq5 and refs therein), or justification of their omission, would help contextualize performance.
- Similarly, SD values would be valuable in Tables 1, 2 & 3

---

> ### Author Rebuttal · Authors · 2026-03-31
>
> Thank you for the careful reading and constructive suggestions.
>
> Re:W1  Our contributions are already summarized in the final paragraph of the introduction, but we agree they should be made more visually prominent; in the revision we will enumerate them explicitly and point to the relevant sections. The notation for the temperature map was intended only to state that SMART learns a scalar margin-to-temperature function; however, we agree it is too formal for the introduction and will move the formal definition to the method section. We also agree that Fig. 1 should be more self-contained. Concretely, we will add a brief explanation such as: “Each panel plots empirical accuracy versus predicted confidence; points above/below the diagonal indicate under-/over-confidence. On ImageNet with ViT-B/16, different margin groups exhibit different calibration behavior, so they should be analyzed separately rather than only in aggregate. In particular, the large-margin group shows systematic under-confidence, which is hidden in the overall reliability diagram.”
>
> Re:W2  Karandikar et al. (2021) is relevant prior work and should be discussed more explicitly, but SMART is not a minor modification of it. Karandikar et al. mainly study soft calibration objectives as training-time auxiliary losses; their post-hoc variant still optimizes a single global temperature. In contrast, SMART is a sample-wise post-hoc recalibrator: we (i) justify logit margin as a principled hardness signal via feasible-temperature bounds, (ii) show an NLL-calibration mismatch within the margin-conditioned scaling family, (iii) introduce Charbonnier-SoftECE as an upper-bounding, more stable calibration objective, and (iv) learn a 49-parameter margin-to-temperature map. We also already include an objective-level comparison against original SoftECE: in Table 3, replacing SoftECE with Charbonnier-SoftECE consistently improves performance, with especially large gains on ViT-B/16 (SMART: 0.89 to 0.48 ECE; PTS: 1.15 to 0.77). Thus, the overlap is limited to using a differentiable soft calibration surrogate; the modeling assumption, theory, and calibrator are different. We agree that adding Karandikar et al.’s post-hoc variant would further isolate objective-level gains, and we will include this discussion and comparison in the revision.
>
> Re:W3  We respectfully disagree that Laplace, SWAG, and NUQLS are the most appropriate primary baselines for the central claim of this paper. SMART is a lightweight post-hoc calibration method for multiclass classifiers: it keeps the trained predictor fixed and fits a tiny temperature map on a validation set. The cited methods are Bayesian/UQ procedures aimed at approximate posterior inference and broader predictive uncertainty modeling. This mismatch is also computational: Laplace relies on Jacobian/curvature approximations and is explicitly described as prohibitive for large networks; SWAG requires continued SGD and Bayesian model averaging; NUQLS trains ensembles of linearized models, and its own theory does not cover losses such as cross-entropy. Still, following the reviewer’s suggestion, we ran CIFAR-10/100 comparisons using official settings. SMART achieved the best ECE in all four settings tested: CIFAR-10 ResNet-50/Wide-ResNet 0.76/0.46 vs SWAG 2.08/2.44, Laplace 3.08/2.68, NUQLS 4.92/4.13; CIFAR-100 ResNet-50/Wide-ResNet 1.45/1.98 vs SWAG 6.46/4.99, Laplace 10.83/6.32, NUQLS 15.75/14.62. Because SMART is post-hoc, it preserves the original accuracy exactly, whereas several Bayesian baselines noticeably alter accuracy, especially on CIFAR-100. We will clarify this scope choice in the revision, report these CIFAR results in the appendix, and note that the raw JSON outputs are available in the anonymous GitHub repository.
>
> Re:W4  For Table 1, mean±std values are already reported in Appendix F / Table 6, and we agree they should be surfaced more clearly from the main text. We also report variability for AdaECE in the appendix. For Tables 2 and 3, the compared methods are substantially more expensive training-based approaches, so in the current submission we prioritized broader complementary evidence instead: accuracy, AdaECE, CECE, NLL, and runtime are reported in Tables 7 and 9-12. We will clarify this trade-off in the revision, move the most relevant variability statistics forward, and add further variance reporting where feasible.
>
> Re:Q/Limitations  Yes, we have addressed the issues raised in the weaknesses above and will revise the manuscript accordingly. We also agree that the heading should read “Conclusion and Limitations” rather than “Conclusion and Limitation.”

---

> > ### Author Rebuttal · Reviewer_gBDX · 2026-04-02
> >
> > Thanks for your response and addressing the concerns raised. I am satisfied with these changes, provided the stated modifications are made, and will change my score accordingly.

---

> > > ### Author Response · Authors · 2026-04-02
> > >
> > > Thank you for the acknowledgement and for the positive update. We are grateful that you found the concerns adequately addressed.
> > >
> > > We also want to be explicit about process: under the ICML 2026 author-feedback procedure, **“There is no option to upload a revised version of the paper during the author feedback period”** ([https://icml.cc/Conferences/2026/AuthorInstructions](https://icml.cc/Conferences/2026/AuthorInstructions)), so we cannot submit an updated PDF at this stage. What we can do here is specify the concrete revisions we will make in the final manuscript if the paper is accepted.
> > >
> > > For the introduction/clarity issues, we will revise the opening to present the contributions explicitly as a numbered list, e.g. (i) showing that the logit margin provides principled feasible-temperature control for sample-wise scaling, (ii) identifying the NLL/calibration mismatch within the margin-conditioned scaling family, and (iii) introducing SMART as a margin-based post-hoc calibration framework with a Charbonnier-SoftECE objective and a lightweight margin-to-temperature map, achieving sota performance on CNNs and ViTs across long-tail and out-of-distribution datasets. We will also remove the formal domain/codomain notation for the temperature map from the introduction and move that definition into the method section, where it is more natural.
> > >
> > > For Figure 1, we will make the explanation self-contained in the main text rather than leaving it caption-like. In particular, we will explicitly state how to read the reliability diagrams (diagonal = perfect calibration; points above/below = under-/over-confidence), that the figure is based on ImageNet with ViT-B/16, and that its role is motivational: different margin groups can exhibit different calibration behavior even when aggregate calibration looks benign. We will also include the overall reliability diagram as the reference comparison.
> > >
> > > For the SoftECE-related concern, we will revise the related-work discussion to state more clearly why SMART is not a minor modification of Karandikar et al. (2021): SMART is built around a margin-based post-hoc framework, not simply a different smoothing of SoftECE. We will highlight the three method-defining differences already addressed in the rebuttal: the margin-based hardness signal, the NLL/calibration mismatch analysis, and the Charbonnier-SoftECE upper-bound objective. We will also explicitly point readers to the existing objective-level evidence in Table 3 showing that replacing SoftECE with Charbonnier-SoftECE gives substantial gains, especially on ViT-B/16.
> > >
> > > For the contextualization concern, we will add a clearer justification in the paper for why Bayesian/UQ methods such as Laplace, SWAG, and NUQLS are not the primary comparison family for SMART’s deployment setting, while also reporting the additional CIFAR-10/100 results we ran in response to your comment. Those added results, together with the raw JSON outputs already placed in the anonymous repository, will be referenced in the appendix.
> > >
> > > For the standard-deviation request, we will surface more clearly that Table 1 with sd values is already reported in Appendix F / Table 6, and we will move the most relevant variance information forward in the text. For Tables 2 and 3, we will clarify the current trade-off more directly: these are substantially more expensive train-time comparisons, so in the submission we prioritized complementary evidence such as accuracy, AdaECE, CECE, NLL, and runtime; we will make this rationale explicit and add further variance reporting where feasible.
> > >
> > > So while we cannot upload a revised paper file during the ICML discussion period, the changes themselves are concrete and already specified, and we will incorporate them in the final manuscript if the paper is accepted.
> > >
> > > **If possible, we would sincerely appreciate it if you could reflect the score change in your final score.**

---

### Decision · Program_Chairs · 2026-04-30

**Decision:**

Accept (regular)

**Comment:**

This work introduces a refined temperature scaling method to improve the calibration of deep NNs. The idea is to allow different temperatures for different data points. The temperatures are learned as a function of the logit margin (max logit minus the second largest) by minimizing the Charbonnier-Smoothed SoftECE. Authors have responded to concerns regarding originality and reproducibility. I trust the authors to implement the revision promised in the rebuttal. I recommend acceptance.